# When Can You Poison Rewards?
# A Tight Characterization of Reward Poisoning in Linear MDPs

Jose E. Aguilar Escamilla [* 1]   Haoyang Hong [* 1]   Jiawei Li [* 2]   Haoyu Zhao [3]   Xuezhou Zhang [4]
Sanghyun Hong [1]   Huazheng Wang [1]

## Abstract

We study reward poisoning attacks in reinforcement learning (RL), where an adversary manipulates rewards under a limited budget to induce a target agent to learn a policy aligned with the attacker's objectives. Most prior work focuses on *constructing* successful attacks, providing sufficient conditions under which poisoning is effective, while offering limited understanding of when such targeted attacks are fundamentally infeasible. In this paper, we provide the first characterization of reward-poisoning attackability in linear MDPs, establishing both necessary and sufficient conditions for whether a target policy can be induced within a bounded attack budget. This draws a clear boundary between the *vulnerable* RL instances and *intrinsically robust* ones which cannot be attacked without large costs even when the learner uses standard, non-robust RL algorithms. We further demonstrate our framework beyond synthetic linear MDPs by approximating deep RL environments as linear MDPs. We show that our theoretical framework effectively distinguishes vulnerability, demonstrating how our theoretical predictions have practical significance.

## 1. Introduction

Reinforcement learning (RL) has become a cornerstone for many intelligent systems, including autonomous driving (Kiran et al., 2022; Tang et al., 2024), healthcare (Yu et al., 2021; Al-Hamadani et al., 2024), recommender systems (Afsar et al., 2022; Yu et al., 2024), and human-feedback-driven

---
[*]Equal contribution   [1]Oregon State University, OR, USA [2]University of Illinois Urbana–Champaign, IL, USA [3]Princeton University, NJ, USA [4]Boston University, MA, USA. Correspondence to: Jose E. Aguilar Escamilla <aguijose@oregonstate.edu>, Haoyang Hong <honghao@oregonstate.edu>, Jiawei Li <jiaweil9@illinois.edu>.

*Proceedings of the 43rd International Conference on Machine Learning*, Seoul, South Korea. PMLR 306, 2026. Copyright 2026 by the author(s).

policy optimization (Cao et al., 2024a;b). A central—yet much more difficult—design choice in RL is the specification of reward functions (Banihashem et al., 2022). In many modern applications, reward signals are derived from human feedback or interaction data, which may be noisy, imperfect, or strategically manipulated.

This reliance exposes a critical vulnerability: *reward poisoning attacks*. In such attacks, an adversary observes the interaction between the learning algorithm (agent) and the environment and modifies reward signals to misguide the agent to learn a policy of the adversary's interest. In scenarios like recommended systems adapting to user preferences or online decision-making platforms, adversaries could subtly manipulate the reward signals, intentionally or inadvertently, during interactions (Zhang et al., 2020; Xu et al., 2024; Tan et al., 2024). Securing RL systems required principled understanding of how reward poisoning attacks can be constructed, but also of when they are *possible*.

Research on reward poisoning has been studied in multi-armed bandits (Wang et al., 2022; Xu et al., 2021; Rangi et al., 2022a) and reinforcement learning (Zhang et al., 2020; Rakhsha et al., 2021; Xu et al., 2024), theoretically and empirically. Most studies focus on designing attack algorithms under specific conditions (Balasubramanian et al., 2024; Wang et al., 2024; Liu & Shroff, 2019) or developing robust RL methods that tolerate bounded reward manipulation (Wu et al., 2023; Xu et al., 2023; Rangi et al., 2022b). More recently, a distinct line of work has investigated *intrinsic robustness*: the idea that certain decision-making problems are inherently resistant to reward poisoning, independent of the learning algorithm (Wang et al., 2022; Balasubramanian et al., 2024). While intrinsic robustness has been studied in linear (Wang et al., 2022) and combinatorial bandit (Balasubramanian et al., 2024) settings, its existence and structure in general RL remain largely under-explored.

In this paper, we address this gap by formally studying vulnerability and intrinsic robustness in *finite-horizon linear Markov decision processes (linear MDPs)*. In our context, *intrinsic robustness* refers to the inability by a sublinear budget attacker to influence the optimal state-action choices at will. This novel perspective exposes an additional factor in-

fluencing adversarial robustness: robustness is *also* a result of the underlying environment's intrinsic geometry, not just the algorithm robustness features. We make the following contributions on the intrinsic robustness of linear MDPs:

**Contribution 1**: **A characterization of vulnerability and intrinsic robustness in linear MDPs.** We show that reward-poisoning attackability in linear MDPs admits a complete characterization. Specifically, whether a given instance is attackable can be decided by solving a *convex quadratic program (CQP)*. Intuitively, the CQP tests whether an adversary can construct a reward perturbation that shifts the optimal Q-function in their favor within the linear feature span of the respective MDP.

When this CQP is infeasible, the instance exhibits *intrinsic robustness*: any successful attack requires a budget that grows linearly with the horizon $T$, even against standard non-robust algorithms (e.g., vanilla LSVI-UCB (Jin et al., 2020)). When the CQP is feasible, it yields an efficient reward-poisoning strategy against a broad class of no-regret learners. Our results complement prior findings of intrinsic robustness in bandit problems (Balasubramanian et al., 2024; Wang et al., 2022) and the global budget thresholds reported in RL settings (Zhang et al., 2020; Rangi et al., 2022b). For clarity, our characterization adopts a *pointwise* (state–action–stage) optimality notion that we make explicit in the technical sections.

**Contribution 2**: **White-box and black-box attack procedures.** Building on the characterization, we design budget-efficient white-box and black-box attack methods. In the white-box setting we provide theoretical guarantees showing sublinear attack budgets on attackable instances. In the black-box setting, we employ a sampled-constraint and parameter estimation to approximate the linear MDP model and then apply the same CQP-based test and attack; this approximation-based method is validated empirically, and we explicitly state the assumptions under which the approximation preserves attackability prediction validity.

**Contribution 3**: **Empirical evidence on RL benchmarks.** We empirically validate that the characterization is predictive of robustness in practical RL tasks. Although real-world environments often break linear MDP assumptions, we hypothesize that attackability is approximately preserved under learned linear representations. We use contrastive learning (Zhang et al., 2022) to obtain a linear representation of the environment and solve the above CQP on the learned representation. Across environments characterized as robust vs. attackable (or vulnerable), we observe markedly different poisoning difficulty and outcomes, which supports the practical relevance of our framework.

**Conflict of Interest Disclosure** The authors declare that they have no known competing financial interests or personal relationships that could appear to influence the work reported in this paper.

## 2. Related Work

**Reward Poisoning Attacks.** The literature on reward poisoning spans three threads: (i) constructing attack procedures, (ii) defenses, and (iii) theoretical characterizations. Attacks have been investigated in single-agent RL (Zhang et al., 2020; Rakhsha et al., 2021; Wu et al., 2023; Xu et al., 2023; 2024; Rangi et al., 2022b; Lin et al., 2017), multi-agent RL (Wu et al., 2023; Ma et al., 2022; McMahan et al., 2024a), and bandits (Wang et al., 2024; Liu & Shroff, 2019; Xu et al., 2021; Rangi et al., 2022a). Defenses include robust training and detection schemes (Banihashem et al., 2023; Nika et al., 2023; Bouhaddi & Adi, 2024). On the theory side, intrinsic robustness has been characterized for linear and combinatorial bandits (Wang et al., 2022; Balasubramanian et al., 2024), and lower/upper bounds have been derived for RL settings under poisoning (Nika et al., 2023; Rakhsha et al., 2021). Unlike bandits, existing RL literature has not provided an instance-level criteria that separates attackable environments from intrinsically robust ones in an algorithm-agnostic manner. Some related work like McMahan et al. (2024b) have studied similar settings where state, observation, action, and reward attacks are modeled via meta-MDPs.

Other works have shown that bounded-budget reward poisoning can be infeasible in tabular MDPs (Zhang et al., 2020; Rangi et al., 2022b). Our study complements these results by working in the linear MDP framework: we provide a test via a convex quadratic program (introduced later) that decides whether an instance is attackable or intrinsically robust. Tabular MDPs appear as a special case of linear representations, and our discussion explains when previously observed budget thresholds are recovered. For clarity and comparability with prior RL poisoning papers, our main analysis adopts a pointwise (state–action–stage) notion of optimality for the target policy; extensions to stochastic policies are discussed in the technical sections.

**Other Adversarial Attacks on RL.** Beyond reward manipulation, adversaries may perturb states before the agent observes them (Sun et al., 2021) or actions before they are executed (Liu & Lai, 2021; Rangi et al., 2022b); related multi-agent variants have also been studied (Zheng et al., 2023; Mohammadi et al., 2023; Wu et al., 2024). These attack surfaces are orthogonal to our focus on reward-only perturbations. In our black-box pipeline we follow the standard estimation-then-attack paradigm: we first approximate the environment and then apply the same test/attack as in the white-box case. Consistent with prior estimation-based approaches, the black-box guarantees are primarily empirical, whereas our formal statements concern the white-box

setting.

## 3. Problem Formulation

We follow the finite-horizon online-learning setting of Jin et al. (2020): An agent interacts with an unknown linear MDP $(\mathcal{S}, \mathcal{A}, H, \{\mathbb{P}_h, r_h\}_{h=1}^H)$ for $T$ episodes while minimizing cumulative regret.

**Definition 3.1** (Linear MDP and occupancy measure). Let the feature map be $\phi : \mathcal{S} \times \mathcal{A} \to \mathbb{R}^d$ with $\|\phi(s, a)\|_2 \leq 1$. For each stage $h \in [H]$, the reward and transition have linear forms:

$$r_h(s, a) = \phi(s, a)^\top \theta_h, \quad \|\theta_h\|_2 \leq \sqrt{d}, \quad r_h(s, a) \in [0, 1],$$

and there exist probability distributions $\{\mathbb{P}_h^{(i)}(\cdot)\}_{i=1}^d$ on $\mathcal{S}$ such that

$$\mathbb{P}_h(\cdot \mid s, a) = \sum_{i=1}^d \phi_i(s, a) \, \mathbb{P}_h^{(i)}(\cdot).$$

Under this model (see Jin et al., 2020), the optimal value recursion implies that, for each $h$, there exists $w_h \in \mathbb{R}^d$ with $Q_h^*(s, a) = \phi(s, a)^\top w_h$.

Let $\rho$ be the initial-state distribution. For a (possibly stochastic) policy $\pi = \{\pi_h(\cdot \mid s)\}_{h=1}^H$, the occupancy measure at step $h$ is

$$d_h^\pi(s) = \Pr\big[s_h = s \mid s_1 \sim \rho, \ a_t \sim \pi_t(\cdot \mid s_t), \ 1 \leq t < h\big],$$

and it satisfies the recursion $d_1^\pi = \rho$ and

$$d_{h+1}^\pi(s') = \sum_{s \in \mathcal{S}} \sum_{a \in \mathcal{A}} d_h^\pi(s) \, \pi_h(a \mid s) \, \mathbb{P}_h(s' \mid s, a).$$

Throughout, our characterization adopts a *pointwise* (state–action–stage) optimality notion for the target policy; extensions to stochastic policies are discussed later.

In words, a linear MDP is a Markov decision process whose reward and transition kernels are linear in a bounded feature map. The linearity concerns the parameterization of $r_h$ and $\mathbb{P}_h$ rather than the feature map $\phi$ itself, which can be learned or predefined. Tabular MDPs are included as a special case, just as linear bandits include classical stochastic bandits.

**Definition 3.2** (Tabular MDP). An MDP $(\mathcal{S}, \mathcal{A}, H, \mathbb{P}, r)$ is *tabular* if it has finite state and action sets, i.e., $|\mathcal{S}| < \infty$ and $|\mathcal{A}| < \infty$, with episode length $H \geq 1$, transition kernel $\mathbb{P}$, and reward $r(s, a) \in [0, 1]$.

### 3.1. Threat Model

We formalise the attackability criterion under a *white-box* adversary. Consider a linear MDP $(\mathcal{S}, \mathcal{A}, H, \mathbb{P}, r)$. The adversary has a target policy $\pi^\dagger$ and wants the learner to execute $\pi^\dagger$ instead of the environment's optimal policy $\pi^*$ by modifying the observed rewards. At each episode $t$ and step $h$, after observing the current state $s_{h,t}$, action $a_{h,t}$, and feedback $r_h(s_{h,t}, a_{h,t})$, the adversary outputs a perturbed reward $\hat{r}_h(s_{h,t}, a_{h,t})$ that is fed to the learner. The total attack cost is

$$C = \sum_{t=1}^T \sum_{h=1}^H \big| r_h(s_{h,t}, a_{h,t}) - \hat{r}_h(s_{h,t}, a_{h,t}) \big|.$$

Unless otherwise stated, an attack is deemed successful when, under the perturbed rewards, the target policy is optimal in the pointwise sense at every stage and state reachable by the learner. This criterion matches the strong notion used in prior RL poisoning works and makes our results comparable. A practical instance is training on an untrusted simulator where reward channels can be tampered with, potentially inducing backdoors or trojans (Gong et al., 2024; Rathbun et al., 2024; Kiourti et al., 2020).

## 4. Theoretical Characterization of Vulnerability/Intrinsic Robustness

We first define attackability under a pointwise, support-restricted criterion for a fixed target policy in a linear MDP. We then introduce our main white-box characterization under this criterion, followed by a white-box attack inspired by it.

**Definition 4.1** (Attackability of a Linear MDP). A finite-horizon linear MDP is *attackable* w.r.t. a target policy $\pi^\dagger$ if, for any no-regret learner, there exists an attack strategy with total cost $o(T)$ that makes the learner follow $\pi^\dagger$ in at least $T - o(T)$ episodes for every state-stage pair $(h, s)$ with occupancy $d_h^{\pi^\dagger}(s) > 0$, with high probability for all sufficiently large $T$.

**Theorem 4.2.** *Let* $(\mathcal{S}, \mathcal{A}, H, \phi, \{\theta_h\}_{h=1}^H)$ *be a finite-horizon linear MDP and* $\pi^\dagger$ *a target policy. Consider the convex quadratic program (CQP)*

$$
\begin{aligned}
\epsilon^* = \max_{\epsilon, \{\theta_h^\dagger\}_{h=1}^H} \quad & \epsilon \\
s.t. \quad & Q_{\{\theta_h^\dagger\}}(s, \pi^\dagger(s)) \geq \epsilon + Q_{\{\theta_h^\dagger\}}(s, a), \\
& \quad \forall h, \ \forall s, \ \forall a \neq \pi^\dagger(s) \text{ with } d_h^{\pi^\dagger}(s) > 0, \\
& \langle \phi(s, \pi^\dagger(s)), \theta_h^\dagger \rangle = \langle \phi(s, \pi^\dagger(s)), \theta_h \rangle, \\
& \quad \forall h, \ \forall s, \text{ with } d_h^{\pi^\dagger}(s) > 0, \\
& \|\theta_h^\dagger\|_2 \leq \sqrt{d}, \forall h.
\end{aligned}
$$

$$(1)$$

*Then the MDP is attackable w.r.t.* $\pi^\dagger$ *if and only if* $\epsilon^* > 0$.

**Gap $\epsilon$'s Intuition** The $\epsilon$ term in our CQP (also called a "gap") captures the reward separation between action $a^\dagger = \pi^\dagger(s)$ and any other competing action $a \neq \pi^\dagger(s)$ under the relevant support. When $\epsilon < 0$, the adversary is unable to make its actions appear strictly optimal over all other actions. If $\epsilon > 0$, the adversary can make its target policy $\pi^\dagger$ dominate with a positive gap over all other prospective actions. For the case $\epsilon = 0$, no reward separation makes $\pi^\dagger$ optimal under the relevant support. Our CQP characterizes this setting as *intrinsically robust* because $\pi^\dagger$ cannot be differentiated from other competing policies. This breaks the learner's no-regret property, which the adversary exploits to force the learner to select $\pi^\dagger$.

**Intuition of CQP Constraints** (1) The $Q$-margin constraint ensures a uniform positive gap $\epsilon$ that makes $\pi^\dagger(s)$ strictly better than any other $a \neq \pi^\dagger(s)$ at every visitable state. (2) The equality on $\langle \phi(s, \pi^\dagger(s)), \theta_h^\dagger \rangle$ means the attacker pays zero cost whenever the learner takes the adversary's target action. (3) The norm bound keeps per-stage perturbations bounded, matching the regularity of the linear-MDP class. Therefore, the variables $\{\theta_h^\dagger\}$ ask *whether one can reshape rewards* so that $\pi^\dagger$ attains a uniform positive $Q$-margin on its occupancy support while leaving the target branch unchanged.

**Intrinsic Robustness** An important consequence of this characterization is the existence of intrinsically robust linear MDPs under our adopted pointwise criterion. In particular, attackability is not only a property of the learner but also of the chosen environment geometry: some linear-MDP instances/representations do not admit sublinear-cost attacks within the shaping class captured by our CQP. Later, we demonstrate this phenomenon empirically using linear representations of the same underlying environment to investigate whether our theoretical predictions hold. While the characterization correctly predicts the intrinsic robustness/vulnerability of our tested linear MDP representations, we clarify that no guarantee might hold for the original underlying (often non-linear) environment.

**Continuous Spaces** Notably, this characterization carries over to continuous state-action spaces by replacing the discrete occupancy measure $d_h^\pi(s)$ with an occupancy density $f_h^\pi(s, a)$, under which the positivity conditions (e.g., $d_h^\pi(s) > 0$) become requirements that $f_h^\pi(s, a)$ is positive almost everywhere in the support of the policy. The linear constraints in the finite-horizon setting then generalize to integral constraints over continuous spaces, but the core attack-gap analysis remains fundamentally unchanged.

*Proof Sketch of Theorem 4.2.* **Sufficiency ($\epsilon^* > 0$ implies vulnerability).** If the convex program in Equation (1) has a strictly positive solution $\epsilon^*$, the attacker can create a persis-

tent Q function value gap favoring $\pi^\dagger$. A **white-box attack strategy** is designed based on the solution of the convex program (1), $\theta^\dagger$: The attacker will perturb the reward as $\hat{r}(s, a) = \langle \phi(s, a), \theta^\dagger \rangle$ whenever the action chosen deviates from $\pi^\dagger(s)$ or the state $s$ won't be visited by $\pi^\dagger$, so that $\pi^\dagger$ appears strictly better by at least $\epsilon^*$ with any no-regret algorithm. When the learner sticks to $\pi^\dagger$, the reward gets no adjustment without any cost, but the reward can still be regarded as $\langle \phi(s, a), \theta^\dagger \rangle$, due to the equality constraint in Equation (1). Any no-regret learning algorithm will deviate from the best policy by $o(T)$ times. Hence, the total attack cost remains $o(T)$, proving the MDP attackable.

**Necessity ($\epsilon^* \leq 0$ implies intrinsic robustness).** We first establish impossibility in the strict-negative regime $\epsilon^* < 0$. In this case, we need to identify at least one no-regret learner—vanilla LSVI-UCB—for which no sublinear-cost reward-poisoning strategy can force $\pi^\dagger$ to be followed on $T - o(T)$ episodes. The contradiction is that, if such an attack existed, then the learner's estimates on the repeatedly visited target support would allow us to construct a feasible point of equation 1 with nonnegative gap, contradicting $\epsilon^* < 0$.

Under Definition 4.1, a successful attack must still force strict pointwise optimality of $\pi^\dagger$ on the relevant occupancy support for $T - o(T)$ episodes. Within the white-box shaping class of Equation (1), such forcing would again yield a feasible point with $\epsilon > 0$, contradicting $\epsilon^* = 0$. Hence the boundary case is non-attackable as well. $\qquad\square$

*Remark* 4.3 (Extension to Target Policy Sets). The CQP formulation in equation 1 admits a natural generalization from a single target policy $\pi^\dagger$ to a *target policy set* $\Pi^\dagger$. This extension accommodates policies that are "very close" to the original target, allowing the adversary to enforce slight modifications if the exact target is strictly unattackable. Crucially, the set $\Pi^\dagger$ is constructed as the Cartesian product of permissible action sets $\mathcal{A}^\dagger(s)$ at each state (i.e., $\Pi^\dagger = \bigotimes_s \mathcal{A}^\dagger(s)$), restricted to policies that share the *same occupancy support* $\{s : d_h^\pi(s) > 0\}$. The modified CQP equation 2 identifies the specific policy $\pi^\dagger \in \Pi^\dagger$ that maximizes the attack margin $\epsilon^*$:

$$\epsilon^* = \max_{\epsilon, \{\theta_h^\dagger\}_{h=1}^H, \pi^\dagger \in \Pi^\dagger} \epsilon$$
$$\text{s.t.} \quad Q_{\{\theta_h^\dagger\}}(s, \pi^\dagger(s)) \geq \epsilon + Q_{\{\theta_h^\dagger\}}(s, a),$$
$$\forall h, \forall s, \forall a \notin \mathcal{A}^\dagger(s), \text{ with } d_h^{\pi^\dagger}(s) > 0,$$
$$\langle \phi(s, \pi^\dagger(s)), \theta_h^\dagger \rangle = \langle \phi(s, \pi^\dagger(s)), \theta_h \rangle,$$
$$\forall h, \forall s, \text{ with } d_h^{\pi^\dagger}(s) > 0,$$
$$\|\theta_h^\dagger\|_2 \leq \sqrt{d}, \forall h.$$

$$(2)$$

The assumption of shared occupancy support is necessary; if candidate policies imply distinct occupancy measures, they cannot be jointly optimized within a single CQP due to the learner's pursuit of expected return maximization. Consider a counter-example with two policies, $\pi^1$ and $\pi^2$. Suppose solving the CQP for $\pi^1$ yields a feasible attack $\theta_1$ with $\epsilon^* > 0$, while $\pi^2$ is intrinsically robust ($\epsilon^* \leq 0$) due to constraints at some state $s'$ (where $d^{\pi^1}(s') = 0$). If, at the initial state $s_0$, the attacked Q-values satisfy $Q_{\theta_1}(s_0, \pi^2(s_0)) > Q_{\theta_1}(s_0, \pi^1(s_0))$, an algorithm like LSVI-UCB will select $\pi^2(s_0)$. If this action leads to that state $s'$ where $d^{\pi^1}(s') = 0$, the learner deviates from the support of $\pi^1$ entirely, causing the attack to fail despite the theoretical feasibility of $\pi^1$. Therefore, for policies with distinct occupancy measures, one must enumerate candidate policies and solve the CQP equation 1 for each individually.

*Remark* 4.4 (Relation to previous results). For $H = 1$ with reward poisoning, the attackability gap $\epsilon^*$ in Theorem 4.2 recovers the linear bandit conditions established in (Wang et al., 2022), allowing $o(T)$-cost attacks when $\epsilon^* > 0$. Under one-hot feature vectors $\phi(s, a) = \mathbf{1}_{(s,a)}$, our framework generalizes the attack feasibility results of Zhang et al. (2020). Instead of bounding the scale of poisoned reward as the third condition shows in Equation (1), Zhang et al. (2020) considers the feasibility of bounded attack in tabular MDP with the constraint on the reward perturbation as follows,

$$|r(s, a) - r_{\theta_h^\dagger}(s, a)| \leq \Delta, \quad \forall h, s, a : d_h^{\pi^\dagger}(s) > 0.$$

Our proof still holds if our third condition is replaced by this constraint without loss of generality. Specifically, their Theorem 4 can be directly recovered by our gap $\epsilon^* = \frac{2}{1+\gamma}(\Delta - \Delta_3)$, where $\Delta_3$ is the critical threshold for reward perturbation. We put the proof of this part in Appendix E.1. Rangi et al. (2022b) also constructed an infeasible target in tabular MDP against rewarding poisoning: their counterexample (Appendix A.1 of (Rangi et al., 2022b)) is also covered by our Theorem 4.2 as a special case with $\epsilon^* = -0.1$ when $H = 2$.

*Remark* 4.5 (Technical challenge). Note that the main difference between linear bandit and linear MDP is that linear MDP observes states through trajectory rollouts, leading to the dependencies among the feasible states and actions at the current step and historical trajectories. Consequently, it is necessary to investigate state-action pairs under the occupancy measure and analyze the confidence bounds on such pairs. Furthermore, instead of analyzing confidence bounds on rewards in linear bandits, we derive/analyze confidence bounds for the Q value functions. We do this because they are more directly related to the policy, and we examine the relationship between Q value functions and reward functions to bridge the gap in analysis.

**Efficient computation of equation 1** When $|\mathcal{S}|, |\mathcal{A}| < \infty$, the program can be solved efficiently by expressing $Q$ as an affine function of $\{\theta_h^\dagger\}$. Let $\mathbf{P}^{h,\dagger}$ be the transition matrix under $\pi^\dagger$, where $\mathbf{P}_{ij}^{h,\dagger} = \Pr_h\{s_{h+1} = j \mid s_h = i, \pi^\dagger(i)\}$. Stack $Q$-values at stage $h$ into a row vector $\mathbf{Q}^{h,\dagger}$ and collect $\{\phi(s, \pi^\dagger(s))\}_s$ as columns of a matrix $\Phi$. Then

$$\begin{aligned}
\mathbf{Q}^{h,\dagger} &= (\theta_h^\dagger)^\top \Phi + \mathbf{Q}^{h+1,\dagger} \mathbf{P}^{h,\dagger} \\
&= (\theta_h^\dagger)^\top \Phi + (\theta_{h+1}^\dagger)^\top \Phi \mathbf{P}^{h,\dagger} + \mathbf{Q}^{h+2,\dagger} \mathbf{P}^{h+1,\dagger} \mathbf{P}^{h,\dagger} \\
&= \cdots
\end{aligned}$$

Thus each constraint in equation 1 is linear or convex quadratic in the decision variables, enabling standard solvers to compute $\epsilon^*$.

## 5. Black-box Attack

In the black-box setting, the attacker does not know the environment parameters. In each episode $t$, after observing $(s_h^t, a_h^t, r_h(s_h^t, a_h^t), s_{h+1}^t)$, the attacker may replace the learner's observed reward by $\hat{r}_h^t \in [0, 1]$. The goal is to force the learner to follow a deterministic target policy $\pi^\dagger$ in $T - o(T)$ episodes while paying only $o(T)$ total shaping cost. Unlike the white-box characterization, the black-box result below is a conditional certified-support forcing theorem under explicit estimation and certification.

Our construction has two stages. In Stage 1, we manipulate the target branch so that, on the certified support, the learner is driven to view the target policy as best. Once Stage 1 has produced a large clean target-policy history, we switch to a support-restricted penalty design and check whether the manipulated target policy can be kept from becoming suboptimal.

Let $\phi_h^\dagger(s) := \phi_h(s, \pi^\dagger(s))$. The attack is restricted to a certified support $\{\mathcal{C}_h\}_{h=1}^H$. We assume that the initial-state distribution is supported on $\mathcal{C}_1$, and that the certified support is closed under all certified actions. The learner is assumed to be a UCRL-type optimistic model-based learner (Ayoub et al., 2020).

**Stage 1: making the target policy look best.** Stage 1 begins by solving the relaxed geometric program

$$\begin{aligned}
\epsilon_{1,h}^\star &:= \max_{\|w\|_2 \leq \frac{1}{2S}} \min_{s \in \mathcal{C}_h} \left( \langle \phi_h^\dagger(s), w \rangle - \max_{a \neq \pi^\dagger(s)} \langle \phi_h(s, a), w \rangle \right), \\
\epsilon_1^\star &:= \min_{h \in [H]} \epsilon_{1,h}^\star.
\end{aligned} \tag{3}$$

If $\epsilon_1^\star \leq 0$, we return NON-ATTACKABLE. Otherwise, let $w_h$ be an optimizer and define $q_h := Sw_h$, $q_{H+1} := 0$.

The relaxed program is purely geometric: it only identifies a direction that favors the target action over the certified comparison actions. To turn this direction into an actual steering

reward, define the surrogate value $v_h(s) := \langle \phi_h^\dagger(s), q_h \rangle$ and the associated one-step continuation term $\mu_h(s, a) := \mathbb{E}[v_{h+1}(s_{h+1}) \mid s_h = s, a_h = a]$. Under the linear MDP model, this continuation term is linear in $\phi_h(s, a)$, so it can be estimated by ridge regression using the observed targets $v_{h+1}(s_{h+1}^t)$. This yields a provisional Stage 1 steering reward

$$\widetilde{r}_{h,t}^{(1)}(s, a) = \phi_h(s, a)^\top (q_h - \widehat{b}_{h,t}), \tag{4}$$

where $\widehat{b}_{h,t}$ is the current estimate of the continuation coefficient.

Then the attacker sets

$$\widehat{\theta}_h := q_h - \widehat{b}_h^{\text{frz}}, \qquad \widetilde{r}_h^{(1)}(s, a) := \phi_h(s, a)^\top \widehat{\theta}_h, \tag{5}$$

where $\widehat{b}_h^{\text{frz}}$ denotes the continuation-term estimate used at freezing. From that point on, Stage 1 is a fixed attacked MDP.

**Stage 2: checking and stabilizing the manipulated target policy.** Using the clean Stage 1 history, we estimate the target-policy $Q$-function. Because $Q^{\pi^\dagger}$ is linear in $\phi_h(s, a)$, clean Monte Carlo returns along $\pi^\dagger$ give a ridge estimate and a high-probability confidence interval. This yields a conservative upper bound $\widehat{g}_h(s, a)$ on how much worse a certified comparison action must be made relative to the target action.

Stage 2 then solves the support-restricted penalty design

$$\epsilon_2^\star := \max_{\epsilon, \{u_h\}_{h=1}^H} \epsilon \tag{6}$$

$$\text{s.t.} \quad \langle \phi_h(s, a), u_h \rangle \geq \widehat{g}_h(s, a) + \epsilon, \quad \forall s \in \mathcal{C}_h, \ \forall a \neq \pi^\dagger(s), \tag{7}$$

$$\langle \phi_h^\dagger(s), u_h \rangle = 0, \quad \forall s \in \mathcal{C}_h, \tag{8}$$

$$\|u_h\|_2 \leq \sqrt{d}, \quad \forall h \in [H]. \tag{9}$$

If $\epsilon_2^\star \leq 0$, we return NON-ATTACKABLE. Otherwise, the resulting penalty leaves the target action unchanged and uniformly suppresses certified non-target actions on the certified support.

During Stage 1, target visits use the steering reward rather than the clean reward, so the learner enters Stage 2 with a finite target-side discrepancy. We incorporate this through an integrated compensation schedule. Concretely, on target visits the attacker adds a predictable correction $c_h^t$, while on non-target visits it applies the fixed Stage 2 penalty. We assume an admissible predictable schedule can be chosen with total mass at most $HT_1$, so it remains sublinear whenever $T_1 = o(T)$. Thus Stage 2 starts immediately at episode $T_1 + 1$: its non-target part is stationary, and its target-side correction is a finite-mass predictable adjustment.

At a high level, the methodology is therefore simple. Stage 1 makes the learner discover the target policy as best on the

---

**Algorithm 1** Black-box Two-Stage Attack

1: **Inputs:** horizon $H$, total episodes $T$, Stage 1 length $T_1$, budget $S$, features $\phi$, target policy $\pi^\dagger$, certified sets $\{\mathcal{C}_h\}$.
2: Solve the relaxed Stage 1 CQP equation 3. If $\epsilon_1^\star \leq 0$, **return** NON-ATTACKABLE.
3: Form the Stage 1 geometric direction $q_h = Sw_h$ and the surrogate values $v_h(s) = \langle \phi_h^\dagger(s), q_h \rangle$.
4: **for** $t = 1, \ldots, T_1$ **do**
5:     Estimate the Stage 1 continuation term and update the provisional steering reward $\widetilde{r}_{h,t}^{(1)}$.
6:     **if** Stage 1 certifies **then**
7:         Set the Stage 1 reward to $\widetilde{r}_h^{(1)}$.
8:     **end if**
9: **end for**
10: **if** Stage 1 does not certify by the end of episode $T_1$ **then**
11:     **return** NON-ATTACKABLE.
12: **end if**
13: Extract clean target episodes from the Stage 1 history.
14: Estimate $Q^{\pi^\dagger}$, construct $\widehat{g}_h(s, a)$, and solve the Stage 2 penalty design equation 6–equation 9. If $\epsilon_2^\star \leq 0$, **return** NON-ATTACKABLE.
15: Compute the finite Stage 1 target-side discrepancy and choose an admissible predictable compensation schedule.
16: **for** $t = T_1 + 1, \ldots, T$ **do**
17:     On target visits, feed the clean reward plus compensation; on non-target visits, feed the fixed Stage 2 penalty reward.
18: **end for**

---

certified support. Stage 2 checks whether this manipulated target policy can be kept from becoming suboptimal after we switch to a stationary support-restricted penalty. If the Stage 2 program succeeds, then under a UCRL-type learner certified non-target actions can still be selected only sublinearly often, and the total shaping cost remains sublinear.

Next, the black-box result is stated under a collection of standard support, excitation, and confidence conditions, including bounded features, admissible attacked rewards, initial support on $\mathcal{C}_1$, support closure on the certified support, sufficient excitation, and the high-probability confidence event for the Stage 1 continuation-term regression. We refer to Appendix C for the precise formulation.

**Theorem 5.1** (Black-box Cost)**.** *Under the standing black-box conditions above, suppose that Stage 1 succeeds over the first $T_1$ episodes in the sense that it certifies a fixed attacked MDP on the certified support where $\pi^\dagger$ is best with gap $\Theta(1/S)$, and leaves a clean target-policy history of length $T_1 - \widetilde{O}(\sqrt{T_1})$. Suppose also that the Stage 2 penalty design returns $\epsilon_2^\star = \Theta(1/S)$, and that the target-side compensation schedule is admissible with total mass at most $HT_1$.*

*Then, with probability at least $1 - 4\delta$,*

$$\text{Cost}(T, T_1) \leq \widetilde{O}\left(HT_1 + d^{3/2}H^{5/2}S\left(\sqrt{TT_1} + \frac{T}{\sqrt{T_1}}\right)\right). \tag{10}$$

*In particular, choosing $T_1 = \Theta(\sqrt{T})$ gives*

$$\text{Cost}(T, \Theta(\sqrt{T})) \leq \widetilde{O}(d^{3/2} H^{5/2} S\, T^{3/4}), \quad (11)$$

*and the learner follows $\pi^\dagger$ in $T - \widetilde{O}(T^{3/4})$ episodes on the certified support.*

*Remark* 5.2. Both CQP(1) and the Stage 2 penalty design remain sufficient certificates on the chosen certified support. Thus, if Stage 1 does not certify within the first $T_1$ episodes, or if the Stage 2 program returns $\epsilon_2^\star \leq 0$, we can only conclude that our current two-stage construction is not certified on this support with the available data. This does not rule out attackability under a different support choice, a richer shaping class, or a non-stationary design.

**Proof sketch.** Stage 1 starts from the geometric direction given by the relaxed CQP and estimates only the one-step continuation term of that direction by value-targeted regression. Rather than learning the full transition model, we estimate only the value-relevant quantity needed to certify that the target policy has become best on the certified support. Once this certification occurs within the first $T_1$ episodes, Stage 1 yields a fixed attacked MDP together with a clean target-policy history of length $T_1 - \widetilde{O}(\sqrt{T_1})$.

From that point on, the learner interacts with a fixed attacked MDP in which the target policy is best on the certified support. The UCRL regret guarantee therefore implies that certified non-target actions can be selected only sublinearly often, which yields the clean target-policy history needed for Stage 2.

Stage 2 then uses this clean history to estimate the target-policy $Q$-function, upper bound the clean comparison gap, and solve the support-restricted penalty design. If this design succeeds, then the manipulated target policy does not become suboptimal after the switch to Stage 2. The appendix combines this certified margin with the assumed warm-start regret interface to show that certified non-target actions remain sublinear, and the total shaping cost is therefore sublinear as well.

The main technical work, deferred to the appendix, is to quantify the Stage 1 certification condition and the Stage 2 warm-start regret tradeoff. These yield the bound $\sqrt{T\,T_1} + T/\sqrt{T_1}$, and choosing $T_1 = \Theta(\sqrt{T})$ gives the final $\widetilde{O}(T^{3/4})$ cost.

## 6. Empirical Validation

To verify our theory, we present a set of experiments in which we implement our white-box, black-box, and baseline (Xu & Singh, 2025) attack algorithms. We begin this section by describing three claims/predictions that our theoretical results make. Then, we introduce our empirical setup, shortly explaining how we implement our theory and

experiments. Finally, we describe our results through plots and analyze the validity of our claims. We leave the rest of the details in this section to the appendix D.

### 6.1. Empirical Objectives

We design our empirical demonstrations with the aim of testing/observing the predicted behavior of both the adversary and victim algorithms under environments characterized as "robust" or "vulnerable". Our theory predicts the following behaviors:

(1) A "*vulnerable*" environment's intrinsic geometry allows attackers to only need **sub-linear** cumulative perturbations to succeed, while "*robust*" environments force **super-linear** cumulative perturbations.
(2) Adversaries acting in a "vulnerable" environment will cause the victim's policy to converge towards the adversarial target, while adversaries in a "robust" environment will not.
(3) Our characterization's CQP program predicts the *success/failure* of carrying out targeted reward poisoning attacks.

An important behavior central to our discussion is **sub/super-linearity**. In the context of cumulative perturbations, sub-linearity refers to the asymptotic "tapering" effect of the growth of a function, often visualized as a "flattening" of the curve over time. This behavior results from the reduction of perturbations by the adversary over time, indicating the victim agent's actions align with the adversary's. On the other hand, super-linearity refers to an apparent "acceleration" over time, often visualized as a "curving-up" of the curve over time. This often indicates a continual disagreement between the victim agent's and adversary's actions.

### 6.2. Experimental Setup

Linear MDPs, as described in our assumptions, are often hard to construct because most realistic problem settings are non-linear. We address this challenge by using **ConT**rastive **R**epresentation **L**earning (CTRL) (Zhang et al., 2022).

**Linear MDP Representation** We use CTRL, a soft actor-critic (SAC) (Haarnoja et al., 2018) algorithm, to learn a linear MDP representation from the MuJoCo benchmark environments. CTRL uses noise-contrastive estimation (NCE) to learn $\pi^*$, $\phi(\cdot, \cdot)$, $\mu(\cdot)$, and $\theta/\omega$ through interaction with the environments. Despite the restrictions imposed by linear MDPs, CTRL achieves state-of-the-art performance (Zhang et al., 2022) on comparable benchmark environments despite using the linearity assumption.

**Black-box vs White-box Attacks.** The core difference between our white-box and black-box (see algorithm 1) at-

tacks lies in the type of environment access given. In the white-box case, we allow knowledge of $\phi$, $\mu$, and $\theta/\omega$. For black-box, the adversary must estimate these functions and variables on their own. Both attack algorithms solve our CQP program to characterize the environment and obtain an attack strategy. We tackle the *theoretical-practical* implementation gap by solving for each variable needed in our CQP through a combination of optimization and approximation, and then using the popular library CVXPY to solve the CQP. We leave implementation details in appendix D.

**Baseline Attack.** The AT attack (Xu & Singh, 2025) strategy computes a series of perturbations $\Delta^t$ for all timesteps $t$ such that the attack is $(\epsilon, C, B)$-Efficient: $\frac{1}{T}\mathbb{E}\left[\sum_{t=1}^{T} d(a^t, \pi^\dagger(s^t))\right] = \epsilon$, $\mathbb{E}\left[\sum_{t=1}^{T}|\Delta^t|\right] = C$, and $\max_t |\Delta^t| = B$. That is, an attack that uses at most $C$ perturbation budget, where the single largest individual perturbation is at most $B$, and where the average action distance between the victim and adversarial policy is at most $\epsilon$.

**Benchmarking tasks.** We test our results on linear MDP representations of three continuous MDP problems from the MuJoCo library. We use "*Mountain-Car*", "*Half-Cheetah*", and "*Pendulum*". Mountain-Car challenges an agent to reach the top of a hill as fast as possible. Half-Cheetah tests the ability of an agent to control a 2D legged robot to run as fast as possible. Lastly, the Pendulum requires the agent to flip the pendulum upright and balance it. We include further details in appendix D.

**CQP Accuracy** Our experiments also investigate directly whether our characterization prediction is accurate. This is important in assessing whether our theoretical characterizations hold in practical scenarios. Due to the intrinsic stochasticity of CTRL, the learned representation will diverge across trials, leading to different characterizations for the same environment. While this is expected, we are interested in observing whether our theory accurately predicts the victim-adversary behavior in these environments despite CTRL's approximation variability. We collect 100 environments for each characterization and allow up to $T = 20,000$ episodes per environment. We then use *two metrics* to assess attack success/failure. The first metric (**M1**) looks at the sub/super-linearity of cumulative perturbations, while the second metric (**M2**) looks directly at the learned behavior of the victim's policies (see 6.3). We leave the rest of our experiment details in the appendix.

### 6.3. Adversary's Target Policy

Since manually defining the adversarial target policy $\pi^\dagger$ is a complex endeavor, we instead learn the adversarial target policies through environment design. We accomplish this by modifying the reward function of the environments to incentivize actions aligning with our targeted behavior:

**MountainCar** The reward is modified such that the target mountain to climb is opposite to the original target. Because of this, the adversarial policy needs to learn to swing back and forth to reach the top of the left mountain, using the right mountain for propulsion.

**HalfCheetah** We modify the environment such that the optimal policy is instead the policy that makes the robot run backwards (to the left) with high stability.

**Pendulum** Our modification to the pendulum environment incentivizes the agent to maintain a 90-degree angle to the right. In doing this, the adversarial policy is pressed to reach the right angle and output enough force to keep the pendulum stationary at 90 degrees.

Then, we use a learning algorithm (e.g., CTRL or LSVI-UCB) to learn $\pi^\dagger$ from these "hacked" environments. We avoid any modifications to other parts of the environment, such as the state-action spaces and transition dynamics. We discard the modified environments after learning $\pi^\dagger$.

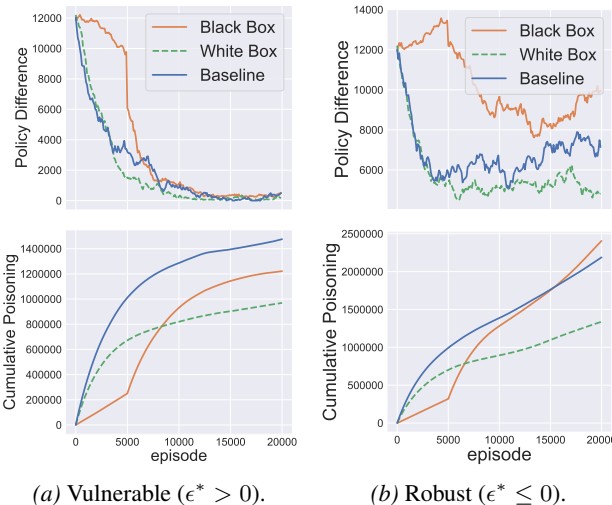

*(a)* Vulnerable ($\epsilon^* > 0$).     *(b)* Robust ($\epsilon^* \leq 0$).

*Figure 1.* Average policy difference (*top row*) and cumulative perturbations (*bottom row*) per episode for vulnerable (1a) or robust (1b) "Half-Cheetah" environments against LSVI-UCB.

*Table 1.* Percentage of environments successfully attacked according to metrics $M1$ and $M2$.

| **Environment** | Sub-lin. ($M1$) | Adv.'s Goal ($M2$) |
|---|---|---|
| Half-Cheetah | 100% | 100% |
| MountainCar | 100% | 98% |
| Pendulum | 100% | 97% |

### 6.4. Results and Analysis

Due to space limits, our plots in the main text only cover half-cheetah; we leave the rest in appendix D, including a

*Table 2.* Percentage of environments where attack failed according to metrics $M1$ and $M2$.

| Environment | Sub-lin. ($M1$) | Adv.'s Goal ($M2$) |
|---|---|---|
| Half-Cheetah | 94% | 91% |
| MountainCar | 92% | 88% |
| Pendulum | 95% | 93% |

link to our GitHub repository. In figure 1, we present our results for the policy difference and cumulative perturbations for "vulnerable" (1a) and "robust" (1b) environments. We run up to $T = 20,000$ episodes for 20 trials per characterization ("vulnerable" or "robust"). **Cumulative perturbation** refers to the total perturbations used thus far at each time step ($\sum_t \delta_t^\dagger$). **Policy difference** refers to the L2 norm between the victim and adversary policies ($||\pi^\dagger - \tilde{\pi}_t||_2$). Both policies are parameterized as neural networks, and we use their weight matrices to compute their similarity. We use figure 1 to test our first two claims.

**(claim 1) Cumulative Perturbation** When characterized as "vulnerable" ($\epsilon^* > 0$), figure 1a's cumulative perturbation plots show a downward curve as time passes. This aligns with sub-linear behavior and happens when the perturbation strength decreases over time. As actions taken by the victim agent match those of the adversarial policy, the adversary will use fewer perturbations to further convince the victim algorithm, leading to a sub-linear perturbation cost.

On the other hand, when the environment is characterized as "robust" ($\epsilon^* \leq 0$), figure 1b's cumulative perturbation plots do not show the same downward curve behavior. Rather, the plots have a more "straight" or "curve-up" look in the later episodes. This behavior results from non-decreasing perturbations, suggesting the adversary is not decreasing its perturbations over time.

**(claim 2) Policy Convergence** While perturbations capture the adversary's poisoning behavior, the policy difference ($||\pi_i - \pi^\dagger||_2$) in figures 1a and 1b show how the adversary affects the victim's policy as the attack is carried out. In "vulnerable" environments, we observe fast convergence towards zero, indicating the adversary's attack success. On the other hand, "robust" environments show a lack of convergence: a sign that the adversary is unable to succeed.

**(claim 3) CQP Accuracy** Tables 1 and 2 present our results on the accuracy of our CQP characterization. Both tables demonstrate good correlation between the CQP characterization and the behavior of perturbations and the victim's learned policy. In the "vulnerable" table 1, the high accuracy percentages suggest that the adversary eventually tapers off their attacks while achieving the intended adversarial behavior (see empirical objectives 6.1). In the converse, we also observe high accuracy percentages in table 2, indicating the adversary does not reduce perturbation strength over time and fails to achieve the intended adversarial behavior.

## 7. Conclusion

We studied the intrinsic robustness of reinforcement learning under reward poisoning in linear MDPs. Our core result is a convex optimization characterization indicating when a target policy can be enforced with sublinear poisoning costs, thereby separating vulnerable from robust instances. This distinction arises from the geometric relation between environmental features and the adversary's target policy. We further validated our theory by approximating real-world environments as linear MDPs through contrastive representation learning (Zhang et al., 2022). Empirical results show that solving our optimization on these approximations accurately predicts attack success in the original environments. Hence, even under approximate linearity, intrinsic robustness remains a strong safeguard against reward poisoning in applied RL systems.

## Acknowledgment

We thank anonymous reviewers for the constructive feedback. This work is in part supported by National Science Foundation under grant IIS-2403401 and the Samsung Strategic Alliance for Research and Technology (START) program. The findings and conclusions in this work are those of the author(s) and do not necessarily represent the views of the funding agency.

## Impact Statement

This research studied the vulnerability and intrinsic robustness of linear MDPs against reward poisoning attacks. Our findings on characterization and the attack algorithm have the potential to inspire the design of robust learning settings, where RL algorithms could be ensured to resist reward poisoning attacks as well as ward off unwanted or risky policies. We believe our work expands our understanding of adversarial attacks as a phenomenon in decision-making systems, establishing a new perspective on the trustworthiness of learning algorithms, complementing the classic algorithmic attack and defense perspectives.

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

# A. Notation

| | |
|---|---|
| $\mathcal{S}$ | State space of the MDP. |
| $\mathcal{A}$ | Action space of the MDP. |
| $H$ | Horizon (number of steps in each episode). |
| $\phi$ | Feature map for linear MDPs. |
| $P_h$ | The step-$h$ transition probability. |
| $\mu_h$ | Measure vectors. |
| $\theta_h$ | Parameter vector for the reward function at step $h$. |
| $d_h^\pi$ | Probability of visiting $(s, a)$ at step $h$ under policy $\pi$. |
| $Q_h$ | $Q$-function at step $h$. |
| $\pi^\dagger$ | Target policy that the adversary aims to enforce. |
| $\theta^\dagger$ | Parameter vector for the poisoned reward function. |
| $\epsilon^*$ | Optimal margin from our convex program measuring how much $\pi^\dagger$ can be made strictly better than other actions. |
| $T$ | Total number of episodes played by the learner. |
| $C_T$ | Total attack cost incurred by the adversary up to episode $T$. |
| $\Delta^t$ | Per-timestep reward perturbation in the Adaptive Target attack |
| $L$ | Maximum action–action distance used to normalize $\Delta_t$ |

*Table 3.* Key notation used throughout the paper.

| | |
|---|---|
| $r(s, a)$ | Randomized reward for taking action $a$ in state $s$. |
| $\bar{r}(s, a)$ | Expected mean reward without being poisoned. |
| $\hat{r}(s, a)$ | Poisoned reward, possibly modified by the adversary. |
| $Q_h^*(s, a)$ | Optimal $Q$-function at step $h$ regarding to non-poisoned reward. |
| $\hat{Q}_h(s, a)$ | Empirical $Q$-function estimation learned by an algorithm (such as LSVI-UCB) under reward poisoning attack at step $h$. |
| $\mathrm{CB}_{h,t}(s, a)$ | Confidence Bound at step $h$ and episode $t$ (measuring uncertainty in $\hat{Q}_h(s, a)$). |
| $\tilde{Q}_h(s, a)$ | Upper Confidence Bound (UCB) on $\hat{Q}_h(s, a)$. |
| $w_h^*$ | True parameter vectors at step $h$ of $Q_h^*$ under linear $Q$-function parameterization. |
| $\hat{w}_h$ | Estimated parameter vectors at step $h$ of $\hat{Q}_h$ under linear $Q$-function parameterization. |
| $\mathbb{A}_h$ | Set of all state-action pairs visited by the target policy $\pi^\dagger$ at step $h$. |
| $(\cdot)_{\mathbb{A}_h}^{\parallel}$ | Component of a vector that lies in the linear subspace spanned by $\{\phi_h(s, a) \mid (s, a) \in \mathbb{A}_h\}$. |
| $(\cdot)_{\mathbb{A}_h}^{\perp}$ | Component of a vector that is orthogonal to the subspace spanned by $\{\phi_h(s, a) \mid (s, a) \in \mathbb{A}_h\}$. |
| $\mathbb{E}_{s'}[\cdot]$ | The abbreviated form of $\mathbb{E}_{s'}[\cdot \mid s, a]$, expectation concerning the random next state $s'$ given the current state and action pair $(s, a)$. |
| $S$ | Number of states in the MDP, i.e., $\|\mathcal{S}\|$. |
| $d$ | Dimension of feature map $\phi$. |
| $\lambda_t$ | Regularization parameters in algorithm LSVI-UCB. |
| $\beta_t$ | Confidence-width sequence in the UCB bound |
| $\sigma_{h,t}$ | Self-normalised norms of $\phi_t^h$ used to bound cumulative variance. |

*Table 4.* Notation used in the appendix.

| $T_1$ | Stage 1 budget in episodes |
|---|---|
| $\delta$ | Failure probability in high-probability statements |
| $\lambda$ | Ridge regularization parameter |
| $\mathcal{C}_h$ | Certified support at stage $h$ |
| $\mathcal{A}_h^-(s)$ | Non-target action set at stage $h$: $\mathcal{A} \setminus \{\pi^\dagger(s)\}$ |
| $\epsilon_{1,h}^\star, \epsilon_1^\star$ | Stage 1 relaxed CQP value at stage $h$, and its minimum over stages |
| $q_h, v_h, \mu_h$ | Stage 1 geometric direction, surrogate value, and one-step continuation term |
| $\widehat{b}_{h,t}, \widehat{b}_h^{\mathrm{f}}$ | Stage 1 continuation-term estimate at episode $t$, and the estimate used at freezing |
| $\widetilde{r}_{h,t}^{(1)}, \widetilde{r}_h^{(1)}$ | Provisional and frozen Stage 1 attacked rewards |
| $\underline{\Delta}_t, \eta_1, \tau_1$ | Lower-confidence Stage 1 margin, certification threshold, and certification time |
| $\mathcal{T}_1, N_1, S_0$ | Clean Stage 1 suffix episodes, number of bad Stage 1 suffix episodes, and Stage 1 warm-start corruption mass |
| $\Sigma_h, \widehat{w}_h, \widehat{Q}_h$ | Stage 2 ridge design matrix, parameter estimate, and plug-in estimate of $Q^{\pi^\dagger}$ |
| $g_h(s,a)$ | Conservative upper bound on the clean disadvantage of action $a$ relative to the target action |
| $\epsilon_2^\star$ | Optimal value of the Stage 2 margin-certified penalty design |
| $\bar{r}_h$ | Stage 2 stationary reference reward |
| $D_h, C_{\mathrm{c}}(T_1)$ | Target-side discrepancy left by Stage 1, and total compensation mass |
| $\gamma_\dagger$ | Stage 2 target-side estimation error |
| $R_T^{(2)}(\bar{r})$ | Stage 2 pseudo-regret with respect to the reference reward $\bar{r}$ |
| $\mathcal{E}_0, \mathcal{E}_2, \mathcal{E}_w$ | High-probability events for Stage 1 confidence, Stage 2 confidence, and the warm-start regret interface |
| $N_2$ | Number of bad Stage 2 episodes |

*Table 5.* Notation for the black-box attack section.

# B. Proof of Theorem 4.2

### B.1. Sufficiency Proof

The sufficiency proof aims to show that if there exists a solution to the following optimization problem and the result $\epsilon > 0$, then there exists an attacker algorithm to perturb the reward for sublinear times and convince the no-regret algorithm to execute a given policy $\pi^\dagger$ for times linear on $T$.

The true $Q$-function is parameterized by $\{w_h\}_{h=1}^H$, such that $Q_{\theta_h}(s,a) = \langle \phi_h(s,a), w_h \rangle$. The attacker can modify $\theta_h$ to $\theta_h^\dagger$, resulting in an attacked $Q$-function $Q_{\theta_h^\dagger}(s,a) = \langle \phi_h(s,a), w_h^\dagger \rangle = \langle \phi_h(s,a), \theta_h^\dagger \rangle + \mathbb{E}_{s_{h+1}, \pi^\dagger} Q_{\theta_{h+1}^\dagger}(s_{h+1}, \pi^\dagger(s_{h+1}))$. The environment is considered attackable if an $o(T)$-cost attack can force the learner to execute $\pi^\dagger$ for $T - o(T)$ episodes.

Note that these modifications will only affect the agent on the state with $d_h^{\pi^\dagger}(s) > 0$, where $d_h^\pi(s)$ denotes the probability of visiting $s$ at step $h$ under policy $\pi$, we define the following program to measure the 'gap' $\epsilon$ by which we can make $\pi^\dagger$ strictly better than all other actions via a carefully designed reward attack while preserving the reward of $\pi^\dagger$ on states that it visits:

$$
\begin{aligned}
\epsilon^* = \max_{\epsilon, \{\theta_h^\dagger\}_{h=1}^H} \quad & \epsilon \\
\text{s.t.} \quad & Q_{\theta_h^\dagger}(s, \pi^\dagger(s)) \geq \epsilon + Q_{\theta_h^\dagger}(s,a), \quad \forall h, s, a \neq \pi^\dagger(s): d_h^{\pi^\dagger}(s) > 0, \\
& \langle \phi_h(s, \pi^\dagger(s)), \theta_h^\dagger \rangle = \langle \phi_h(s, \pi^\dagger(s)), \theta_h \rangle, \quad \forall h, s: d_h^{\pi^\dagger}(s) > 0, \\
& \|\theta_h^\dagger\|_2 \leq \sqrt{d}, \quad \forall h,
\end{aligned}
\tag{12}
$$

Here $Q_{\theta_h^\dagger}$ denotes the $Q$-function when the reward parameters are replaced by $\{\theta_h^\dagger\}$. We say the linear MDP is *attackable*

concerning $\pi^\dagger$ if $\epsilon^* > 0$.

*Proof.* Assume $\epsilon^* > 0$. We will construct a perturbation demonstrating that any no-regret learning algorithm ends up playing the target policy $\pi^\dagger$ for $T - o(T)$ episodes. At the same time, the adversary's cost remains $o(T)$, leading to the conclusion that the linear MDP is attackable.

First, by the constraints $\langle \phi_h(s, \pi^\dagger(s)), \theta_h^\dagger \rangle = \langle \phi_h(s, \pi^\dagger(s)), \theta_h \rangle$, $\forall h, s$ with $d_h^{\pi^\dagger}(s) > 0$, whenever the learner follows $\pi^\dagger$, no parameter change occurs. Thus, if the learner picks $\pi^\dagger(s)$, the adversary does not intervene and incurs zero cost.

If the learner deviates by choosing an action $a \neq \pi^\dagger(s)$, the attacker manipulates the reward so that $Q_{\theta_h^\dagger}(s, \pi^\dagger(s)) - \max_{a \neq \pi^\dagger(s)} Q_{\theta_h^\dagger}(s, a) \geq \epsilon^* > 0, \forall h, s : d_h^{\pi^\dagger}(s) > 0$. Hence, from the learner's perspective, $\pi^\dagger(s)$ is at least $\epsilon^*$ better than any alternative action $a \neq \pi^\dagger(s)$.

In short, the adversary only pays a cost if and when the learner attempts another action $a \neq \pi^\dagger(s)$. Let $N_{\text{dev}}$ denotes the total number of such deviations across all episodes. By the boundedness condition $\|\theta_h^\dagger\|_2 \leq \sqrt{d}$ (or $\left| r_h^\dagger(s, a) - r_h(s, a) \right| \leq \Delta$ in the remark), each intervention cost is uniformly bounded, implying the adversary's total manipulation cost is $O(N_{\text{dev}})$. Since a no-regret learner facing a strictly suboptimal gap $\epsilon^* > 0$, we get $N_{\text{dev}} \leq \frac{R_t}{\epsilon^*} = o(T)$, where $R_t$ is the regret of at episode $t$. Therefore, the total adversarial cost is also sublinear in $T$.

Putting everything together, the agent sees $\pi^\dagger(s)$ as strictly optimal and only explores suboptimal actions $o(T)$ times, leading to $o(T)$ total interventions. In the remaining $T - o(T)$ episodes, $\pi^\dagger$ is selected with no additional cost. Thus, if $\epsilon^* > 0$, we conclude that the MDP is *attackable* under sublinear cost. $\qquad\square$

## B.2. Necessity Proof

---
**Algorithm 2** LSVI-UCB
---
1: **Input:** Episode length $H$, number of states $S$, feature map dimension $d$, total number of episodes $T$, and hyperparameter $\delta$.
2: **for** episode $t = 1, 2, 3 \ldots$ **do**
3:     Receive the initial state $x_1^t.s$
4:     **for** step $h = H, \ldots, 1$ **do**
5:         $\Lambda_{h,t} \leftarrow \sum_{\tau=1}^{t-1} \phi(s_h^\tau, a_h^\tau) \phi(s_h^\tau, a_h^\tau)^\top + \lambda_t \cdot \boldsymbol{I}$ where $\lambda_t = 4HS\sqrt{dt}$.
6:         $\boldsymbol{w}_{h,t} \leftarrow \Lambda_{h,t}^{-1} \sum_{\tau=1}^{t-1} \phi(s_h^\tau, a_h^\tau)[r_h(s_h^\tau, a_h^\tau) + \max_a \widetilde{Q}_{h+1,t}(s_{h+1}^\tau, a)]$.
7:         $\widetilde{Q}_{h,t}(\cdot, \cdot) \leftarrow \min\{\boldsymbol{w}_{h,t}^\top \phi(\cdot, \cdot) + \beta_t[\phi(\cdot, \cdot)^\top \Lambda_{h,t}^{-1} \phi(\cdot, \cdot)]^{1/2}, H\}$, where $\beta_t = dH(\sqrt{\log \frac{\det(\Lambda_h^t)}{\det(\lambda_t I_d)}} + \sqrt{2\log \frac{1}{\delta}}) + \frac{\sqrt{\lambda_t}}{2S}$.
8:     **end for**
9:     **for** step $h = 1, \ldots, H$ **do**
10:        Take action $a_h^t \leftarrow \arg\max_{a \in \mathcal{A}} \widetilde{Q}_h(s_h^t, a)$ and observe $s_{h+1}^t, r_h(s_h^t, a_h^t)$.
11:     **end for**
12: **end for**
---

Define the confidence bound for step $h$ at episode $t$ as

$$\text{CB}_{h,t}(s, a) := \beta_t[\phi(s, a)^\top \Lambda_{h,t}^{-1} \phi(s, a)]^{1/2}.$$

**Lemma B.1.** *Suppose the LSVI-UCB (Algorithm 2) visits $(s, a)$ at step $h$ for $n$ times till episode $t$. Then its confidence bound satisfies*

$$CB_{h,t}(s, a) \leq \frac{\beta_t}{\sqrt{n}}.$$

*Proof.* By definition, we have $\Lambda_{h,t}$ at episode $t$ as

$$\Lambda_{h,t} := \sum_{\tau=1}^{t-1} \phi(s_h^\tau, a_h^\tau) \phi(s_h^\tau, a_h^\tau)^\top + \lambda_t \cdot \boldsymbol{I}$$
$$\succeq n \cdot \phi(s, a)\phi(s, a)^\top.$$

Then, we have

$$\mathrm{CB}_{h,t}(s,a) := \beta_t[\phi(s,a)^\top \mathbf{\Lambda}_{h,t}^{-1}\phi(s,a)]^{1/2} \leq \frac{\beta_t}{\sqrt{n}}.$$

$\square$

**Lemma B.2.** *For a fixed step $h$. Suppose that each of the state-actions pairs in $\mathbb{A}_h = \{(s,a)\}$ is visited by LSVI-UCB (Algorithm 2) at step $h$ for at least $n$ times till episode $t$. Then for any state-action pair $(s,a)$ such that $\phi_h(s,a)$ is spanned by $\{\phi_h(s',a')|(s',a') \in \mathbb{A}_h\}$, the confidence bound satisfies*

$$CB_{h,t}(s,a) \leq \beta_t \cdot \frac{b}{\sqrt{n}},$$

*where $b$ is a constant depending on $(s,a)$, and the states-actions pairs in $\mathbb{A}_h$.*

*Proof.* Because $\phi_h(s,a)$ can be spanned by $\{\phi_h(s',a')|(s',a') \in \mathbb{A}_h\}$, we have

$$\phi_h(s,a) = \sum_{(s',a') \in \mathbb{A}_h} \lambda_{s',a'}\phi_h(s',a').$$

By the definition of CB, we have

$$\mathrm{CB}_{h,t}(s,a) := \beta_t \left[ \left( \sum_{(s',a') \in \mathbb{A}_h} \lambda_{s',a'}\phi_h(s',a') \right)^\top \mathbf{\Lambda}_{h,t}^{-1} \left( \sum_{(s',a') \in \mathbb{A}_h} \lambda_{s',a'}\phi_h(s',a') \right) \right]^{1/2}.$$

Note that for vectors $\mathbf{v}_1, \mathbf{v}_2, \ldots, vv_n$ and positive-definite matrix $\mathbf{A}$, we have

$$(\mathbf{v}_1 + \mathbf{v}_2 + \cdots + \mathbf{v}_n)^\top \mathbf{A}^{-1}(\mathbf{v}_1 + \mathbf{v}_2 + \cdots + \mathbf{v}_n) \leq n\sum_{i=1}^n \mathbf{v}_i^\top \mathbf{A}^{-1}\mathbf{v}_i,$$

we have

$$\mathrm{CB}_{h,t}(s,a) \leq \beta_t|\mathbb{A}_h|^{\frac{1}{2}} \left[ \sum_{(s',a') \in \mathbb{A}_h} \lambda_{s',a'}^2 \phi_h(s',a')^\top \mathbf{\Lambda}_{h,t}^{-1}\phi_h(s',a') \right]^{\frac{1}{2}} \leq \frac{1}{\sqrt{n}}\beta_t|\mathbb{A}_h|^{\frac{1}{2}} \left[ \sum_{(s',a') \in \mathbb{A}_h} \lambda_{s',a'}^2 \right]^{\frac{1}{2}}.$$

The last inequality comes from Lemma B.1. $\square$

**Lemma B.3.** *For a fixed step $h$. Suppose that each of the state-action pairs in $\mathbb{A}_h = \{(s,a)\}$ is visited by LSVI-UCB (Algorithm 2) at step $h$ for at least $n$ times till episode $t$, and all other state-action pairs are visited by $m$ at step $h$ in total. Then for any state-action pair $(s,a)$ such that $\phi_h(s,a)$ is not spanned by $\{\phi_h(s',a')|(s',a') \in \mathbb{A}_h\}$, the confidence bound satisfies*

$$CB_{h,t}(s,a) \geq \beta_t \left( \frac{b_1}{\sqrt{m+\lambda_t}} - \frac{b_2}{\sqrt{n}}, \right)$$

*where $b_1, b_2$ are constants depending on $(s,a)$, the states-actions pairs in $\mathbb{A}$, and possibly the total number of states and actions $S, A$.*

*Proof.* We decompose $\phi_h(s,a)$ as $\phi_{\mathbb{A}_h}^{\parallel}(s,a)$ and $\phi_{\mathbb{A}_h}^{\perp}(s,a)$, where $\phi_{\mathbb{A}_h}^{\parallel}(s,a)$ is the component of $\phi_h(s,a)$ lying in the span of $\{\phi_h(s',a')|(s',a') \in \mathbb{A}_h\}$, and $\phi_{\mathbb{A}_h}^{\perp}(s,a)$ is the component of $\phi_h(s,a)$ perpendicular to the span of $\{\phi_h(s',a')|(s',a') \in \mathbb{A}\}$. Then by the reverse triangle inequality, we have

$$\|\phi_h(s,a)\|_{\Lambda_{h,t}^{-1}} \geq \|\phi_{\mathbb{A}_h}^{\perp}(s,a)\|_{\Lambda_{h,t}^{-1}} - \|\phi_{\mathbb{A}_h}^{\parallel}(s,a)\|_{\Lambda_{h,t}^{-1}}.$$

We first analyze the term $\|\phi^\perp_{\mathbb{A}_h}(s, a)\|_{\Lambda^{-1}_{h,t}}$. We have

$$\Lambda_{h,t} = \sum_{(s',a')\in\mathbb{A}_h} c_{s',a'}\phi_h(s', a')\phi_h(s', a')^\top + \sum_{(s'',a'')\notin\mathbb{A}_h} c_{s'',a''}\phi_h(s'', a'')\phi_h(s'', a'')^\top + \lambda_t\boldsymbol{I},$$

where $c_{s',a'}$ denote the number of visits of the state-action pair $(s', a')$ at step $h$ till episode $t$. Let

$$\Lambda''_{h,t} = \sum_{(s'',a'')\notin\mathbb{A}_h} c_{s'',a''}\phi_h(s'', a'')\phi_h(s'', a'')^\top + \lambda_t\boldsymbol{I}.$$

Since we have $\phi^\perp_{\mathbb{A}_h}(s, a)$ is perpendicular to the span of $\{\phi_h(s', a')|(s', a') \in \mathbb{A}_h\}$, we have

$$(\phi^\perp_{\mathbb{A}_h}(s, a))^\top \Lambda_{h,t}\phi^\perp_{\mathbb{A}_h}(s, a) = (\phi^\perp_{\mathbb{A}_h}(s, a))^\top \Lambda''_{h,t}\phi^\perp_{\mathbb{A}_h}(s, a).$$

Because we know that $\sum_{(s'',a'')\notin\mathbb{A}_h} c_{s'',a''} \leq m$, we can bound

$$(\phi^\perp_{\mathbb{A}_h}(s, a))^\top \Lambda''_{h.t}\phi^\perp_{\mathbb{A}_h}(s, a) \leq m \cdot \max_{(s'',a'')\notin\mathbb{A}_h} \langle\phi^\perp_{\mathbb{A}_h}(s, a), \phi_h(s'', a'')\rangle^2 + \lambda \cdot \|\phi^\perp_{\mathbb{A}_h}(s, a)\|^2 \leq b_3(m + \lambda),$$

where $b_3$ is a constant depending on $(s, a)$, the states-actions pairs in $\mathbb{A}$, and possibly the total number of states and actions $S, A$. Then we have

$$\|\phi^\perp_{\mathbb{A}_h}(s, a)\|_{\Lambda^{-1}_{h,t}} = \sqrt{(\phi^\perp_{\mathbb{A}_h}(s, a))^\top \Lambda^{-1}_{h,t}\phi^\perp_{\mathbb{A}_h}(s, a)} \geq \frac{\|\phi^\perp_{\mathbb{A}_h}(s, a)\|^2}{\sqrt{(\phi^\perp_{\mathbb{A}_h}(s, a))^\top \Lambda_{h,t}\phi^\perp_{\mathbb{A}_h}(s, a)}} \geq \frac{b_1}{\sqrt{m + \lambda_t}},$$

where $b_1$ is a constant depending on $(s, a)$, the states-actions pairs in $\mathbb{A}_h$, and possibly the total number of states and actions $S, A$.

Then we bound $\|\phi^\|_{\mathbb{A}_h}(s, a)\|_{\Lambda^{-1}_{h,t}}$. Applying Lemma B.2, we know that

$$\|\phi^\|_{\mathbb{A}_h}(s, a)\|_{\Lambda^{-1}_{h,t}} \leq \frac{b_2}{\sqrt{n}},$$

where $b_2$ is a constant depending on $(s, a)$, the states-actions pairs in $\mathbb{A}_h$, and possibly the total number of states and actions $S, A$, and we conclude the proof of Lemma B.3. $\qquad\square$

**Lemma B.4.** *For a fixed step $h$, define $\mathbb{A}_h := \{(s, a) \mid d^{\pi^\dagger}_h(s, a) > 0\}$, $\Delta^{\pi^\dagger}_h = \min_{(s,a)\in\mathbb{A}_h} d^{\pi^\dagger}_h(s, a)$, and assume the number of episodes $T_h$ is large enough, specifically $T_h > \frac{12}{\Delta^{\pi^\dagger}_h} \cdot \log\big(\mathrm{poly}(|\mathcal{S}|, |\mathcal{A}|, 1/\Delta^{\pi^\dagger}_h, \delta)\big)$. Suppose the non-target policies $\{\pi \neq \pi^\dagger\}$ are selected $o(T_h)$ times, the target policy $\pi^\dagger$ is selected $T_h - o(T_h)$ times, and at step $T_h$ the target policy $\pi^\dagger$ is selected. Let the total manipulation be $C_{T_h}$, which is $o(T_h)$ under a sublinear attack cost budget. Let $r_h(s, a), \bar{r}_h(s, a)$ denote the randomized reward and its mean separately, and $Q^*(s, a)$ denote the Q functions of the best policy in this reward feedback. Let $\hat{r}_h(s, a)$ denote the poisoned reward, $\hat{Q}_{h,t}(s, a)$ denote the empirical Q functions in Algorithm 2 under poisoned reward feedback, and $\tilde{Q}_{h,t}(s, a)$ denote the upper confidence bound of empirical Q functions. Let $\hat{w}$ and $w^*$ denote the vector corresponding to $\hat{Q}$ and $Q^*$ under linear approximation.*

*If for step $h + 1$, the following conditions hold:*

1. *For all $(s, a)$ with $d^{\pi^\dagger}_{h+1}(s, a) > 0$, the estimated Q-value satisfies*

$$\left|\tilde{Q}_{h+1,T_h}(s, a) - Q^*_{h+1}(s, a)\right| = \frac{o(T_h)}{T_h}.$$

2. *For all $s$ with $d^{\pi^\dagger}_{h+1}(s) > 0$, we have*

$$\arg\max_a \tilde{Q}_{h+1,T_h}(s, a) \in \{a' \mid (s, a') \in \mathbb{A}_{h+1}\}.$$

3. *For all $s$ with $d_h^{\pi^\dagger}(s) > 0$, we have*

$$\arg\max_a Q_h^*(s, a) \in \{a' \mid (s, a') \in \mathbb{A}_h\}.$$

4. *For all $(s, a)$ with $d_{h+1}^{\pi^\dagger}(s, a) > 0$, $(s, a)$ is visited at least $\Omega(T_h)$ times.*

5. *For all $(s, a)$ with $d_h^{\pi^\dagger}(s, a) > 0$, $(s, a)$ is visited at least $\Omega(T_h)$ times.*

*We define $(w_h)_{\mathbb{A}_h}^\parallel$ to be the projection of $w_h$ onto the subspace spanned by $\{\phi(s, a) \mid (s, a) \in \mathbb{A}_h\}$, and $(w_h)_{\mathbb{A}_h}^\perp x$ is the component of $w_h$ perpendicular to $\{\phi(s, a) \mid (s, a) \in \mathbb{A}_h\}$.*

*Then with probability at least $1 - O(\delta)$, the Q-value estimation error for step $h$ satisfies*

$$\left|\phi(s, a)^\top(\hat{w}_h)_{\mathbb{A}_h}^\parallel - \phi(s, a)^\top(w_h^*)_{\mathbb{A}_h}^\parallel\right| = \frac{o(T_h)}{T_h}, \quad \forall (s, a) \in \mathbb{A}_h.$$

*In addition, for all $(s, a) \in \mathbb{A}_h$,*

$$\left|\tilde{Q}_{h,T_h}(s, a) - Q_h^*(s, a)\right| = \frac{o(T_h)}{T_h}.$$

*Proof.* By definition of the LSVI-UCB algorithm, we have

$$\begin{aligned}
&\left|\phi(s, a)^\top(\hat{w}_h)_{\mathbb{A}_h}^\parallel - \phi(s, a)^\top(w_h^*)_{\mathbb{A}_h}^\parallel\right| \\
&= \left|\phi(s, a)^\top \hat{w}_h - \phi(s, a)^\top w_h^*\right| \\
&= \left|\phi(s, a)^\top \left(\boldsymbol{\Lambda}_{h,T_h}^{-1} \sum_{\tau=1}^{T_h - 1} \phi(s_h^\tau, a_h^\tau)\left[\hat{r}_h(s_h^\tau, a_h^\tau) + \max_{a'} \tilde{Q}_{h+1,T_h}(s_{h+1}^\tau, a')\right] - w_h^*\right)\right| \\
&= \left|\phi(s, a)^\top \boldsymbol{\Lambda}_{h,T_h}^{-1} \left(\sum_{\tau=1}^{T_h - 1} \phi(s_h^\tau, a_h^\tau)\left[\hat{r}_h(s_h^\tau, a_h^\tau) + \max_{a'} \tilde{Q}_{h+1,T_h}(s_{h+1}^\tau, a')\right] - \boldsymbol{\Lambda}_{h,T_h} w_h^*\right)\right| \\
&= \left|\phi(s, a)^\top \boldsymbol{\Lambda}_{h,T_h}^{-1} \left(\sum_{\tau=1}^{T_h - 1} \phi(s_h^\tau, a_h^\tau)\left[\hat{r}_h(s_h^\tau, a_h^\tau) + \max_{a'} \tilde{Q}_{h+1,T_h}(s_{h+1}^\tau, a')\right.\right.\right. \\
&\qquad\qquad\qquad\qquad\qquad\qquad \left.\left.\left. - \phi(s_h^\tau, a_h^\tau)^\top w_h^*\right] - \lambda_{T_h} w_h^*\right)\right|.
\end{aligned}$$

Recall that for a linear MDP,

$$\phi(s, a)^\top w_h^* = \bar{r}_h(s, a) + \mathbb{E}_{s'} \max_{a'} Q_{h+1}^*(s', a'),$$

so we can rewrite this difference as

$$\begin{aligned}
&\left|\phi(s, a)^\top(\hat{w}_h)_{\mathbb{A}_h}^\parallel - \phi(s, a)^\top(w_h^*)_{\mathbb{A}_h}^\parallel\right| \\
&\leq \underbrace{\left|\phi(s, a)^\top \boldsymbol{\Lambda}_{h,T_h}^{-1} \sum_{\tau=1}^{T_h - 1} \phi(s_h^\tau, a_h^\tau)\left[\hat{r}_h(s_h^\tau, a_h^\tau) - r_h(s_h^\tau, a_h^\tau)\right]\right|}_{\text{Term A}} \\
&\quad + \underbrace{\left|\phi(s, a)^\top \boldsymbol{\Lambda}_{h,T_h}^{-1} \left(\sum_{\tau=1}^{T_h - 1} \phi(s_h^\tau, a_h^\tau)\left[r_h(s_h^\tau, a_h^\tau) - \bar{r}_h(s_h^\tau, a_h^\tau)\right] - \lambda_{T_h} w_h^*\right)\right|}_{\text{Term B}} \\
&\quad + \underbrace{\left|\phi(s, a)^\top \boldsymbol{\Lambda}_{h,T_h}^{-1} \sum_{\tau=1}^{T_h - 1} \phi(s_h^\tau, a_h^\tau)\left[\max_{a'} \tilde{Q}_{h+1,T_h}(s_{h+1}^\tau, a') - \mathbb{E}_{s_{h+1}'} \max_{a'} Q_{h+1}^*(s_{h+1}', a')\right]\right|}_{\text{Term C}}.
\end{aligned}$$

**Term A.** Since $(s, a)$ is visited at least $\Omega(T_h)$ times for $(s, a) \in \mathbb{A}_h$, and $\mathbf{\Lambda}_{h,T_h} := \sum_{\tau=1}^{T_h-1} \phi(s_h^\tau, a_h^\tau) \phi(s_h^\tau, a_h^\tau)^\top + \lambda_{T_h} \mathrm{I}$, it follows that

$$\left\| \phi(s, a)^\top \mathbf{\Lambda}_{h,T_h}^{-1} \right\|_2 \leq O\left(\tfrac{1}{T_h}\right).$$

Meanwhile, the total reward manipulation, which is also the total attack cost, $C_{T_h}$, is sublinear in $T_h$, so

$$\sum_{\tau=1}^{T_h-1} \phi(s_h^\tau, a_h^\tau) \left[ \hat{r}_h(s_h^\tau, a_h^\tau) - r_h(s_h^\tau, a_h^\tau) \right] \leq \left\| \max_{s,a} \phi(s, a) \right\| \cdot C_{T_h} = o(T_h).$$

Therefore, Term A is at most $\frac{o(T_h)}{T_h}$.

**Term B.** Using a self-normalized bound for vector-valued martingales (Theorem 1 in Abbasi-Yadkori et al. (2011)) and the fact that each $(s, a) \in \mathbb{A}_h$ is visited sufficiently often, one can show with high probability that

$$\left\| \phi(s, a)^\top \mathbf{\Lambda}_{h,T_h}^{-1} \Big( \sum_{\tau=1}^{T_h-1} \phi(s_h^\tau, a_h^\tau) \left[ r_h(s_h^\tau, a_h^\tau) - \bar{r}_h(s_h^\tau, a_h^\tau) \right] - \lambda_{T_h} w_h^* \Big) \right\|_2$$
$$\leq O(\sqrt{\log(T_h/\delta)}) \left\| \mathbf{\Lambda}_{h,T_h}^{-1/2} \phi(s, a) \right\|_2 + \lambda_{T_h} \left\| w_h^* \right\|_2 \left\| \phi(s, a)^\top \mathbf{\Lambda}_{h,T_h}^{-1} \right\|_2 \qquad (13)$$

According to Lemma B.1. in Jin et al. (2020), $\left\| w_h^* \right\|_2$ is bounded by $2H\sqrt{d}$, and in this way the above formula is bounded by $\frac{o(T_h)}{T_h}$ as $\lambda_{T_h}$ is $o(T_h)$. Hence, Term B is also bounded by $o(T_h)/T_h$.

**Term C.** From our assumptions and the argument that each $(s_{h+1}^\tau, a')$ with $d_{h+1}^{\pi^\dagger}(s_{h+1}^\tau, a') > 0$ is visited sufficiently many times, we have

$$\left| \max_{a'} \widetilde{Q}_{h+1,T_h}(s_{h+1}^\tau, a') - \mathbb{E}_{s_{h+1}'} \max_{a'} Q_{h+1}^*(s_{h+1}', a') \right|$$
$$\leq \left| \max_{a'} \widetilde{Q}_{h+1,T_h}(s_{h+1}^\tau, a') - \mathbb{E}_{s_{h+1}'} \max_{a'} \widetilde{Q}_{h+1,T_h}(s_{h+1}', a') \right|$$
$$+ \left| \mathbb{E}_{s_{h+1}'} \max_{a'} \widetilde{Q}_{h+1,T_h}(s_{h+1}', a') - \mathbb{E}_{s_{h+1}'} \max_{a'} Q_{h+1}^*(s_{h+1}', a') \right|$$

For the second term, we have

$$\left| \mathbb{E}_{s_{h+1}'} \max_{a'} \widetilde{Q}_{h+1,T_h}(s_{h+1}', a') - \mathbb{E}_{s_{h+1}'} \max_{a'} Q_{h+1}^*(s_{h+1}', a') \right|$$
$$\leq \max_{(s_{h+1}', a') \in \mathbb{A}_{h+1}} \left| \widetilde{Q}_{h+1,T_h}(s_{h+1}', a') - Q_{h+1}^*(s_{h+1}', a') \right|$$
$$\leq \frac{o(T_h)}{T_h}.$$

The first inequality is based on Condition 2 and Condition 3. And the second inequality comes from Condition 1. As with Term A, multiplying by $\phi(s_h^\tau, a_h^\tau)^\top \mathbf{\Lambda}_{h,T_h}^{-1}$ and using the fact that $\|\phi(s, a)^\top \mathbf{\Lambda}_{h,T_h}^{-1}\| \leq O(1/T_h)$ implies the second term is also bounded by $o(T_h)/T_h$.

For the first term, we bound it by Lemma B.2, concentration inequality with Condition 4 and Condition 5, and that $\beta_{T_h}$ is $o(T_h)$,

$$\left| \max_{a'} \widetilde{Q}_{h+1,T_h}(s_{h+1}^\tau, a') - \mathbb{E}_{s_{h+1}'} \max_{a'} \widetilde{Q}_{h+1,T_h}(s_{h+1}', a') \right|$$
$$\leq \left| \max_{a'} \left( \hat{Q}_{h+1,T_h}(s_{h+1}^\tau, a') + \mathrm{CB}_{h+1,t}(s_{h+1}^\tau, a') \right) \right.$$
$$\left. - \min\{H, \mathbb{E}_{s_{h+1}'} \max_{a'} \left( \hat{Q}_{h+1,T_h}(s_{h+1}', a') + \mathrm{CB}_{h+1,t}(s_{h+1}', a') \right)\} \right|$$
$$\leq \left| \max_{a'} \left( \hat{Q}_{h+1,T_h}(s_{h+1}^\tau, a') + \mathrm{CB}_{h+1,t}(s_{h+1}^\tau, a') \right) - \mathbb{E}_{s_{h+1}'} \max_{a'} \left( \hat{Q}_{h+1,T_h}(s_{h+1}', a') + \mathrm{CB}_{h+1,t}(s_{h+1}', a') \right) \right|$$
$$\leq \left| \max_{a'} \left( \hat{Q}_{h+1,T_h}(s_{h+1}^\tau, a') - \mathbb{E}_{s_{h+1}'} \max_{a'} \left( \hat{Q}_{h+1,T_h}(s_{h+1}', a') \right) \right|$$
$$+ \left| \max_{a'} \mathrm{CB}_{h+1,t}(s_{h+1}^\tau, a') - \mathbb{E}_{s_{h+1}'} \max_{a'} \mathrm{CB}_{h+1,t}(s_{h+1}', a') \right|$$
$$\leq \frac{o(T_h)}{T_h}.$$

Thus, the upper bound of term C is sublinear in $T_h$.

Combining Terms A, B, and C, we conclude that for every $(s, a) \in \mathbb{A}_h$,

$$\left| \phi(s,a)^\top (\hat{w}_h)^\|_{\mathbb{A}_h} - \phi(s,a)^\top (w_h^*)^\|_{\mathbb{A}_h} \right| = \frac{o(T_h)}{T_h}.$$

As for the upper confidence bound of empirical Q function values for any $(s, a) \in \mathbb{A}_h$,

$$\left| \tilde{Q}_{h,T_h}(s,a) - Q_h^*(s,a) \right| = \left| \hat{Q}_{h,T_h}(s,a) + \mathrm{CB}_{h,t}(s,a) - Q_h^*(s,a) \right| \leq \left| \hat{Q}_{h,T_h}(s,a) - Q_h^*(s,a) \right| + \left| \mathrm{CB}_{h,t}(s,a) \right|.$$

Since $T_h > \frac{12}{\Delta_h^{\pi^\dagger}} \cdot \log\left( \mathrm{poly}(|\mathcal{S}|, |\mathcal{A}|, 1/\Delta_h^{\pi^\dagger}, \delta) \right)$, $(s, a)$ is visited at least $b_4 \cdot T_h$ times. From Lemma B.2,

$$\left| \mathrm{CB}_{h,t}(s,a) \right| \leq O\left( \frac{(T_h)^{\frac{1}{4}}}{\sqrt{b_4 \cdot T_h}} \right) = \frac{o(T_h)}{T_h}.$$

We conclude that for every $(s, a) \in \mathbb{A}_h$,

$$\begin{aligned}
\left| \tilde{Q}_{h,T_h}(s,a) - Q_h^*(s,a) \right| &\leq \left| \hat{Q}_{h,T_h}(s,a) - Q_h^*(s,a) \right| + \left| \mathrm{CB}_{h,t}(s,a) \right| \\
&= \left| \phi(s,a)^\top \hat{w}_h - \phi(s,a)^\top w_h^* \right| + \left| \mathrm{CB}_{h,t}(s,a) \right| \\
&= \frac{o(T_h)}{T_h}.
\end{aligned}$$

This completes the proof of Lemma B.4. $\qquad\square$

Now we return to the proof of our main theorem.

**Theorem B.5.** *Suppose the optimal objective $\epsilon^*$ from our convex program is negative ($\epsilon^* < 0$). Then there exists a linear MDP (with true parameters $\{w_h^*\}$) and a no-regret algorithm (LSVI-UCB) such that any successful attack with sublinear cost forcing the learner to play $\pi^\dagger$ for $T - o(T)$ episodes when $T$ is large enough must incur the objective in our convex program is non-negative ($\epsilon^* \geq 0$). Hence, the environment is not attackable when $\epsilon^* < 0$.*

*Proof.* We aim to prove that if $\epsilon^* < 0$, then no $o(T)$-cost attack can make the learner play $\pi^\dagger$ in $T - o(T)$ episodes under a linear MDP environment. We show this by demonstrating a no-regret algorithm, LSVI-UCB, that resists any such attempt when $\epsilon^* < 0$.

**Step 1.** We first show that: for large enough $T$, if the target policy $\pi^\dagger$ is selected by $T - o(T)$ times, then with probability at least $1 - \delta$, for all $h \in [H]$, every state-action pair $(s, a) \in \mathbb{A}_h$ is visited by at least $\frac{\Delta_h^{\pi^\dagger} \cdot T}{3}$ times, where $\Delta_h^{\pi^\dagger}$ is defined in Lemma B.4.

The proof is a straightforward application of the Chernoff bound. First, since the target policy is selected by $T - o(T)$ times and $T$ is large enough, we assume the target policy is selected by at least $\frac{2}{3}T$ times. For any fixed $s, a$ with $d_h^{\pi^\dagger}(s, a) > 0$, the expected visit times of $(s, a)$ at step $h$ is at least $2\Delta_h^{\pi^\dagger} T/3$ times. Thus we have, the probability that visit $(s, a)$ at step $h$ is less than $\Delta_h^{\pi^\dagger} T/3$ is bounded by

$$\exp\left( -(1/2) \cdot (1/2)^2 \cdot 2\Delta_h^{\pi^\dagger} T/3 \right) = \exp\left( -\frac{\Delta_h^{\pi^\dagger} T}{12} \right) \leq 1/poly(|\mathcal{S}|, |\mathcal{A}|, 1/\Delta_h^{\pi^\dagger}, \delta).$$

Applying the union bound on all $(s, a) \in \mathbb{A}_h$ and all $h \in [H]$ leads to the result.

**Step 2.** Next we do the proof conditioned on the event that for all $h \in [H]$, every state-action pair $(s, a) \in \mathbb{A}_h$ is visited by at least $\frac{\Delta_h^{\pi^\dagger} \cdot T}{3}$ times, which happens with high probability.

Now we pick a step $T_0$ where the victim algorithm selects the target policy $\pi^\dagger$. Since we assume that the victim algorithm selects $\pi^\dagger$ for $T - o(T)$ times and the event in Step 1 happens with high probability, there are infinitely many $T_0$ that satisfy the constraint.

Because the victim algorithm select $\pi^\dagger$, we have for all step $h \in [H]$ and all $(s,a) \in \mathbb{A}_h$,

$$\hat{Q}_h(s,a) + \mathrm{CB}_{h,T_0}(s,a) \geq \hat{Q}_h(s,a') + \mathrm{CB}_{h,T_0}(s,a'), \forall a' \neq a.$$

**Step 2.1.** Now we first show that, with probability at least $1 - o(\delta)$ (where we hide the dependency on $H$), for all $h \in [H]$ and all $(s,a)$, we have

$$\left| \phi(s,a)^\top (\hat{w}_h)^{\|}_{\mathbb{A}_h} - \phi(s,a)^\top (w_h^*)^{\|}_{\mathbb{A}_h} \right| = o(T)/T.$$

The proof is a direct application of Lemma B.4 with induction. First for the base case step $H$, since $\tilde{Q}_{H+1,T}(s,a) = Q_{H+1}^*(s,a) = 0$ for all $s,a$, condition 1 and 2 for Lemma B.4 are satisfied. Then, because the victim algorithm selects $\pi^\dagger$, condition 3 is also satisfied. Then because we condition on the event that for all $h \in [H]$, every state-action pair $(s,a) \in \mathbb{A}_h$ is visited by at least $\frac{\Delta_h^{\pi^\dagger} \cdot T}{3}$ times, condition 4 and 5 are also satisfied. Then Lemma B.4 for step $H$ holds.

Then if Lemma B.4 for step $h+1$ holds, for step $h$. Condition 1 holds because from step $h+1$, we have for all $s,a$ such that $d_{h+1}^{\pi^\dagger}(s,a) > 0$, satisfies

$$|\tilde{Q}_{h+1,T}(s,a) - Q_{h+1}^*(s,a)| = o(T)/T.$$

**Step 2.2.** Now, for all $h \in [H]$ and all $(s,a)$, we have

$$\hat{Q}_h(s,\pi^\dagger(s)) + \mathrm{CB}_{h,T_0}(s,\pi^\dagger(s)) \geq \hat{Q}_h(s,a) + \mathrm{CB}_{h,T_0}(s,a), \quad \forall a \neq \pi^\dagger(s).$$

Writing $\hat{w}_h$ as the sum of its components $(\hat{w}_h)^{\|}_{\mathbb{A}_h} + (\hat{w}_h)^{\perp}_{\mathbb{A}_h}$ and substituting the expression for the parallel component, we obtain:

$$\phi_h\big(s,\pi^\dagger(s)\big)^\top \big((w_h^*)^{\|}_{\mathbb{A}_h} + (\hat{w}_h)^{\perp}_{\mathbb{A}_h}\big) + \mathrm{CB}_{h,T}(s,\pi^\dagger(s)) \geq \phi_h(s,a)^\top \big((w_h^*)^{\|}_{\mathbb{A}_h} + (\hat{w}_h)^{\perp}_{\mathbb{A}_h}\big) + \mathrm{CB}_{h,T}(s,a).$$

Thus, this perpendicular contribution can be absorbed into the existing $o(T)/T$ terms.

Since for $s$ with $d_h^{\pi^\dagger}(s) > 0$ and $a = \pi^\dagger(s)$ we have $\phi_h(s,\pi^\dagger(s))^\top (\hat{w}_h)^{\perp}_{\mathbb{A}_h} = 0$ due to orthogonality, we rearrange to obtain:

$$\phi_h\big(s,\pi^\dagger(s)\big)^\top (w_h^*)^{\|}_{\mathbb{A}_h} - \phi_h(s,a)^\top \big((w_h^*)^{\|}_{\mathbb{A}_h} + (\hat{w}_h)^{\perp}_{\mathbb{A}_h}\big) \geq -\frac{o(T)}{T} + \mathrm{CB}_{h,T}(s,a) - \mathrm{CB}_{h,T}\big(s,\pi^\dagger(s)\big). \tag{14}$$

Because the confidence bounds $\mathrm{CB}_{h,T}(s,a)$ tend to zero as $T \to \infty$, the inequality equation 14 implies that the attack gap cannot be made positive if $\epsilon^* < 0$. When $a = \pi^\dagger(s)$, since $(s,\pi^\dagger(s)) \in \mathbb{A}_h$, we have

$$Q'_h(s,\pi^\dagger(s)) = \phi_h\big(s,\pi^\dagger(s)\big)^\top (w_h^*)^{\|}_{\mathbb{A}_h} + \phi_h\big(s,\pi^\dagger(s)\big)^\top (\hat{w}_h)^{\perp}_{\mathbb{A}_h} = \phi_h\big(s,\pi^\dagger(s)\big)^\top (w_h^*)^{\|}_{\mathbb{A}_h}.$$

Hence the inequality (14) can be rewritten as $Q'_h(s,\pi^\dagger(s)) \geq Q'_h(s,a)$, which satisfies the first condition of the optimization problem 1.

For the third condition of the optimization problem 1, to let $\|\theta\|_2 \leq \sqrt{d}$, we need

$$w_H^\top w_H \leq d,$$

$$\Big\| w_h - \sum_{s'} \mu_h(s') \langle \phi_{h+1}(s',\pi(s')), w_{h+1} \rangle \, ds' \Big\|_2^2 \leq d, \quad \forall h = 1, \dots, H-1,$$

where the second line expands to

$$w_h^\top w_h - 2 \sum_{s'} w_h^\top \mu_h(s') \, \phi_{h+1}(s',\pi(s'))^\top w_{h+1} \, ds'$$
$$+ \sum_{s_1} \sum_{s_2} w_{h+1}^\top \mu_h(s_1) \, \mu_h(s_2) \, \phi_{h+1}(s_1,\pi(s_1)) \, \phi_{h+1}(s_2,\pi(s_2))^\top w_{h+1} \, ds_1 \, ds_2 \leq d.$$

Therefore, we obtain the following simplified quadratic program:

$$w_H^\top w_H \leq d,$$
$$w_h^\top w_h \ - \ w_h^\top C\, w_{h+1} \ + \ w_{h+1}^\top D\, w_{h+1} \leq d, \quad h = 1, \ldots, H-1,$$

in which the matrices $C$ and $D$ are determined by $\pi$, $\mu_h$ and $\phi_h$, and they satisfy $\|C\|_2 \leq 2\sqrt{d}\, S$ and $\|D\|_2 \leq d\, S^2$, with $S = |\mathcal{S}|$ and the boundedness of $\mu_h$ and $\phi_h$.

To find a sufficient condition satisfying the above QP, we upper bound the LHS of the above inequality,

$$w_h^\top w_h \ - \ w_h^\top C\, w_{h+1} \ + \ w_{h+1}^\top D\, w_{h+1} \leq \|w_h\|_2^2 \ + \ 2\, S\sqrt{d}\, \|w_h\|_2\, \|w_{h+1}\|_2 \ + \ d\, S^2\, \|w_{h+1}\|_2^2$$
$$= \left(\|w_h\|_2 + S\sqrt{d}\, \|w_{h+1}\|_2\right)^2.$$

Thus, a sufficient condition for the above QP is

$$\|w_H\|_2 \leq \sqrt{d}.$$
$$\|w_h\|_2 + S\sqrt{d}\, \|w_{h+1}\|_2 \leq \sqrt{d}, \quad \forall h = 1, \ldots, H-1,$$

Take

$$\|w_h\|_2 \leq \frac{1}{2S}, \tag{15}$$

and it will satisfy the above conditions.

To ensure the condition on the norm of $w_h$ holds, by Lemma B.2 from Jin et al. (2020), we need

$$\lambda_t \geq \frac{2H\sqrt{dt}}{1/(2S)} = 4HS\sqrt{dt}. \tag{16}$$

Note that the original LSVI-UCB algorithm chooses $\lambda_t = 1$ in Jin et al. (2020). However, we show that the LSVI-UCB algorithm with the chosen $\lambda_t = 4HS\sqrt{dt}$ is still no-regret in Proposition E.3.

Finally, we need to examine whether the second condition holds.

**Lemma B.6.** *In white-box setting, given the policy $\pi$, if $\|w_h\| \leq \frac{1}{2S}, \forall h$, there exists a one-to-one mapping between $Q^\pi$ function and $r$ function under the occupancy measure of $\pi$ in the following sense: for each step $h$, if we denote the subspace spanned by*

$$\left\{ \phi_h\big(s, \pi(s)\big) \ \Big| \ d_h^\pi(s) > 0 \right\} \quad as \quad \mathbb{A}_h^\pi,$$

*then the correspondence between $w_h$ and $\theta_h$ is one-to-one only on the projected components $(w_h)_{\mathbb{A}_h^\pi}^{\|}$ and $(\theta_h)_{\mathbb{A}_h^\pi}^{\|}$ under the occupancy measure of $\pi$. The component $(\theta_h)_{\mathbb{A}_h^\pi}^{\perp}$ is not identifiable from $(w_h)_{\mathbb{A}_h^\pi}^{\|}$ under the occupancy measure of $\pi$ and can be chosen arbitrarily without changing $\langle \phi_h(s, \pi(s)), \theta_h \rangle$ for states with $d_h^\pi(s) > 0$.*

*Proof.* We use the Bellman identity under a *fixed* policy $\pi$ and work on the occupancy support at each step.

**(i) From $\theta$ to $w$ on $\mathbb{A}_h^\pi$.** We first conclude $(w_h)_{\mathbb{A}_h^\pi}^{\|}$ is determined by $\{\theta_\ell\}_{\ell=h}^H$ by backward induction.

For $h = H$, for all $s$ with $d_H^\pi(s) > 0$, we have

$$Q_H^\pi\big(s, \pi(s)\big) = r_H\big(s, \pi(s)\big) = \big\langle \phi_H\big(s, \pi(s)\big), \theta_H \big\rangle = \big\langle \phi_H\big(s, \pi(s)\big), w_H \big\rangle,$$

which implies $(w_H)_{\mathbb{A}_H^\pi}^{\|} = (\theta_H)_{\mathbb{A}_H^\pi}^{\|}$.

For $h < H$, assume $w_{h+1}$ is fixed. Using the linear MDP model $\mathbb{P}_h(s'|s, a) = \langle \phi_h(s, a), \mu_h(s') \rangle$ and the Bellman equation under $\pi$, for all $s$ with $d_h^\pi(s) > 0$,

$$
\begin{aligned}
\langle \phi_h(s, \pi(s)), w_h \rangle &= Q_h^\pi(s, \pi(s)) \\
&= r_h(s, \pi(s)) + \sum_{s'} \mathbb{P}_h(s'|s, \pi(s)) Q_{h+1}^\pi(s', \pi(s')) \, \mathrm{d}s' \\
&= \langle \phi_h(s, \pi(s)), \theta_h \rangle + \sum_{s'} \langle \phi_h(s, \pi(s)), \mu_h(s') \rangle Q_{h+1}^\pi(s', \pi(s')) \, \mathrm{d}s'.
\end{aligned}
$$

Therefore, on the subspace $\mathbb{A}_h^\pi$,

$$
(w_h)_{\mathbb{A}_h^\pi}^{\|} = (\theta_h)_{\mathbb{A}_h^\pi}^{\|} + \sum_{s'} (\mu_h(s'))_{\mathbb{A}_h^\pi}^{\|} Q_{h+1}^\pi(s', \pi(s')) \, \mathrm{d}s',
$$

which shows $(w_h)_{\mathbb{A}_h^\pi}^{\|}$ is determined once $(\theta_h)_{\mathbb{A}_h^\pi}^{\|}$ and $Q_{h+1}^\pi(\cdot, \pi(\cdot))$ are fixed.

**(ii) From $w$ to $\theta$ on $\mathbb{A}_h^\pi$.** Conversely, given $\{w_\ell\}_{\ell=h}^H$, we can reconstruct $(\theta_h)_{\mathbb{A}_h^\pi}^{\|}$ recursively on the occupancy support.

For $h = H$, the same identity gives $(\theta_H)_{\mathbb{A}_H^\pi}^{\|} = (w_H)_{\mathbb{A}_H^\pi}^{\|}$. For $h < H$, for all $s$ with $d_h^\pi(s) > 0$,

$$
\begin{aligned}
r_h(s, \pi(s)) &= Q_h^\pi(s, \pi(s)) - \sum_{s'} \mathbb{P}_h(s'|s, \pi(s)) Q_{h+1}^\pi(s', \pi(s')) \, \mathrm{d}s' \\
&= \langle \phi_h(s, \pi(s)), w_h \rangle - \sum_{s'} \langle \phi_h(s, \pi(s)), \mu_h(s') \rangle Q_{h+1}^\pi(s', \pi(s')) \, \mathrm{d}s'. \quad (17)
\end{aligned}
$$

Thus, on $\mathbb{A}_h^\pi$,

$$
(\theta_h)_{\mathbb{A}_h^\pi}^{\|} = (w_h)_{\mathbb{A}_h^\pi}^{\|} - \sum_{s'} (\mu_h(s'))_{\mathbb{A}_h^\pi}^{\|} Q_{h+1}^\pi(s', \pi(s')) \, \mathrm{d}s'.
$$

**(iii) Non-identifiability of the perpendicular component.** By construction, for any $v \in (\mathbb{A}_h^\pi)^\perp$ and any $s$ with $d_h^\pi(s) > 0$,

$$
\langle \phi_h(s, \pi(s)), v \rangle = 0.
$$

Hence $(\theta_h)_{\mathbb{A}_h^\pi}^{\perp}$ does not affect $\langle \phi_h(s, \pi(s)), \theta_h \rangle$ on the occupancy support and is not identifiable from occupancy-restricted observations. This completes the proof. $\qquad\square$

From Lemma B.6, for each step $h$ and a fixed policy $\pi^\dagger$, the mapping between $w_h$ and $\theta_h$ is uniquely determined only on the projected component onto the subspace spanned by

$$
\left\{ \phi_h(s, \pi^\dagger(s)) \,\middle|\, d_h^{\pi^\dagger}(s) > 0 \right\}.
$$

For simplicity, in the rest of this proof we use $\mathbb{A}_h$ to denote this subspace (i.e., $\mathbb{A}_h := \mathbb{A}_h^{\pi^\dagger}$).

Given the poisoned reward parameters $\hat{\theta}_h$, we can write $\hat{\theta}_h = (\hat{\theta}_h)_{\mathbb{A}_h}^{\|} + (\hat{\theta}_h)_{\mathbb{A}_h}^{\perp}$. Similarly, write the non-poisoned reward parameter $\theta_h$ as $\theta_h = (\theta_h)_{\mathbb{A}_h}^{\|} + (\theta_h)_{\mathbb{A}_h}^{\perp}$.

We now construct $\theta_h'$ to satisfy the second condition in the optimization problem (Equation (1)). Define

$$
\theta_h' := (\theta_h)_{\mathbb{A}_h}^{\|} + \alpha_h (\hat{\theta}_h)_{\mathbb{A}_h}^{\perp},
$$

where $\alpha_h \in [0, 1]$ is chosen so that $\|\theta_h'\|_2 \le \sqrt{d}$ (such an $\alpha_h$ always exists since scaling only affects the perpendicular component).

Then for any $s$ with $d_h^{\pi^\dagger}(s) > 0$, we have $\phi_h(s, \pi^\dagger(s)) \in \mathbb{A}_h$ and hence $\phi_h(s, \pi^\dagger(s))^\top (\hat{\theta}_h)_{\mathbb{A}_h}^{\perp} = 0$ by orthogonality. Therefore,

$$
\phi_h(s, \pi^\dagger(s))^\top \theta_h' = \phi_h(s, \pi^\dagger(s))^\top (\theta_h)_{\mathbb{A}_h}^{\|} = \phi_h(s, \pi^\dagger(s))^\top \theta_h,
$$

which shows that the newly defined reward matches the non-poisoned reward on all target actions that $\pi^\dagger$ visits, and thus the second condition of (Equation (1)) is satisfied.

$\square$

# C. Proof of Theorem 5.1

Throughout, we write

$$\mathcal{C}_h := \widehat{\mathcal{C}}_h, \qquad \mathcal{A}_h^{\mathrm{cmp}}(s) := \{a \in \mathcal{A} : a \neq \pi^\dagger(s)\},$$

The main text states the black-box theorem in a story-driven form. In this appendix, we make its quantitative conditions explicit. In particular, the main-text statement "Stage 1 certifies by time $T_1$ and yields a fixed attacked MDP on which the target policy is best on the certified support" is quantified here through

$$\underline{\Gamma}_{1,T_1} \geq \eta_1, \qquad \eta_1 = \Theta(1/S), \qquad \tau_{\mathrm{fix}} = \widetilde{O}(\sqrt{T_1}), \tag{18}$$

where $\underline{\Gamma}_{1,t}$ is the lower-confidence Stage 1 margin and $\tau_{\mathrm{fix}}$ is the certification time. Likewise, the main-text condition that Stage 2 succeeds is quantified here as

$$\epsilon_2^\star = \Theta(1/S), \tag{19}$$

where $\epsilon_2^\star$ is the optimal value of the Stage 2 margin-certified penalty design.

We also make explicit the learner-side interface used in the main theorem: the Stage 2 policy $\pi_t$ is generated by the *actual* attacked rewards, which include the target-side compensation schedule, while the pseudo-regret is measured against a fixed *reference* reward $\bar{r}$ that differs from the actual Stage 2 reward only on target actions. This interface is stated precisely in equation 75 below.

## C.1. Detailed black-box algorithm

## C.2. Basic model and notation

We work in the finite-horizon linear MDP model. For each stage $h \in [H]$, there exist a reward vector $\theta_h^\star \in \mathbb{R}^d$ and a vector-valued signed measure

$$\mu_h^\star = (\mu_h^{\star,(1)}, \ldots, \mu_h^{\star,(d)})$$

such that

$$r_h(s,a) = \phi_h(s,a)^\top \theta_h^\star, \qquad P_h^\star(\cdot \mid s,a) = \sum_{i=1}^d \phi_{h,i}(s,a)\, \mu_h^{\star,(i)}(\cdot), \tag{20}$$

with $\|\phi_h(s,a)\|_2 \leq 1$ and $r_h(s,a) \in [0,1]$.

Hence, for every bounded measurable $v : \mathcal{S} \to \mathbb{R}$, there exists a vector

$$m_h(v) \in \mathbb{R}^d$$

such that

$$\mathbb{E}[v(s_{h+1}) \mid s_h = s, a_h = a] = \phi_h(s,a)^\top m_h(v). \tag{21}$$

For the target policy $\pi^\dagger$, define

$$\phi_h^\dagger(s) := \phi_h(s, \pi^\dagger(s)).$$

We assume the initial-state distribution is supported on the certified set:

$$\rho(\mathcal{C}_1) = 1, \tag{22}$$

and the certified support is closed under all certified actions:

$$P_h^\star(\mathcal{C}_{h+1} \mid s,a) = 1, \qquad \forall h \in [H-1], \ \forall s \in \mathcal{C}_h, \ \forall a \in \mathcal{A}_h^{\mathrm{cmp}}(s) \cup \{\pi^\dagger(s)\}. \tag{23}$$

Therefore, every trajectory that starts in $\mathcal{C}_1$ and thereafter takes certified actions remains in the certified domain.

We also assume throughout that the attacked rewards fed to the learner are admissible, i.e. they lie in $[0,1]$. In particular, the Stage 1 rewards $\widetilde{r}_{h,t}^{(1)}$ before freezing and $\widetilde{r}_h^{(1)}$ after freezing are assumed admissible on the certified domain, and the Stage 2 actual rewards $\widehat{r}_{h,t}^{(2)}$ are assumed admissible by construction. This guarantees that every overwritten step contributes at most one unit to the shaping cost.

Finally, we will repeatedly use the standard performance-difference identity.

---

**Algorithm 3** Black-box Two-Stage Attack

---

1: **Inputs:** horizon $H$, total episodes $T$, Stage 1 length $T_1$, budget $S$, regularization $\lambda$, features $\phi$, target policy $\pi^\dagger$, certified support $\{\mathcal{C}_h\}_{h=1}^H$, threshold $\eta_1$.

2: Solve the relaxed Stage 1 CQP equation 26 to obtain $\epsilon_0^\star$ and $\{w_{0,h}\}_{h=1}^H$. If $\epsilon_0^\star \leq 0$, return NON-ATTACKABLE.

3: For each $h \in [H]$, set $q_{0,h} \leftarrow S w_{0,h}$, $q_{0,H+1} \leftarrow 0$, and $v_{0,h}(s) \leftarrow \langle \phi_h^\dagger(s), q_{0,h} \rangle$. Mark Stage 1 as *not frozen*.

4: **for** $t = 1, \ldots, T_1$ **do**

5:     **for** $h = 1, \ldots, H$ **do**

6:         Update $\Lambda_{0,h,t} \leftarrow \lambda I + \sum_{\tau=1}^{t-1} \phi_h(s_h^\tau, a_h^\tau)\phi_h(s_h^\tau, a_h^\tau)^\top$, $\widehat{b}_{0,h,t} \leftarrow \Lambda_{0,h,t}^{-1} \sum_{\tau=1}^{t-1} \phi_h(s_h^\tau, a_h^\tau) v_{0,h+1}(s_{h+1}^\tau)$, and $\widetilde{r}_{h,t}^{(1)}(s,a) \leftarrow \phi_h(s,a)^\top(q_{0,h} - \widehat{b}_{0,h,t})$.

7:     **end for**

8:     Compute $\underline{\Gamma}_{1,t} \leftarrow \min_{h \in [H]}\left(S\epsilon_0^\star - 2\sum_{j=h}^{H-1} \overline{U}_{0,j,t}\right)$.

9:     **if** $\underline{\Gamma}_{1,t} \geq \eta_1$ and Stage 1 is not frozen **then**

10:         Set $\tau_{\text{fix}} \leftarrow t$; for each $h \in [H]$, freeze $\widehat{\theta}_{0,h} \leftarrow q_{0,h} - \widehat{b}_{0,h,t}$ and $\widetilde{r}_h^{(1)}(s,a) \leftarrow \phi_h(s,a)^\top \widehat{\theta}_{0,h}$. Mark Stage 1 as *frozen*.

11:     **end if**

12:     **for** $h = 1, \ldots, H$ **do**

13:         Feed the current Stage 1 reward: if Stage 1 is not frozen, feed $\widetilde{r}_{h,t}^{(1)}(s_h^t, a_h^t)$; otherwise feed $\widetilde{r}_h^{(1)}(s_h^t, a_h^t)$.

14:     **end for**

15: **end for**

16: **if** Stage 1 is not frozen **then**

17:     Return NON-ATTACKABLE.

18: **end if**

19: Extract the clean Stage 1 suffix $\mathcal{T}_{\text{clean}} \leftarrow \{t \in [T_1] : t \geq \tau_{\text{fix}}, B_t^{(1)} = 0\}$.

20: **for** $h = 1, \ldots, H$ **do**

21:     For each $t \in \mathcal{T}_{\text{clean}}$, form $x_{h,t} \leftarrow \phi_h^\dagger(s_h^t)$ and $G_{h,t} \leftarrow \sum_{j=h}^H r_j(s_j^t, \pi^\dagger(s_j^t))$.

22:     Update $\Gamma_h^{\text{ls}} \leftarrow \lambda I + \sum_{t \in \mathcal{T}_{\text{clean}}} x_{h,t}x_{h,t}^\top$ and $\widehat{w}_h \leftarrow (\Gamma_h^{\text{ls}})^{-1} \sum_{t \in \mathcal{T}_{\text{clean}}} x_{h,t}G_{h,t}$.

23:     Define $\widehat{Q}_h^\dagger(s,a) \leftarrow \phi_h(s,a)^\top \widehat{w}_h$ and $g_h(s,a) \leftarrow \left[\widehat{Q}_h^\dagger(s,a) - \widehat{Q}_h^\dagger(s, \pi^\dagger(s))\right]_+ + \alpha_h\left(\|\phi_h(s,a)\|_{(\Gamma_h^{\text{ls}})^{-1}} + \|\phi_h^\dagger(s)\|_{(\Gamma_h^{\text{ls}})^{-1}}\right)$.

24: **end for**

25: Solve the Stage 2 margin-certified penalty design equation 6–equation 9 to obtain $\epsilon_2^\star$ and $\{u_h\}_{h=1}^H$. If $\epsilon_2^\star \leq 0$, return NON-ATTACKABLE.

26: Define the Stage 2 reference reward $\bar{r}_h(s,a) \leftarrow r_h(s,a)$ if $a = \pi^\dagger(s)$, and $\bar{r}_h(s,a) \leftarrow \left[r_h(s,a) - \langle\phi_h(s,a), u_h\rangle\right]_{[0,1]}$ otherwise.

27: **for** $h = 1, \ldots, H$ **do**

28:     Compute $D_h^{\text{tar}} \leftarrow \sum_{\substack{t \leq T_1: \\ a_h^t = \pi^\dagger(s_h^t)}} \left(\widetilde{r}_h^{(1)}(s_h^t, a_h^t) - r_h(s_h^t, a_h^t)\right)$.

29: **end for**

30: Choose an admissible predictable compensation schedule $\{c_h^t\}_{t > T_1}$ such that $\sum_{h=1}^H \sum_{t > T_1} |c_h^t| \leq \sum_{h=1}^H |D_h^{\text{tar}}|$.

31: **for** $t = T_1 + 1, \ldots, T$ **do**

32:     **for** $h = 1, \ldots, H$ **do**

33:         Feed $\hat{r}_{h,t}^{(2)}(s,a) \leftarrow r_h(s,a) + c_h^t$ if $a = \pi^\dagger(s)$, and $\hat{r}_{h,t}^{(2)}(s,a) \leftarrow \bar{r}_h(s,a)$ otherwise.

34:     **end for**

35: **end for**

---

**Lemma C.1** (Performance-difference identity). *Fix any finite-horizon MDP and any deterministic policy $\pi^\dagger$. Then for every policy $\pi$,*

$$V_1^{\pi^\dagger}(s_1) - V_1^\pi(s_1) = \mathbb{E}_\pi\left[\sum_{h=1}^{H}\left(Q_h^{\pi^\dagger}(s_h, \pi^\dagger(s_h)) - Q_h^{\pi^\dagger}(s_h, a_h)\right)\right]. \tag{24}$$

*Proof.* This is the standard finite-horizon performance-difference decomposition. It follows by writing

$$V_h^{\pi^\dagger}(s) - V_h^\pi(s) = \mathbb{E}_\pi\left[Q_h^{\pi^\dagger}(s_h, \pi^\dagger(s_h)) - Q_h^{\pi^\dagger}(s_h, a_h) + V_{h+1}^{\pi^\dagger}(s_{h+1}) - V_{h+1}^\pi(s_{h+1}) \;\middle|\; s_h = s\right]$$

and telescoping from $h = 1$ to $H$. $\qquad\square$

### C.3. Phasewise UCRL regret on fixed attacked suffixes

We will use the following phasewise regret bound on fixed attacked suffixes.

**Proposition C.2** (Phasewise UCRL-VTR regret). *Assume the learner is UCRL-VTR and the transition model belongs to a linear-mixture family of dimension $d$. Consider any attacked phase that starts after an arbitrary stopping time and thereafter interacts with a fixed attacked MDP of horizon $H$ for $K$ episodes. Then there exists a universal constant $C_{\mathrm{ucrl}} > 0$ such that, conditional on the history at the phase start, with probability at least $1 - \delta$,*

$$\mathrm{Reg}_{\mathrm{UCRL}}(K; \delta) \leq C_{\mathrm{ucrl}}\, d\, H^{3/2}\sqrt{K\log\frac{2TH}{\delta}} =: \mathfrak{R}_{\mathrm{UCRL}}(K, \delta). \tag{25}$$

*Proof.* For a fresh run on a fixed linear-mixture MDP, this is the standard UCRL-VTR regret rate. Conditioning on a stopping time only adds past data before the new phase starts; this can only shrink the confidence set and improve the design matrices. Hence the same linear-mixture order applies on the suffix process. $\qquad\square$

### C.4. Stage 1: continuation-term regression and certification

Recall the relaxed CQP(1) from the main text:

$$\begin{aligned}
\epsilon_{0,h}^\star &:= \max_{\|w\|_2 \leq \frac{1}{2S}} \min_{s \in \mathcal{C}_h}\left(\langle\phi_h^\dagger(s), w\rangle - \max_{a \neq \pi^\dagger(s)}\langle\phi_h(s, a), w\rangle\right), \\
\epsilon_0^\star &:= \min_{h \in [H]} \epsilon_{0,h}^\star.
\end{aligned} \tag{26}$$

If $\epsilon_0^\star \leq 0$, the relaxed geometric certificate fails and the two-stage construction returns NON-ATTACKABLE. Assume henceforth that $\epsilon_0^\star > 0$.

Let $w_{0,h}$ be an optimizer and define

$$q_{0,h} := Sw_{0,h}, \qquad q_{0,H+1} := 0. \tag{27}$$

Define the Stage 1 surrogate values

$$v_{0,h}(s) := \langle\phi_h^\dagger(s), q_{0,h}\rangle, \qquad v_{0,H+1} \equiv 0. \tag{28}$$

The only model-dependent quantity needed in Stage 1 is the one-step continuation term

$$\mu_{0,h}(s, a) := \mathbb{E}[v_{0,h+1}(s_{h+1}) \mid s_h = s, a_h = a]. \tag{29}$$

**Lemma C.3** (Stage 1 continuation-term linearity). *For each stage $h \in [H]$, there exists a vector $b_{0,h}^\star \in \mathbb{R}^d$ such that*

$$\mu_{0,h}(s, a) = \phi_h(s, a)^\top b_{0,h}^\star \qquad \text{for all } (s, a). \tag{30}$$

*Proof.* This is immediate from equation 21 by taking $v = v_{0,h+1}$. $\qquad\square$

At episode $t$, define the value-targeted regression target

$$y_{0,h,t} := v_{0,h+1}(s_{h+1}^t). \tag{31}$$

Let

$$\Lambda_{0,h,t} := \lambda I + \sum_{\tau=1}^{t-1} \phi_h(s_h^\tau, a_h^\tau)\phi_h(s_h^\tau, a_h^\tau)^\top, \tag{32}$$

and define the ridge estimate

$$\widehat{b}_{0,h,t} := \Lambda_{0,h,t}^{-1} \sum_{\tau=1}^{t-1} \phi_h(s_h^\tau, a_h^\tau)\, y_{0,h,\tau}. \tag{33}$$

**Lemma C.4** (Continuation-term confidence). *There exists an event $\mathcal{E}_0$ with probability at least $1 - \delta$ such that for every stage $h$, every episode $t$, and every state-action pair $(s, a)$,*

$$\left| \phi_h(s, a)^\top (\widehat{b}_{0,h,t} - b_{0,h}^\star) \right| \leq U_{0,h,t}(s, a), \tag{34}$$

*where*

$$U_{0,h,t}(s, a) := \beta_{0,h,t} \|\phi_h(s, a)\|_{\Lambda_{0,h,t}^{-1}} \tag{35}$$

*for a standard self-normalized confidence radius $\beta_{0,h,t} = \widetilde{O}(\sqrt{d})$.*

*Proof.* For each $t$, the regression target satisfies

$$\mathbb{E}[y_{0,h,t} \mid s_h^t, a_h^t, \mathcal{F}_{t,h-1}] = \mu_{0,h}(s_h^t, a_h^t) = \phi_h(s_h^t, a_h^t)^\top b_{0,h}^\star$$

by Lemma C.3. Since $|y_{0,h,t}| \leq \sup_s |v_{0,h+1}(s)| \leq \|q_{0,h+1}\|_2$, the noise is bounded, and the standard self-normalized least-squares bound gives equation 34. $\qquad\square$

Using the current continuation-term estimate, define the provisional Stage 1 reward

$$\widetilde{r}_{h,t}^{(1)}(s, a) := \phi_h(s, a)^\top \big(q_{0,h} - \widehat{b}_{0,h,t}\big). \tag{36}$$

**Lemma C.5** (Stage 1 one-step lower-confidence gap). *On the event $\mathcal{E}_0$, for every episode $t$, every stage $h$, every $s \in \mathcal{C}_h$, and every $a \in \mathcal{A}_h^{\mathrm{cmp}}(s)$,*

$$\widetilde{r}_{h,t}^{(1)}(s, \pi^\dagger(s)) + \mathbb{E}[v_{0,h+1}(s_{h+1}) \mid s_h = s, a_h = \pi^\dagger(s)]$$
$$- \widetilde{r}_{h,t}^{(1)}(s, a) - \mathbb{E}[v_{0,h+1}(s_{h+1}) \mid s_h = s, a_h = a]$$
$$\geq S\epsilon_0^\star - U_{0,h,t}(s, \pi^\dagger(s)) - U_{0,h,t}(s, a). \tag{37}$$

*Proof.* By definition,

$$\widetilde{r}_{h,t}^{(1)}(s, a) + \mathbb{E}[v_{0,h+1}(s_{h+1}) \mid s_h = s, a_h = a]$$
$$= \phi_h(s, a)^\top (q_{0,h} - \widehat{b}_{0,h,t}) + \phi_h(s, a)^\top b_{0,h}^\star$$
$$= \langle \phi_h(s, a), q_{0,h} \rangle + \phi_h(s, a)^\top (b_{0,h}^\star - \widehat{b}_{0,h,t}).$$

Subtract the two actions and apply the CQP margin together with Lemma C.4. $\qquad\square$

Define the uniform Stage 1 bonus

$$\overline{U}_{0,h,t} := \sup_{\substack{s \in \mathcal{C}_h, \\ a \in \mathcal{A}_h^{\mathrm{cmp}}(s) \cup \{\pi^\dagger(s)\}}} U_{0,h,t}(s, a). \tag{38}$$

Define the target-value tracking error under the provisional Stage 1 reward:

$$e_{0,h,t} := \sup_{s \in \mathcal{C}_h} \left| v_{0,h}(s) - V_h^{\pi^\dagger, \widetilde{r}_t^{(1)}}(s) \right|, \qquad e_{0,H+1,t} := 0. \tag{39}$$

**Lemma C.6** (Stage 1 tracking recursion). *On the event $\mathcal{E}_0$, for every $t$ and every $h \in [H]$,*

$$e_{0,h,t} \leq \overline{U}_{0,h,t} + e_{0,h+1,t}. \tag{40}$$

*Consequently,*

$$e_{0,h,t} \leq \sum_{j=h}^{H-1} \overline{U}_{0,j,t}. \tag{41}$$

*Proof.* Fix $s \in \mathcal{C}_h$. By definition of $v_{0,h}$ and $\widetilde{r}_t^{(1)}$,

$$v_{0,h}(s) = \widetilde{r}_{h,t}^{(1)}(s, \pi^\dagger(s)) + \mathbb{E}[v_{0,h+1}(s_{h+1}) \mid s_h = s, a_h = \pi^\dagger(s)] + \phi_h^\dagger(s)^\top (\widehat{b}_{0,h,t} - b_{0,h}^\star).$$

Also,

$$V_h^{\pi^\dagger, \widetilde{r}_t^{(1)}}(s) = \widetilde{r}_{h,t}^{(1)}(s, \pi^\dagger(s)) + \mathbb{E}[V_{h+1}^{\pi^\dagger, \widetilde{r}_t^{(1)}}(s_{h+1}) \mid s_h = s, a_h = \pi^\dagger(s)].$$

Subtracting, using equation 23, and applying Lemma C.4 yields

$$|v_{0,h}(s) - V_h^{\pi^\dagger, \widetilde{r}_t^{(1)}}(s)| \leq \overline{U}_{0,h,t} + e_{0,h+1,t}.$$

Taking the supremum over $s \in \mathcal{C}_h$ proves the recursion, and iteration gives the sum bound. $\square$

**Proposition C.7** (Stage 1 lower-confidence certified $Q$-gap). *On the event $\mathcal{E}_0$, for every episode $t$, every stage $h$, every $s \in \mathcal{C}_h$, and every $a \in \mathcal{A}_h^{\mathrm{cmp}}(s)$,*

$$Q_h^{\pi^\dagger, \widetilde{r}_t^{(1)}}(s, \pi^\dagger(s)) - Q_h^{\pi^\dagger, \widetilde{r}_t^{(1)}}(s, a) \geq S\epsilon_0^\star - 2 \sum_{j=h}^{H-1} \overline{U}_{0,j,t}. \tag{42}$$

*Proof.* Fix $h, s, a$. Write

$$\begin{aligned}
&Q_h^{\pi^\dagger, \widetilde{r}_t^{(1)}}(s, \pi^\dagger(s)) - Q_h^{\pi^\dagger, \widetilde{r}_t^{(1)}}(s, a) \\
&= \widetilde{r}_{h,t}^{(1)}(s, \pi^\dagger(s)) - \widetilde{r}_{h,t}^{(1)}(s, a) \\
&\quad + \mathbb{E}\left[V_{h+1}^{\pi^\dagger, \widetilde{r}_t^{(1)}}(s_{h+1}) \mid s_h = s, a_h = \pi^\dagger(s)\right] - \mathbb{E}\left[V_{h+1}^{\pi^\dagger, \widetilde{r}_t^{(1)}}(s_{h+1}) \mid s_h = s, a_h = a\right].
\end{aligned}$$

Add and subtract the two continuation terms

$$\mathbb{E}[v_{0,h+1}(s_{h+1}) \mid s_h = s, a_h = \pi^\dagger(s)], \qquad \mathbb{E}[v_{0,h+1}(s_{h+1}) \mid s_h = s, a_h = a].$$

By Lemma C.5,

$$\widetilde{r}_{h,t}^{(1)}(s, \pi^\dagger(s)) + \mathbb{E}[v_{0,h+1}(s_{h+1}) \mid s_h = s, a_h = \pi^\dagger(s)] - \widetilde{r}_{h,t}^{(1)}(s, a) - \mathbb{E}[v_{0,h+1}(s_{h+1}) \mid s_h = s, a_h = a] \geq S\epsilon_0^\star - 2\overline{U}_{0,h,t}.$$

The remaining continuation mismatch is bounded by

$$2e_{0,h+1,t} \leq 2 \sum_{j=h+1}^{H-1} \overline{U}_{0,j,t}$$

using Lemma C.6. Combining the two bounds yields

$$Q_h^{\pi^\dagger, \widetilde{r}_t^{(1)}}(s, \pi^\dagger(s)) - Q_h^{\pi^\dagger, \widetilde{r}_t^{(1)}}(s, a) \geq S\epsilon_0^\star - 2 \sum_{j=h}^{H-1} \overline{U}_{0,j,t},$$

as claimed. $\square$

Define the lower-confidence Stage 1 margin at episode $t$ by

$$\underline{\Gamma}_{1,t} := \min_{h \in [H]} \left( S\epsilon_0^\star - 2 \sum_{j=h}^{H-1} \overline{U}_{0,j,t} \right). \tag{43}$$

Fix a target certification threshold $\eta_1 > 0$ and define

$$\tau_{\text{fix}} := \inf\{t \leq T_1 : \underline{\Gamma}_{1,t} \geq \eta_1\}. \tag{44}$$

At time $\tau_{\text{fix}}$, freeze the Stage 1 steering reward:

$$\widehat{\theta}_{0,h} := q_{0,h} - \widehat{b}_{0,h,\tau_{\text{fix}}}, \qquad \widetilde{r}_h^{(1)}(s,a) := \phi_h(s,a)^\top \widehat{\theta}_{0,h}. \tag{45}$$

**Lemma C.8** (Thresholded Stage 1 certification). *If*

$$\underline{\Gamma}_{1,T_1} \geq \eta_1, \tag{46}$$

*then $\tau_{\text{fix}} \leq T_1$, and the frozen Stage 1 reward satisfies*

$$Q_h^{\pi^\dagger, \widetilde{r}^{(1)}}(s, \pi^\dagger(s)) - Q_h^{\pi^\dagger, \widetilde{r}^{(1)}}(s, a) \geq \eta_1, \qquad \forall h, \ \forall s \in \mathcal{C}_h, \ \forall a \in \mathcal{A}_h^{\text{cmp}}(s). \tag{47}$$

*Proof.* Since $\underline{\Gamma}_{1,T_1} \geq \eta_1$, the set in equation 44 is nonempty, hence $\tau_{\text{fix}} \leq T_1$. Evaluating Proposition C.7 at $t = \tau_{\text{fix}}$ yields

$$Q_h^{\pi^\dagger, \widetilde{r}^{(1)}}(s, \pi^\dagger(s)) - Q_h^{\pi^\dagger, \widetilde{r}^{(1)}}(s, a) \geq \underline{\Gamma}_{1,\tau_{\text{fix}}} \geq \eta_1.$$

$\square$

**Corollary C.9** (Stage 1 frozen target optimality). *On the event $\mathcal{E}_0$, if equation 46 holds, then on the reachable certified domain, the policy $\pi^\dagger$ is optimal in the frozen Stage 1 attacked MDP $(P^\star, \widetilde{r}^{(1)})$, and the gap bound equation 47 holds with margin $\eta_1$.*

*Proof.* Apply Lemma C.8. Since $\rho(\mathcal{C}_1) = 1$ and equation 23 implies that every reachable state stays inside $\{\mathcal{C}_h\}$, backward induction shows that no policy can outperform $\pi^\dagger$ on the reachable certified domain. $\square$

### C.5. Stage 1 clean history and warm-start corruption

Define the Stage 1 bad-episode indicator on the frozen forcing suffix:

$$B_t^{(1)} := \mathbf{1}\left\{ t \geq \tau_{\text{fix}}, \ \exists h \in [H] \text{ s.t. } s_h^t \in \mathcal{C}_h, \ a_h^t \in \mathcal{A}_h^{\text{cmp}}(s_h^t) \right\}, \qquad M_{\text{bad}}^{(1)} := \sum_{t=1}^{T_1} B_t^{(1)}. \tag{48}$$

Define the set of clean Stage 1 suffix episodes by

$$\mathcal{T}_{\text{clean}} := \{t \in [T_1] : t \geq \tau_{\text{fix}}, \ B_t^{(1)} = 0\}. \tag{49}$$

**Lemma C.10** (Clean suffix episodes are genuine target rollouts). *On the event $\mathcal{E}_0$, if $t \in \mathcal{T}_{\text{clean}}$, then for every stage $h \in [H]$,*

$$s_h^t \in \mathcal{C}_h, \qquad a_h^t = \pi^\dagger(s_h^t).$$

*In particular, the clean reward sequence observed on episode $t$ is a genuine rollout of $\pi^\dagger$ in the clean MDP.*

*Proof.* Since $\rho(\mathcal{C}_1) = 1$, the episode starts in $\mathcal{C}_1$. Because $B_t^{(1)} = 0$, no certified non-target action is selected on that episode. Since $\mathcal{A}_h^{\text{cmp}}(s) \cup \{\pi^\dagger(s)\} = \mathcal{A}$, this means $a_h^t = \pi^\dagger(s_h^t)$ for every $h$ with $s_h^t \in \mathcal{C}_h$. Support closure then propagates $s_{h+1}^t \in \mathcal{C}_{h+1}$. An induction on $h$ proves the claim. $\square$

**Proposition C.11** (Stage 1 suffix regret-to-count). *Assume equation 46 and $\eta_1 = \Theta(1/S)$. Then*

$$M_{\text{bad}}^{(1)} \leq \frac{\mathfrak{R}_{\text{UCRL}}(T_1 - \tau_{\text{fix}} + 1, \delta)}{\eta_1} = \widetilde{O}(\sqrt{T_1}). \tag{50}$$

*Consequently,*

$$|\mathcal{T}_{\text{clean}}| \geq T_1 - \tau_{\text{fix}} + 1 - \widetilde{O}(\sqrt{T_1}). \tag{51}$$

*Proof.* On the frozen suffix, the learner interacts with the fixed attacked MDP $(P^\star, \widetilde{r}^{(1)})$, and $\pi^\dagger$ is optimal on the reachable certified domain by Corollary C.9. For every bad suffix episode $t$, let $h_t^{\text{dev}}$ be its first certified deviation stage. Then

$$Q_{h_t^{\text{dev}}}^{\pi^\dagger, \widetilde{r}^{(1)}}(s_{h_t^{\text{dev}}}^t, \pi^\dagger(s_{h_t^{\text{dev}}}^t)) - Q_{h_t^{\text{dev}}}^{\pi^\dagger, \widetilde{r}^{(1)}}(s_{h_t^{\text{dev}}}^t, a_{h_t^{\text{dev}}}^t) \geq \eta_1.$$

By Lemma C.1, the entire episode therefore incurs at least $\eta_1$ regret relative to $\pi^\dagger$. Summing over bad suffix episodes and applying Proposition C.2 to the suffix of length $T_1 - \tau_{\text{fix}} + 1$ proves the bound. $\square$

Define the total Stage 1 warm-start corruption mass

$$S_0 := \sum_{t=1}^{T_1} \sum_{h=1}^{H} \mathbf{1}\{s_h^t \in \mathcal{C}_h, \ a_h^t \in \mathcal{A}_h^{\text{cmp}}(s_h^t)\}. \tag{52}$$

**Corollary C.12** (Warm-start corruption size). *Assume equation 46, $\eta_1 = \Theta(1/S)$, and $\tau_{\text{fix}} = \widetilde{O}(\sqrt{T_1})$. Then*

$$S_0 \leq H(\tau_{\text{fix}} + M_{\text{bad}}^{(1)}) = \widetilde{O}(\sqrt{T_1}). \tag{53}$$

*Proof.* Before $\tau_{\text{fix}}$, every episode contributes at most $H$ overwritten certified steps. After $\tau_{\text{fix}}$, only bad suffix episodes contribute certified non-target selections. Use Proposition C.11. $\square$

### C.6. Stage 2: clean target-$Q$ confidence and conservative gap estimation

For the clean MDP and the target policy $\pi^\dagger$, define the target-policy $Q$-function

$$Q_h^{\pi^\dagger, r}(s, a) = r_h(s, a) + \mathbb{E}\left[\sum_{j=h+1}^{H} r_j(s_j, \pi^\dagger(s_j)) \,\middle|\, s_h = s, \ a_h = a, \ a_{h+1:H} = \pi^\dagger\right].$$

**Lemma C.13** (Linear representation of the target-policy $Q$-function). *For each stage $h \in [H]$, there exists a vector $w_h^\dagger \in \mathbb{R}^d$ such that*

$$Q_h^{\pi^\dagger, r}(s, a) = \phi_h(s, a)^\top w_h^\dagger \qquad \text{for all } (s, a). \tag{54}$$

*Proof.* This is the standard linear-MDP Bellman closure. We prove it by backward induction. At stage $H$,

$$Q_H^{\pi^\dagger, r}(s, a) = r_H(s, a) = \phi_H(s, a)^\top \theta_H^\star,$$

so take $w_H^\dagger = \theta_H^\star$. Assume

$$Q_{h+1}^{\pi^\dagger, r}(s, a) = \phi_{h+1}(s, a)^\top w_{h+1}^\dagger.$$

Then

$$Q_h^{\pi^\dagger, r}(s, a) = \phi_h(s, a)^\top \theta_h^\star + \mathbb{E}\left[\phi_{h+1}^\dagger(s_{h+1})^\top w_{h+1}^\dagger \,\middle|\, s_h = s, a_h = a\right],$$

and the second term is linear in $\phi_h(s, a)$ by equation 21. $\square$

For each clean episode $t \in \mathcal{T}_{\text{clean}}$ and each stage $h$, define the clean Monte Carlo return

$$G_{h,t} := \sum_{j=h}^{H} r_j(s_j^t, \pi^\dagger(s_j^t)). \tag{55}$$

Since the attacker observes the clean reward before replacing it, the quantity $G_{h,t}$ is available on every clean episode.

Define

$$x_{h,t} := \phi_h^\dagger(s_h^t), \qquad \Gamma_h^{\text{ls}} := \lambda I + \sum_{t \in \mathcal{T}_{\text{clean}}} x_{h,t} x_{h,t}^\top, \tag{56}$$

and the ridge estimator

$$\widehat{w}_h := (\Gamma_h^{\text{ls}})^{-1} \sum_{t \in \mathcal{T}_{\text{clean}}} x_{h,t} G_{h,t}. \tag{57}$$

**Lemma C.14** (Stage 2 target-$Q$ confidence). *There exists an event $\mathcal{E}_2$ with probability at least $1 - 2\delta$ such that for every $h \in [H]$ and every $x \in \mathbb{R}^d$,*

$$|x^\top(\widehat{w}_h - w_h^\dagger)| \le \alpha_h \|x\|_{(\Gamma_h^{\text{ls}})^{-1}}, \tag{58}$$

*where*

$$\alpha_h = \widetilde{O}(H\sqrt{d}). \tag{59}$$

*Proof.* For each $t \in \mathcal{T}_{\text{clean}}$, Lemma C.10 implies that episode $t$ is a genuine target rollout in the clean MDP. Hence

$$\mathbb{E}[G_{h,t} \mid s_h^t, \mathcal{F}_{t,h-1}] = Q_h^{\pi^\dagger, r}(s_h^t, \pi^\dagger(s_h^t)) = x_{h,t}^\top w_h^\dagger,$$

by Lemma C.13. Since $0 \le G_{h,t} \le H - h + 1$, the noise is conditionally sub-Gaussian with scale $O(H)$. The self-normalized least-squares bound then gives equation 58. $\qquad\square$

Define the plug-in estimate

$$\widehat{Q}_h^\dagger(s, a) := \phi_h(s, a)^\top \widehat{w}_h. \tag{60}$$

**Lemma C.15** (Target-side mismatch). *On the event $\mathcal{E}_2$, under the excitation condition from the main theorem and $\tau_{\text{fix}} = \widetilde{O}(\sqrt{T_1})$,*

$$\gamma_{\text{tar}} := \max_{h \in [H]} \sup_{s \in \mathcal{C}_h} \left| \widehat{Q}_h^\dagger(s, \pi^\dagger(s)) - Q_h^{\pi^\dagger, r}(s, \pi^\dagger(s)) \right| \le \widetilde{O}\left( \frac{\sqrt{d}}{\sqrt{T_1}} \right). \tag{61}$$

*Proof.* By Proposition C.11, the clean suffix has size $T_1 - \widetilde{O}(\sqrt{T_1})$. Under the theorem's excitation condition, the design matrix $\Gamma_h^{\text{ls}}$ is therefore of order $T_1$ on every certified stage. Combining this with Lemma C.14 proves the claim. $\qquad\square$

For $s \in \mathcal{C}_h$ and $a \in \mathcal{A}_h^{\text{cmp}}(s)$, define

$$g_h(s, a) := \left[ \widehat{Q}_h^\dagger(s, a) - \widehat{Q}_h^\dagger(s, \pi^\dagger(s)) \right]_+$$
$$+ \alpha_h \left( \|\phi_h(s, a)\|_{(\Gamma_h^{\text{ls}})^{-1}} + \|\phi_h^\dagger(s)\|_{(\Gamma_h^{\text{ls}})^{-1}} \right). \tag{62}$$

**Lemma C.16** (Conservative disadvantage bound). *On the event $\mathcal{E}_2$, for every $h \in [H]$, every $s \in \mathcal{C}_h$, and every $a \in \mathcal{A}_h^{\text{cmp}}(s)$,*

$$Q_h^{\pi^\dagger, r}(s, a) - Q_h^{\pi^\dagger, r}(s, \pi^\dagger(s)) \le g_h(s, a). \tag{63}$$

*Proof.* Add and subtract the plug-in estimate:

$$Q_h^{\pi^\dagger, r}(s, a) - Q_h^{\pi^\dagger, r}(s, \pi^\dagger(s))$$
$$= \left[ \widehat{Q}_h^\dagger(s, a) - \widehat{Q}_h^\dagger(s, \pi^\dagger(s)) \right]$$
$$+ \left[ Q_h^{\pi^\dagger, r}(s, a) - \widehat{Q}_h^\dagger(s, a) \right] + \left[ \widehat{Q}_h^\dagger(s, \pi^\dagger(s)) - Q_h^{\pi^\dagger, r}(s, \pi^\dagger(s)) \right].$$

Apply Lemma C.14 to the last two terms and take the positive part on the plug-in difference. $\qquad\square$

## C.7. Stage 2: margin-certified penalty and integrated compensation

Define the Stage 2 margin-certified program by

$$\epsilon_2^\star := \max_{\epsilon, \{u_h\}_{h=1}^H} \epsilon \tag{64}$$

$$\text{s.t.} \quad \langle \phi_h(s,a), u_h \rangle \geq g_h(s,a) + \epsilon, \qquad \forall h, \ \forall s \in \mathcal{C}_h, \ \forall a \in \mathcal{A}_h^{\mathrm{cmp}}(s), \tag{65}$$

$$\langle \phi_h^\dagger(s), u_h \rangle = 0, \qquad \forall h, \ \forall s \in \mathcal{C}_h, \tag{66}$$

$$\|u_h\|_2 \leq \sqrt{d}, \qquad \forall h. \tag{67}$$

If $\epsilon_2^\star \leq 0$, the Stage 2 certificate fails and the construction returns NON-ATTACKABLE. Assume henceforth that $\epsilon_2^\star > 0$, and let $\{u_h\}_{h=1}^H$ be any optimizer.

Define the stationary Stage 2 reference reward

$$\bar{r}_h(s,a) = \begin{cases} r_h(s,a), & a = \pi^\dagger(s), \\ \left[ r_h(s,a) - \langle \phi_h(s,a), u_h \rangle \right]_{[0,1]}, & a \in \mathcal{A}_h^{\mathrm{cmp}}(s). \end{cases} \tag{68}$$

**Proposition C.17** (Stage 2 certified local gap for the reference reward). *On the event $\mathcal{E}_2$, for every $h \in [H]$, every $s \in \mathcal{C}_h$, and every $a \in \mathcal{A}_h^{\mathrm{cmp}}(s)$,*

$$Q_h^{\pi^\dagger, \bar{r}}(s, \pi^\dagger(s)) - Q_h^{\pi^\dagger, \bar{r}}(s, a) \geq \epsilon_2^\star. \tag{69}$$

*Proof.* For a non-target action $a$,

$$\bar{r}_h(s,a) \leq r_h(s,a) - \langle \phi_h(s,a), u_h \rangle$$

because clipping only decreases the non-target reward. Also, $\bar{r}_h(s, \pi^\dagger(s)) = r_h(s, \pi^\dagger(s))$. Therefore,

$$\begin{aligned}
&Q_h^{\pi^\dagger, \bar{r}}(s, \pi^\dagger(s)) - Q_h^{\pi^\dagger, \bar{r}}(s, a) \\
&\geq Q_h^{\pi^\dagger, r}(s, \pi^\dagger(s)) - \left( Q_h^{\pi^\dagger, r}(s, a) - \langle \phi_h(s,a), u_h \rangle \right) \\
&= \langle \phi_h(s,a), u_h \rangle - \left( Q_h^{\pi^\dagger, r}(s, a) - Q_h^{\pi^\dagger, r}(s, \pi^\dagger(s)) \right).
\end{aligned}$$

By equation 7 and Lemma C.16,

$$\langle \phi_h(s,a), u_h \rangle - \left( Q_h^{\pi^\dagger, r}(s, a) - Q_h^{\pi^\dagger, r}(s, \pi^\dagger(s)) \right) \geq \epsilon_2^\star.$$

This proves equation 69. $\qquad \square$

**Corollary C.18** (Stage 2 reference optimality). *On the event $\mathcal{E}_2$, if $\epsilon_2^\star > 0$, then $\pi^\dagger$ is optimal on the reachable certified domain under the Stage 2 reference reward $\bar{r}$.*

*Proof.* By Proposition C.17, every certified comparison action has strictly smaller $Q^{\pi^\dagger, \bar{r}}$-value than the target action at the same certified state. Since $\rho(\mathcal{C}_1) = 1$ and equation 23 keeps the process inside the certified domain, backward induction yields the claim. $\qquad \square$

During Stage 1, target visits use the steering reward rather than the clean reward, which leaves a finite target-side discrepancy. Define the target debt by

$$D_h^{\mathrm{tar}} := \sum_{\substack{t \leq T_1: \\ a_h^t = \pi^\dagger(s_h^t)}} \left( \widetilde{r}_h^{(1)}(s_h^t, a_h^t) - r_h(s_h^t, a_h^t) \right), \qquad h \in [H]. \tag{70}$$

**Lemma C.19** (Target-debt size).

$$\sum_{h=1}^H |D_h^{\mathrm{tar}}| \leq H T_1. \tag{71}$$

*Proof.* Each summand in equation 70 is the difference of two admissible rewards in $[0, 1]$, hence has absolute value at most 1. For each fixed $(t, h)$, at most one target action is taken. Summing over $h \in [H]$ and $t \leq T_1$ proves the claim. □

Assume there exists an admissible predictable compensation schedule $\{c_h^t\}_{t > T_1}$ such that

$$C_{\text{corr}}(T_1) := \sum_{h=1}^{H} \sum_{t > T_1} |c_h^t| \leq \sum_{h=1}^{H} |D_h^{\text{tar}}|. \tag{72}$$

The actual Stage 2 reward is then

$$\hat{r}_{h,t}^{(2)}(s, a) = \begin{cases} r_h(s, a) + c_h^t, & a = \pi^\dagger(s), \\ \bar{r}_h(s, a), & a \in \mathcal{A}_h^{\text{cmp}}(s). \end{cases} \tag{73}$$

**Learner-side warm-start interface.** Define the Stage 2 pseudo-regret against the stationary reference reward $\bar{r}$ by

$$R_T^{(2)}(\bar{r}) := \sum_{t=T_1+1}^{T} \left( V_1^{\pi^\dagger, \bar{r}}(s_1^t) - V_1^{\pi_t, \bar{r}}(s_1^t) \right), \tag{74}$$

where $\pi_t$ is the learner's episode-$t$ policy induced by the actual Stage 2 observations $\hat{r}_{h,t}^{(2)}$.

The main theorem assumes the following interface: there exists an event $\mathcal{E}_{\text{wr}}$ with probability at least $1 - \delta$ such that

$$R_T^{(2)}(\bar{r}) \leq C_{\text{wr}} \, d \, H^{3/2} \sqrt{T} \left( 1 + S_0 + \gamma_{\text{tar}} \sqrt{T} \right), \tag{75}$$

for some universal constant $C_{\text{wr}} > 0$. This interface already measures the learner's policy $\pi_t$ under the actual Stage 2 observations, and therefore implicitly accounts for the effect of the target-side compensation on the learner's behavior. Its direct shaping cost will be accounted for separately through $C_{\text{corr}}(T_1)$.

**Corollary C.20** (Explicit Stage 2 regret scale). *On* $\mathcal{E}_{\text{wr}} \cap \mathcal{E}_2$,

$$R_T^{(2)}(\bar{r}) \leq \tilde{O}\left( d \, H^{3/2} \sqrt{T} \right) \left( \sqrt{T_1} + \frac{T}{\sqrt{T_1}} \right). \tag{76}$$

*Proof.* Use Corollary C.12 and Lemma C.15 in equation 75. □

## C.8. Stage 2 disagreement count and total cost

Define the Stage 2 bad-episode indicator

$$\begin{aligned} B_t^{(2)} &:= \mathbf{1}\left\{ \exists h \in [H] \text{ s.t. } s_h^t \in \mathcal{C}_h, \, a_h^t \in \mathcal{A}_h^{\text{cmp}}(s_h^t) \right\}, \qquad t > T_1, \\ M_{\text{bad}}^{(2)} &:= \sum_{t > T_1} B_t^{(2)}. \end{aligned} \tag{77}$$

**Proposition C.21** (Stage 2 deviation charging). *On* $\mathcal{E}_2 \cap \mathcal{E}_{\text{wr}}$,

$$\epsilon_2^\star M_{\text{bad}}^{(2)} \leq R_T^{(2)}(\bar{r}). \tag{78}$$

*Consequently,*

$$M_{\text{bad}}^{(2)} \leq \tilde{O}\left( d \, H^{3/2} S \left( \sqrt{T T_1} + \frac{T}{\sqrt{T_1}} \right) \right). \tag{79}$$

*Proof.* Fix a bad episode $t > T_1$, and let $h_t^{\text{dev}}$ be its first certified deviation stage. Since $\rho(\mathcal{C}_1) = 1$ and equation 23 holds, the entire episode remains in the certified domain. By Proposition C.17,

$$Q_{h_t^{\text{dev}}}^{\pi^\dagger, \bar{r}}(s_{h_t^{\text{dev}}}^t, \pi^\dagger(s_{h_t^{\text{dev}}}^t)) - Q_{h_t^{\text{dev}}}^{\pi^\dagger, \bar{r}}(s_{h_t^{\text{dev}}}^t, a_{h_t^{\text{dev}}}^t) \geq \epsilon_2^\star.$$

Applying Lemma C.1 to the reference reward $\bar{r}$, the whole episode contributes at least $\epsilon_2^\star$ to the pseudo-regret $R_T^{(2)}(\bar{r})$. Summing over bad episodes proves equation 78.

Then use Corollary C.20 and the theorem assumption $\epsilon_2^\star = \Theta(1/S)$. □

**Proposition C.22** (Stage 2 penalty cost). *Let*

$$U_{\max} := \sup_{\substack{h \in [H], \ s \in \mathcal{C}_h, \\ a \in \mathcal{A}_h^{\mathrm{cmp}}(s)}} \langle \phi_h(s,a), u_h \rangle. \tag{80}$$

*Then*

$$U_{\max} \leq \sqrt{d}, \tag{81}$$

*and the Stage 2 cost satisfies*

$$\mathrm{Cost}^{(2)} \leq C_{\mathrm{corr}}(T_1) + H U_{\max} M_{\mathrm{bad}}^{(2)} \leq H T_1 + \widetilde{O}\left( d^{3/2} H^{5/2} S \left( \sqrt{T T_1} + \frac{T}{\sqrt{T_1}} \right) \right). \tag{82}$$

*Proof.* The bound $U_{\max} \leq \sqrt{d}$ follows from $\|u_h\|_2 \leq \sqrt{d}$ and $\|\phi_h(s,a)\|_2 \leq 1$.

For every non-target action, $\langle \phi_h(s,a), u_h \rangle \geq g_h(s,a) + \epsilon_2^\star > 0$, so the clipping in equation 68 can only reduce the penalty magnitude. Hence on every non-target step,

$$|r_h(s,a) - \bar{r}_h(s,a)| \leq \langle \phi_h(s,a), u_h \rangle \leq U_{\max}.$$

Every bad Stage 2 episode can incur non-target penalty on at most $H$ steps, and each such step contributes at most $U_{\max}$.

Adding the target-side compensation cost and using Lemma C.19 and equation 72 proves the claim. $\qquad \square$

## C.9. Proof of Theorem 5.1

*Proof of Theorem 5.1.* Let

$$\mathcal{E}_{\mathrm{bb}} := \mathcal{E}_0 \cap \mathcal{E}_2 \cap \mathcal{E}_{\mathrm{wr}} \cap \{\underline{\Gamma}_{1,T_1} \geq \eta_1\}.$$

By the union bound,

$$\Pr(\mathcal{E}_{\mathrm{bb}}) \geq 1 - (\delta + 2\delta + \delta) = 1 - 4\delta.$$

On $\mathcal{E}_{\mathrm{bb}}$, Lemma C.8 yields $\tau_{\mathrm{fix}} \leq T_1$ and a frozen Stage 1 gap of size $\eta_1$. Under the theorem assumption $\eta_1 = \Theta(1/S)$, Proposition C.11 gives a clean Stage 1 suffix of size

$$|\mathcal{T}_{\mathrm{clean}}| \geq T_1 - \tau_{\mathrm{fix}} + 1 - \widetilde{O}(\sqrt{T_1}).$$

Under the quantitative Stage 1 assumption $\tau_{\mathrm{fix}} = \widetilde{O}(\sqrt{T_1})$, this implies

$$|\mathcal{T}_{\mathrm{clean}}| \geq T_1 - \widetilde{O}(\sqrt{T_1}), \qquad S_0 = \widetilde{O}(\sqrt{T_1}).$$

Lemma C.15 gives

$$\gamma_{\mathrm{tar}} = \widetilde{O}\left( \frac{\sqrt{d}}{\sqrt{T_1}} \right).$$

Lemma C.19 and equation 72 give

$$C_{\mathrm{corr}}(T_1) \leq H T_1.$$

Next, Corollary C.20 and Proposition C.21 yield

$$M_{\mathrm{bad}}^{(2)} \leq \widetilde{O}\left( d H^{3/2} S \left( \sqrt{T T_1} + \frac{T}{\sqrt{T_1}} \right) \right).$$

By Proposition C.22,

$$\mathrm{Cost}^{(2)} \leq H T_1 + \widetilde{O}\left( d^{3/2} H^{5/2} S \left( \sqrt{T T_1} + \frac{T}{\sqrt{T_1}} \right) \right).$$

Finally, the Stage 1 phase consists of an estimation prefix of length at most $\tau_{\text{fix}}$ and a frozen forcing suffix up to $T_1$. Since all fed rewards are admissible, at most $H$ rewards can be modified per episode, so

$$\text{Cost}^{(1)} \leq HT_1.$$

Combining the two stages gives

$$\text{Cost}(T, T_1) = \text{Cost}^{(1)} + \text{Cost}^{(2)} \leq \widetilde{O}\left( HT_1 + d^{3/2} H^{5/2} S \left( \sqrt{T T_1} + \frac{T}{\sqrt{T_1}} \right) \right). \tag{83}$$

This proves the claimed symbolic tradeoff. In particular, choosing

$$T_1 = \Theta(\sqrt{T})$$

gives

$$\text{Cost}(T, \Theta(\sqrt{T})) \leq \widetilde{O}\left( d^{3/2} H^{5/2} S T^{3/4} \right).$$

$\square$

# D. Empirical Design and Results

In this appendix, we include the rest of the details about our experiments and results left out of section 6 due to space constraints. We begin by introducing our empirical set up, covering the objectives for both experiments. Then, we present implementation details where we will provide a link to our code. Finally, we present the rest of our result plots for each environment and setting.

## D.1. Main Paper Full Experiment Details

We run two experiments to test our theoretical results. The first experiment seeks to collect data on the perturbations used by the adversary as well as the changes in the victim learning algorithm's policy due to the reward poisoning attack. The second seeks to check how accurate our CQP's characterization is in predicting the outcome of a reward poisoning attack using two metrics.

**Obtaining Linear MDPs via CTRL** A core challenge in implementing and testing our theoretical framework is how to obtain good linear MDP environments. It is well known that most interesting problem settings rarely follow a linear relationship as established in our assumptions. To tackle this challenge, we propose to use the **C**on**T**rastive **R**epresentation **L**earning (CTRL) (Zhang et al., 2022).

CTRL constructs a linear MDP representation by approximating every variable and mapping function required to define it (see our main text for linear MDP definition). It does this by using a Soft Actor-Critic (SAC) (Haarnoja et al., 2018) algorithm that interacts with the environment to represent, collecting data to construct the linear MDP. What is special about CTRL is that the learned linear MDPs are accurate enough to allow SOTA performance by other learning algorithms using the learned representation. While this might seem counterintuitive (real world breaks linearity), the lack of assumptions on the mapping functions themselves ($\phi(\cdot, \cdot)$ and $\mu(\cdot)$) allows enough flexibility to model real-world problems. By parameterizing these mapping functions as neural networks, CTRL can capture complex behaviors while respecting assumptions.

## D.2. Black Box vs White Box

The core difference between the black box and white box attack algorithms lies in the access given to the algorithm. The white box algorithm is given access to the variables and mapping functions of its target linear MDP. The black box algorithm, on the other hand, has no access and must estimate these parameters. Regardless of using the actual or learned linear MDP parameters, both learning algorithms solve the CQP characterization to obtain their attack strategies.

**Black Box Implementation** Implementing the black box algorithm is challenging due to its complexity and the theoretical-practical implementation gap: while our theory might be sound, it is difficult to implement these ideas exactly and in a computationally-feasible manner. We tackle this by using different estimation and approximation techniques to execute Algorithm 1. For stage 1, our algorithm uses light perturbation attacks based on the deviation between victim and adversarial target policies. The idea is to obtain as many $\pi^\dagger$-following trajectories (including the original reward signal) for a good estimation of the environment. We run stage 1 up to step 5,000, after which we estimate the parameters needed to solve the CQP by optimization (e.g. ridge regression).

Stage 2 of the black box algorithm uses the collected data and estimated variables to solve the CQP using the popular CVXPY (Diamond & Boyd, 2016) library. This way, we obtain the equivalent of $u_h$ from our black box theory section. The adversary then carries out their attack by simply replacing the reward signal observed when the victim learning agent disagrees with the adversarial target policy (which we use as an "oracle").

we first provide our plots on reward poisoning against LSVI-UCB (Jin et al., 2020) for the experiment described in section 6.2. Additionally, we provide extended results left out from section 6.4 on reward poisoning against CTRL. Finally, we provide an additional "sanity check" experiment to further back our claim that the linear MDP representation obtained by allows our characterization to be accurate regardless of the linearity of the MDP environments. For each new result, we provide further analysis on each experiment to complement our main analysis from 6.4. We also publish our implementation[1].

---

[1] `https://github.com/aguilarjose11/IntrinsicRobustness_ICML26`

### D.3. Testing Theoretical Predictions on Cumulative Perturbations and Victim Policy Convergence

Our first experiment aims to observe the behavior of the adversary's perturbations and its effects on the victim learning algorithm. As described in section 6.1, we use MuJoCo's Half-Cheetah, Pendulum, and MountainCart continuous environments. Aside of these popular reinforcement learning benchmarking environments, we also test our results on the FrozenLake-8x8 tabular MDP environment. We use CTRL to obtain linear MDP representations for these environments.

A crucial aspect of our first experiments is how we *obtain vulnerable and robust environments* to benchmark from the same environment. Due to the stochasticity of CTRL, the learned linear MDP representations will always differ, becoming more similar as CTRL converges. This variability allows us to randomly sample environments whose intrinsic geometry makes the adversarial objective costly. We run CTRL until we have enough environments for each trial, using early-stopping and manually-tuned hyperparameters[2] to achieve this.

Our main experiment code simulates the interaction between the learning and reward poisoning algorithms in vulnerable and robust environments. We allow up to $T = 20,000$ episodes/trials, with episode lengths varying for each problem[3]. We collect the reward perturbation (difference between poisoned reward and original reward), which we present in our per-step and cumulative plots. We also collect the policy difference metric: L2 norm between parameterizing matrices of both policies. We assume the same policy architecture for the victim and adversarial target policies–a three-layer Deep Neural Network with a number of neurons. We simply concatenate all layers into a single vector and then compute the L2 norm.

**Victim Algorithms** We test two different learning algorithms: the classical LSVI-UCB and CTRL-UCB. LSVI-UCB is implemented following the algorithm description in (Jin et al., 2020). CTRL-UCB uses CTRL's representation learning algorithm to solve linear MDP problems.[4] We note that these algorithms are non-robust, demonstrating how our theory applies to vanilla algorithms when a problem is characterized as intrinsically robust.

### D.4. Result Analysis

Similar to section 6.4, our plots demonstrate significant differences in the behavior of reward perturbations against our victim algorithms. For attackable/vulnerable environments (figures 4 and 2), we observe significant downward curvature in the cumulative perturbation plots in later episodes, subsequent reduction of perturbations. This follows our theoretical prediction on the sublinear nature of the cumulative perturbations. Furthermore, we can observe this flattening behavior in the per-episode perturbation plots as later episodes required less perturbations.

Our hypothesis that a later decrease in perturbation strength is connected to convergence towards the adversarial policy is further strengthen by our policy difference plots. We can appreciate in the plots how the learning algorithm's policy becomes closer (L2 distance layer-wise) to the adversarial target policy (the adversary's objective).[5] All in all, environments characterized as attackable required lower perturbations in later episodes as compared to earlier episodes. The adversary achieves its goal as signaled by the convergence in the policy difference plots.

For environments deemed intrinsically robust by our characterization (figures 5 and 3), we observe no downward curvature in the cumulative perturbation plots. As signaled in the per-episode plots, this difference in cumulative perturbation behavior arises from the consistently high perturbations used in later episodes. The policy difference plots further demonstrate how the adversary is unable to reach its goal: the plot differences remain large and volatile. Thus, our experiments demonstrate the inability of an adversary to efficiently (as defined in section 4) perturb the rewards of an MDP, characterized as robust, to convince an unsuspecting learning algorithm that the adversary's target policy $\pi^\dagger$ is optimal.

### D.5. Examining Accuracy of CQP Characterization

A natural question that arises when using linear MDP representations is what impact, if any, do these approximations have on the characterization's accuracy to predict attackability/robustness. Due to the stochasticity of CTRL's algorithm and environment continuity, the linear MDP representations may deviate from the optimal linear MDP representation for a given environment. This deviation can affect the accuracy of our characterization if the estimates are not good enough. We

---

[2]These details are highly dependent on the problem. Thus, we leave the hyperparameter choices in the README of our repository.

[3]Each environment has its own horizon limits and speed of learning.

[4]CTRL-UCB is also presented in the main CTRL paper (Zhang et al., 2022)

[5]We assume similar NN weights correlate with similar output distributions. We avoid using KL-divergence as metric due to its computational expense.

demonstrate in table 6 results on the accuracy of the characterization's predictions based on two indicators: sublinearity and closeness to the adversary's target policy.

### D.6. Experimental Setup

Similarly to our previous experiments, we use CTRL to learn a linear MDP representation for some given Mu-JoCo/Gymnasium environment. Using this representation, we solve our characterization to obtain the gap $\epsilon^*$ to characterize it. We re-run CTRL until we obtain two linear MDPs, one characterized as robust while the other is attackable. Each linear MDP, along with its characterization, will be used as a label to compare later.

Next, we carry out an adversarial attack using our white box algorithm. We record the perturbations used to attack at each timestep, as well as whether the learned policy by the learning victim algorithm exhibits the goal objectives delineated for each adversarial target (see 6.3). We opted to directly test for the policy behaviors instead of using a policy similarity measure because of the volatility of the policy difference, and the fact that the learning algorithms will only approximate $\pi^\dagger$ even when the attack is successful.

In deciding whether an original environment has been successfully attacked, we use two metrics to decide the success of the attack. The first metric (**M1**) aims to decide whether the cumulative perturbations follow a sublinear relationship. This is accomplished by finding the best-fit line $g(t) = a \cdot t + b$ using the cumulative scores obtained for the first $m = 500$ timesteps. For the next $T - m$ timesteps, the metric follows the equation $M_1 = \text{sign}\left(\sum_{t=m}^{T} g(t) - C_t\right)$ to decide whether the cumulative perturbations follow (+) or not (−) a sublinear growth.

The second metric (**M2**) looks more directly into the behaviour of the learned victim policy after $T = 20,000$ timesteps. We define attack success based on the adversarial policy behavior presented for each environment in section 6.3. Specifically, we observe whether the goal behavior is observable for a majority of the time. This is possible thanks to the specificity of the behaviour used for each adversarial policy. Below we outline our "signatures of attack success" used for each environment:

- FrozenLake: Agent does not reach the gift at (7,7), the original objective.

- Half-Cheetah: Agent's average speed over $T$ is negative (running backwards.)

- MountainCart: Agent's average location is negative (trying to reach the left hill rather than the right one.) We let the starting location be 0.0.

- Pendulum: Agent's average angle is $90° \pm 5°$.

It is important to note that our metrics come with their own limitations. This is primarily due to the complex behavior that the adversary seeks to teach the victim policies as well as determining whether the attack was successful or not. In our case, we focus on $M1$ and $M2$ to test our theoretical predictions dealing with the sublinearity of an adversary's budget in characterized environments as the victim becomes convinced of $\pi^\dagger$'s optimality.

### D.7. Results and Analysis

| Characterization | Attackable | | Robust | |
|---|---|---|---|---|
| **Env.\Metric** | Sub-lin. ($M1$) | Adv.'s Goal ($M2$) | Sub-lin. ($M1$) | Adv.'s Goal ($M2$) |
| FrozenLake | 100% | 98% | 100% | 93% |
| Half-Cheetah | 100% | 100% | 94% | 91% |
| MountainCart | 100% | 98% | 92% | 88% |
| Pendulum | 100% | 97% | 95% | 93% |

*Table 6.* Percentage of environments where the selected metric ($M1$ or $M2$) matched the characterization prediction (attackable or robust.) We ran for $T = 20,000$ episodes on 200 environments evenly split based on their attackable or robust characterization of CTRL's learned linear MDP according to our theory.

As shown in Table 6, the characterization of the learned representation by CTRL does appear to capture the adversarial vulnerability or robustness of the underlying environments. For the attackable environments, the cumulative perturbations "curve down" by remaining below the linear prediction (M1) made after 500 episodes. In terms of the adversary's goal (M2),

we observe a very high number of policies that follow the behavior of $\pi^\dagger$ as described previously (see "signatures of attack success.")

For the case of robust environments, the sublinearity metric (M1) reports high accuracy in following the predicted behavior of perturbations under intrinsically robust environments. While not as high as the attackable case, the robust cumulative perturbations mostly remained above the predicted line. This means that the adversary kept injecting high perturbations that were comparable or surpassed those of earlier episodes. In the case of the adversary's goal metric (M2), we also observe that most policies learned by the victim did not follow the behavior of the adversary's target policy. These results indicate that for environments characterized as robust, the adversary was unable to convince the victim algorithm of $\pi^\dagger$'s optimality.

### D.8. Sensitivity of CQP-based Attackability Test Amid Uncertainty

We are also interested in investigating the sensitivity of our CQP across different sources of noise and different degrees of "certainty" when computing $\epsilon^*$ with our CQP. For this, we design an experiment to test synthetically-generated linear MDPs, which we characterized and split into three categories using $\epsilon$: clearly attackable ($\epsilon^\star > 0.1$), near-boundary ($|\epsilon^\star| \leq 0.1$), and clearly robust ($\epsilon^\star < -0.1$). The linear MDPs have structure $(S, A, d, H) = (20, 5, 10, 10)$. We also inject Gaussian noise at different levels $\sigma \in \{0.0, 0.05, 0.1, 0.2\}$ independently of $\phi$, $\theta$ and $\hat{P}$. We generate linear MDPs until we have 100 instances of clearly attackable, clearly robust, or near-boundary; we record $\epsilon^*$ for each run.

### D.9. Results and Analysis

| Noise/Group Accuracy | Clearly Attackable | Near-boundary | Clearly Robust |
|---|---|---|---|
| $\sigma = 0.0$ | 100% | 98% | 100% |
| $\sigma = 0.05$ | 100% | 94% | 98% |
| $\sigma = 0.1$ | 99% | 73% | 99% |
| $\sigma = 0.2$ | 98% | 60% | 100% |

*Table 7.* CQP sensitivity results: reward parameter noise. Near-boundary is considered $|\epsilon^\star| \leq 0.1$. We record the number of flips of $\epsilon^\star$ as we vary the noise on the linear MDP's parameters. We find that reward noise can moderately affect CQP accuracy, especially in large noise scenarios.

Table 7 presents our findings for the reward noise case. We observe that our CQP is resistant to small amounts of noise on the reward parameter $\theta$. As the noise level increases, we observe a higher rate of disagreement among $\epsilon^\star$ computations. This indicates some stability at moderate reward parameter noise.

We also tested introducing noise into the transition function ($\hat{P}$) and mapping function ($\phi$), where we observed no sign flips when injecting $\hat{P}$. In contrast, introducing noise to $\phi$ can cause a high sign-flip rate even at small noise levels due to its direct influence in the CQP constraints. We only present the reward results in table 7 for this reason.

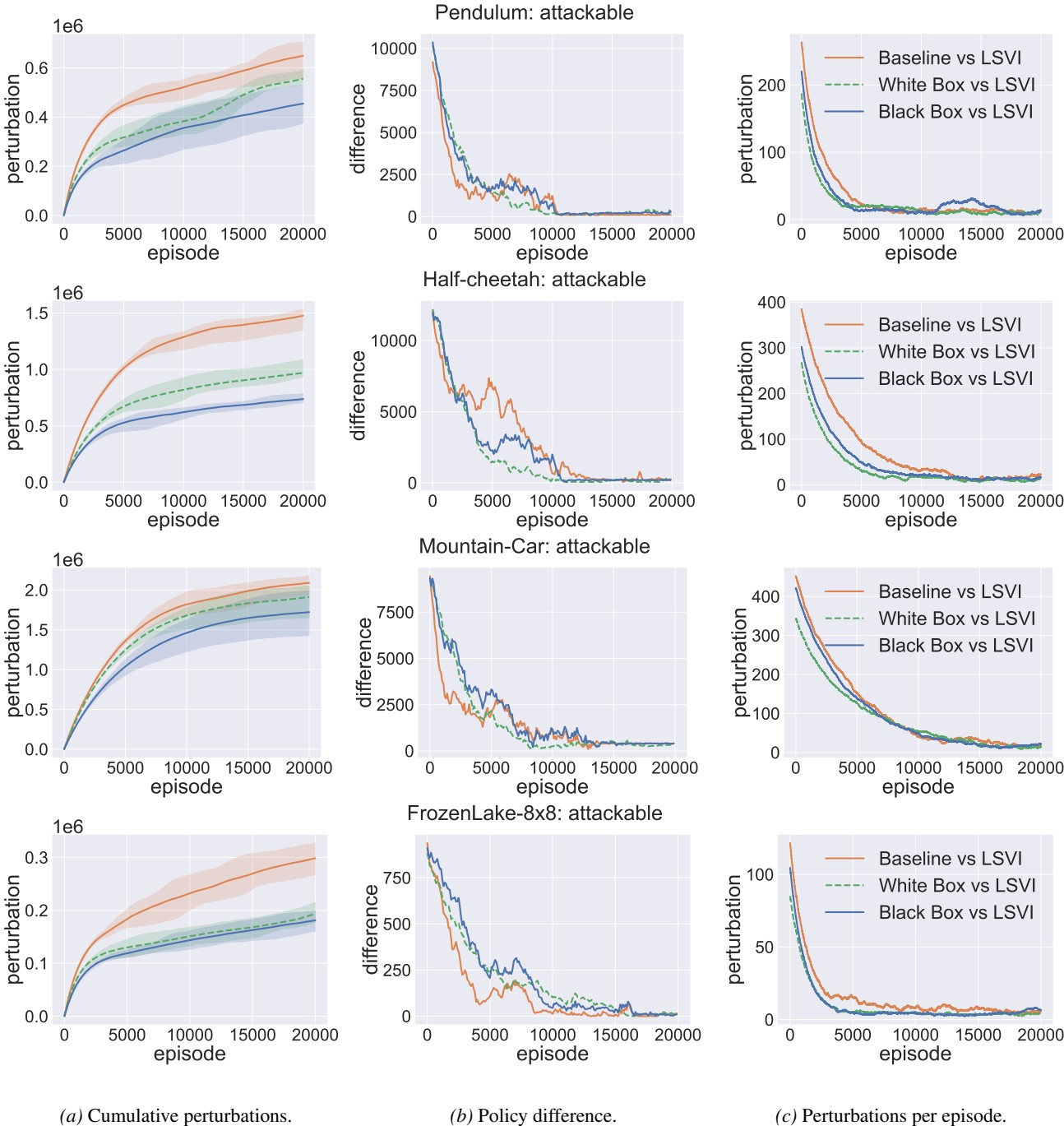

*(a)* Cumulative perturbations.  *(b)* Policy difference.  *(c)* Perturbations per episode.

*Figure 2.* Results from experiments (see section 6) with LSVI-UCB interacting with environments deemed intrinsically attackable according to equation 1. From left to right column-wise, we present the cumulative perturbation (left), policy difference (center), and perturbations per episode (right).

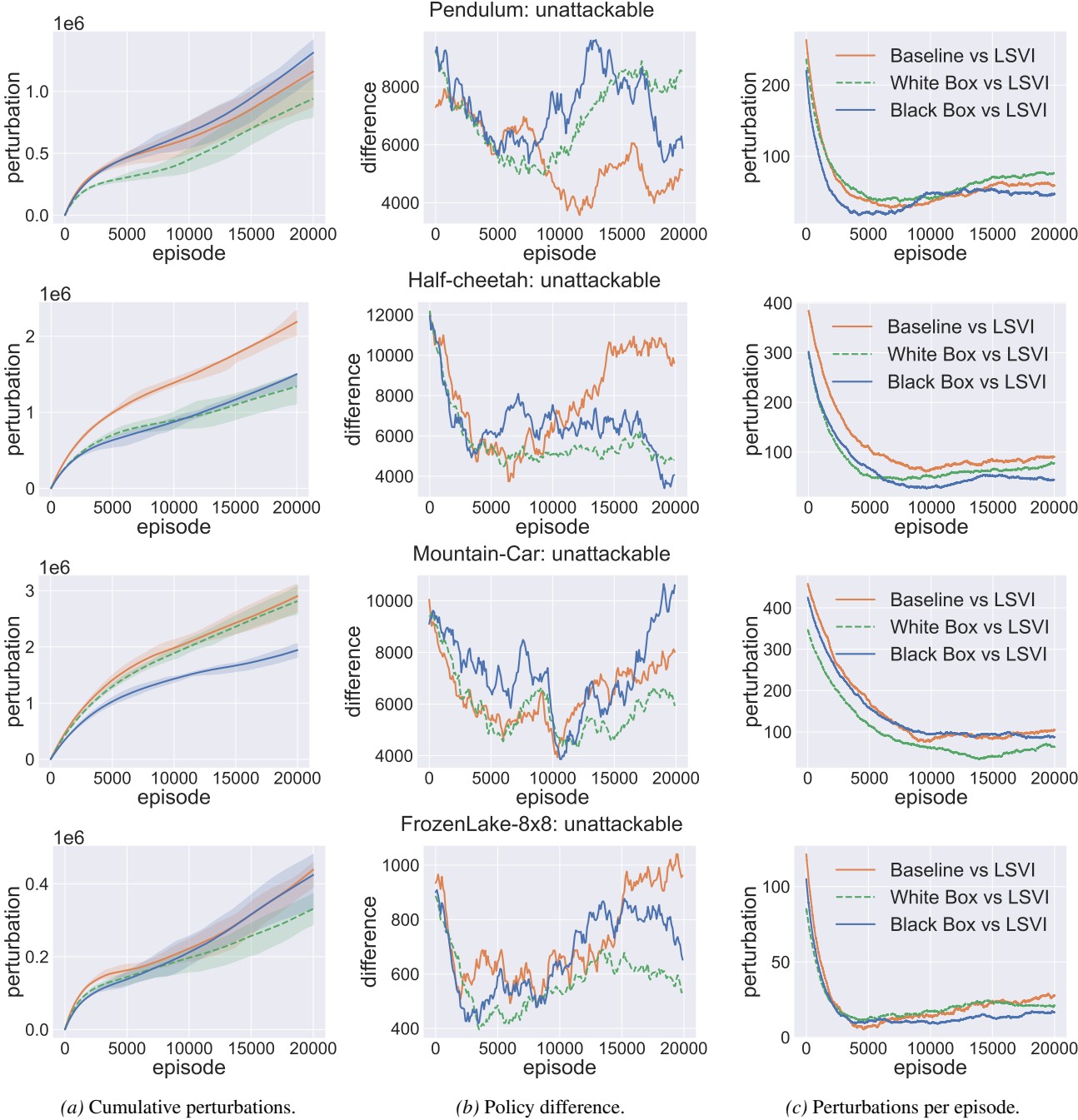

*(a)* Cumulative perturbations.   *(b)* Policy difference.   *(c)* Perturbations per episode.

*Figure 3.* Results from experiments (see section 6) with LSVI-UCB interacting with environments deemed intrinsically robust according to equation 1. From left to right column-wise, we present the cumulative perturbation (left), policy difference (center), and perturbations per episode (right).

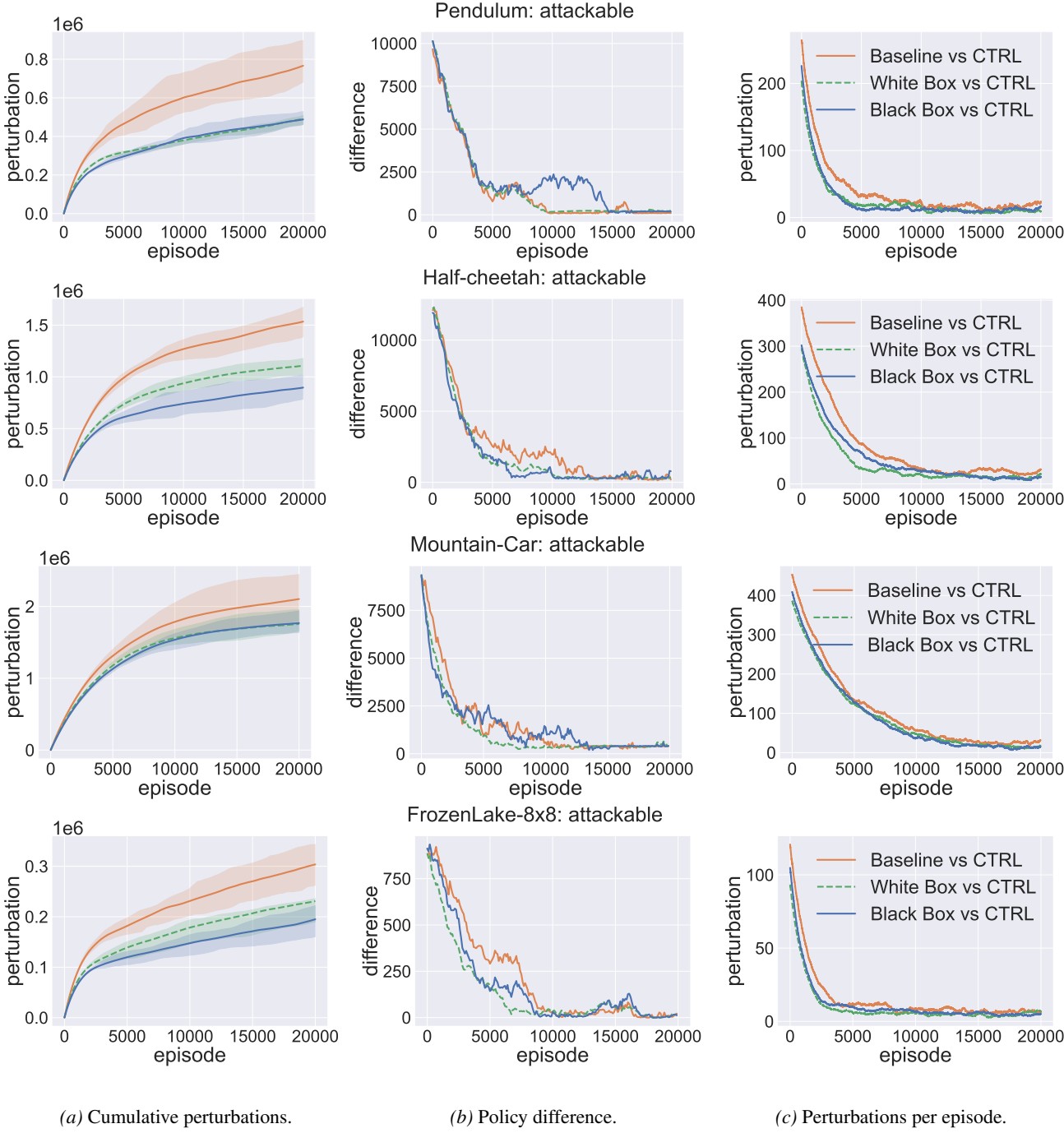

*(a)* Cumulative perturbations.          *(b)* Policy difference.          *(c)* Perturbations per episode.

*Figure 4.* Extended results depicting all *intrinsically attackable* environments poisoned against CTRL as the victim algorithm. From left to right columns, we present the cumulative perturbations (**left**), the policy difference (**center**), and the perturbations per episode (**right**) with CTRL interacting with environments deemed intrinsically attackable according to equation 1.

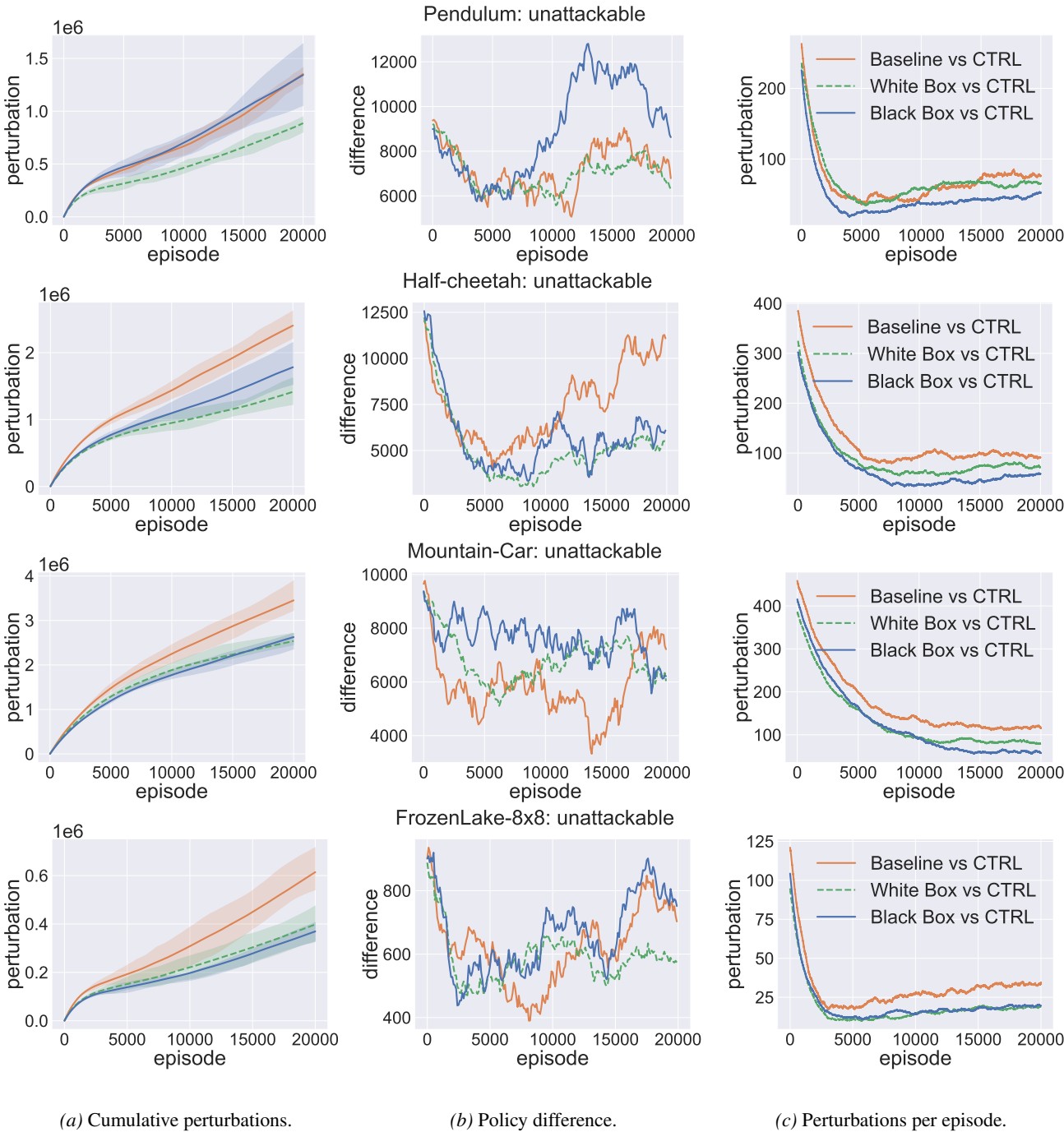

*(a)* Cumulative perturbations.  *(b)* Policy difference.  *(c)* Perturbations per episode.

*Figure 5.* Extended results depicting all environments *intrinsically robust* poisoned against CTRL-UCB as the victim algorithm. From left to right columns, we present the cumulative perturbations (**left**), the policy difference (**center**), and the perturbations per episode (**right**) with CTRL interacting with environments deemed intrinsically robust according to equation 1.

# E. Extended Proof

## E.1. Proof of Remark 4.4

**Theorem E.1.** *If $\Delta > \Delta_3$ as defined in Zhang et al. (2020), then there exists a feasible attack policy with*

$$\epsilon^* = \frac{2}{1+\gamma}(\Delta - \Delta_3) > 0.$$

*Proof.* Under reward poisoning, the Q-learning update becomes

$$Q_{t+1}(s,a) = (1 - \alpha_t) Q_t(s,a) + \alpha_t \big[r_t + \delta_t + \gamma \max_{a'} Q_t(s',a')\big],$$

where the perturbation satisfies $|\delta_t| \leq \Delta$ for all $t$. By the boundedness result in Zhang et al. (2020), for large $t$ we have $|Q_t(s,a) - Q^*(s,a)| \leq \Delta/(1-\gamma)$.

To guarantee that the target action $\pi^\dagger(s)$ dominates any other action by a margin $\epsilon$, we require for each $s \in S^\dagger$ and $a \neq \pi^\dagger(s)$:

$$Q_{\{\theta_h^\dagger\}}(s, \pi^\dagger(s)) \geq \epsilon + Q_{\{\theta_h^\dagger\}}(s,a).$$

Using the above bound on deviations, this condition becomes

$$Q^*(s, \pi^\dagger(s)) + \frac{\Delta}{1-\gamma} \geq \epsilon + Q^*(s,a) - \frac{\Delta}{1-\gamma}.$$

Thus

$$\epsilon \leq \frac{2\Delta}{1-\gamma} - \big(Q^*(s,a) - Q^*(s, \pi^\dagger(s))\big).$$

Taking the worst-case gap and substituting the definition

$$\Delta_3 = \frac{1+\gamma}{2} \max_{s \in S^\dagger} \max_{a \neq \pi^\dagger(s)} \big[Q^*(s,a) - Q^*(s, \pi^\dagger(s))\big]_+,$$

we can obtain that

$$\epsilon \leq \frac{2\Delta}{1-\gamma} - \frac{2\Delta_3}{1+\gamma}.$$

Then

$$\epsilon^* = \frac{2}{1+\gamma}(\Delta - \Delta_3) > 0.$$

Therefore, whenever $\Delta > \Delta_3$, a positive margin $\epsilon^*$ exists, completing the proof. $\qquad\square$

## E.2. Proof of the No-Regret Guarantee for the Chosen LSVI-UCB Algorithm

**Lemma E.2.** *Let $\pi$ be any policy and $h \in [H]$. Write $Q_h^\pi(s,a) := r_h(s,a) + \mathbb{E}\big[\sum_{h'=h+1}^{H} r_{h'}(s_{h'}, \pi_{h'}(s_{h'})) \mid s_h = s, a_h = a, \pi\big]$ and $V_h^\pi(s) := \max_{a \in \mathbb{A}} Q_h^\pi(s,a)$; denote $V_h^*(s) := \max_\pi V_h^\pi(s)$.*

*Fix an episode $t \geq 1$. Set $\Lambda_h^t := \lambda_t I_d + \sum_{\tau < t} \phi_h^\tau \phi_h^{\tau\top}$ with $\lambda_t := 4HS\sqrt{d\,t}$ and define*

$$w_h^t := (\Lambda_h^t)^{-1} \sum_{\tau < t} \phi_h^\tau \big[r_h^\tau + V_{h+1}^t(s_{h+1}^\tau)\big], \qquad S_{h,t} := \sum_{\tau < t} \phi_h^\tau \eta_{h,\tau},$$

*where each noise term $\eta_{h,\tau} = r_h^\tau - \langle \phi_h^\tau, \theta_h \rangle$ is 1-sub-Gaussian. Consider the event*

$$\mathcal{E} = \Big\{ \|(\Lambda_h^t)^{-1/2} S_{h,t}\|_2 \leq \sqrt{2\log\big(\det(\Lambda_h^t)^{1/2}/\det(\lambda_t I_d)^{1/2}\delta\big)} \text{ for all } h, t \Big\},$$

*which holds with probability at least $1 - \delta$ after a union bound over $h$ and $t$.*

*For any $(s,a)$ and any policy $\pi$, on $\mathcal{E}$*

$$\langle \phi_h(s,a), w_h^t \rangle - Q_h^\pi(s,a) = P_h\big(V_{h+1}^t - V_{h+1}^\pi\big)(s,a) + \epsilon_{h,t}(s,a),$$

*where $\epsilon_{h,t} := \varepsilon_{h,t}^{ridge} + \varepsilon_{h,t}^{noise}$ with*

$$\varepsilon_{h,t}^{ridge}(s,a) := -\lambda_t\, \phi_h(s,a)^\top (\Lambda_h^t)^{-1} w_{h,g}, \qquad \varepsilon_{h,t}^{noise}(s,a) := \phi_h(s,a)^\top (\Lambda_h^t)^{-1} S_{h,t},$$

*and $w_{h,g}$ is the coefficient satisfying $P_h(V_{h+1}^t - V_{h+1}^\pi)(s,a) = \langle \phi_h(s,a), w_{h,g} \rangle$. Assuming $\|w_{h,g}\|_2 \le M$ (e.g. $M = \frac{1}{2S}$ in equation 16) and recalling $(\Lambda_h^t)^{-1} \preceq \lambda_t^{-1} I_d$, the two pieces satisfy*

$$|\varepsilon_{h,t}^{ridge}(s,a)| \le \sqrt{\lambda_t}\, M \|\phi_h(s,a)\|_{(\Lambda_h^t)^{-1}}, \quad |\varepsilon_{h,t}^{noise}(s,a)| \le \sqrt{2 \log \frac{\det(\Lambda_h^t)}{\det(\lambda_t I_d)\delta}}\, \|\phi_h(s,a)\|_{(\Lambda_h^t)^{-1}}.$$

*Hence, if we write*

$$\beta_t := c_0 dH\Big(\sqrt{\log \frac{\det(\Lambda_h^t)}{\det(\lambda_t I_d)}} + \sqrt{2\log\tfrac{1}{\delta}}\Big) + \frac{\sqrt{\lambda_t}}{2S}, \tag{84}$$

*where $c_0$ is a constant defined in (Jin et al., 2020, Theorem 3.1).*

*then for all $(s,a)$ $|\epsilon_{h,t}(s,a)| \le \beta_t \|\phi_h(s,a)\|_{(\Lambda_h^t)^{-1}}$.*

*Proof.* Under linear realizability (Jin et al., 2020, Prop. 2.3) every state–action value can be written as $Q_h^\pi(s,a) = \langle \phi_h(s,a), w_h^\pi \rangle$. Fix an episode $t \ge 1$ and abbreviate $g(s') := V_{h+1}^t(s') - V_{h+1}^\pi(s')$. By definition of the linear transition model there exists a vector $w_{h,g}$ with $P_h g(s,a) = \langle \phi_h(s,a), w_{h,g} \rangle$.

Set $X_h^t := [\phi_h^0, \ldots, \phi_h^{t-1}]^\top$ and $y_h^t := [r_h^0 + V_{h+1}^t(s_{h+1}^0), \ldots, r_h^{t-1} + V_{h+1}^t(s_{h+1}^{t-1})]^\top$. The ridge estimator is $w_h^t = (\Lambda_h^t)^{-1} X_h^{t\top} y_h^t$ with $\Lambda_h^t = X_h^{t\top} X_h^t + \lambda_t I_d$. Using $y_h^t - X_h^t w_h^\pi = g + \eta$, where $\eta_{h,\tau}$ is the centred 1-sub-Gaussian noise, we compute

$$\langle \phi_h(s,a), w_h^t - w_h^\pi \rangle = \phi_h(s,a)^\top (\Lambda_h^t)^{-1} X_h^{t\top} (g + \eta).$$

Because $X_h^{t\top} g = (X_h^{t\top} X_h^t) w_{h,g}$, this expression equals

$$P_h g(s,a) - \lambda_t\, \phi_h(s,a)^\top (\Lambda_h^t)^{-1} w_{h,g} + \phi_h(s,a)^\top (\Lambda_h^t)^{-1} S_{h,t},$$

with $S_{h,t} := \sum_{\tau < t} \phi_h^\tau \eta_{h,\tau}$. We identify the ridge-bias term $\varepsilon_{h,t}^{ridge} = -\lambda_t \phi_h^\top (\Lambda_h^t)^{-1} w_{h,g}$ and the stochastic term $\varepsilon_{h,t}^{noise} = \phi_h^\top (\Lambda_h^t)^{-1} S_{h,t}$. We have

$$|\varepsilon_{h,t}^{ridge}| \le \lambda_t \|\phi_h(s,a)\|_{(\Lambda_h^t)^{-1}} \|w_{h,g}\|_{(\Lambda_h^t)^{-1}} \le \sqrt{\lambda_t}\, M \|\phi_h(s,a)\|_{(\Lambda_h^t)^{-1}},$$

where the second inequality uses $(\Lambda_h^t)^{-1} \preceq \lambda_t^{-1} I_d$ and the norm bound $\|w_{h,g}\|_2 \le M$. On the high-probability event $\mathcal{E}$ the noise term satisfies

$$|\varepsilon_{h,t}^{noise}| \le \|\phi_h(s,a)\|_{(\Lambda_h^t)^{-1}} \|S_{h,t}\|_{(\Lambda_h^t)^{-1}} \le \sqrt{2 \log \frac{\det(\Lambda_h^t)}{\det(\lambda_t I_d)\delta}}\, \|\phi_h(s,a)\|_{(\Lambda_h^t)^{-1}}.$$

yields $|\varepsilon_{h,t}^{ridge} + \varepsilon_{h,t}^{noise}| \le \beta_t \|\phi_h(s,a)\|_{(\Lambda_h^t)^{-1}}$. Therefore

$$\langle \phi_h(s,a), w_h^t \rangle - Q_h^\pi(s,a) = P_h\big(V_{h+1}^t - V_{h+1}^\pi\big)(s,a) + \epsilon_{h,t}(s,a), \qquad |\epsilon_{h,t}(s,a)| \le \beta_t \|\phi_h(s,a)\|_{(\Lambda_h^t)^{-1}},$$

. $\square$

**Proposition E.3** (Sub-linear Regret of LSVI-UCB with Time-Growing Regularization). *Under definition 3.1, the algorithm sets the regularization parameter $\lambda_t = 4HS\sqrt{d\,t}$ and forms the design matrix $\Lambda_h^t = \lambda_t I_d + \sum_{\tau < t} \phi_h^\tau \phi_h^{\tau\top}$ at the beginning of episode $t$. Writing $w_h^t$ for the regularised least–squares estimate, define the exploration bonus $\beta_t$ as equation 84.*

*The optimistic action–value estimate is $\widehat{Q}_h^t(s,a) = \langle \phi_h(s,a), w_h^t \rangle + \beta_t \|\phi_h(s,a)\|_{(\Lambda_h^t)^{-1}}$ and the policy $\pi_t$ executes $a_h^t = \arg\max_a \widehat{Q}_h^t(s_h^t, a)$. Denote $V_h^*(s) = \max_a Q_h^*(s,a)$ and let the (pseudo) cumulative regret be $R_T = \sum_{t=1}^T \big[V_1^*(s_1^t) - V_1^{\pi_t}(s_1^t)\big]$. Then, on the event $\mathcal{E}$ of Lemma E.2 (which holds with probability at least $1 - \delta$),*

$$R_T = \widetilde{O}\big(dH^{3/2}T^{3/4}\big).$$

*If the attacker pays a cost of at least $\varepsilon^*$ whenever the learner deviates from the target policy, the total cost is $\widetilde{O}\big(dH^{3/2}T^{3/4}/\varepsilon^*\big)$, which is also sub–linear in $T$.*

*Proof.* For each $(h, t)$, $\wedge_{h,t} = \sum_{\tau < t} \phi_h^\tau \eta_{h,\tau}$, where $\eta_{h,\tau}$ denotes the 1-sub-Gaussian reward noise ($r_h(s, a) = \langle \phi(s, a), \theta_h \rangle + \eta_{h,\tau}$).

Theorem 1 of Abbasi-Yadkori et al. (2011) gives the event

$$\mathcal{E} = \left\{ \| (\Lambda_h^t)^{-1/2} \wedge_{h,t} \|_2 \leq \sqrt{2 \log \frac{\det(\Lambda_h^t)^{1/2}}{\det(\lambda_t I_d)^{1/2} \delta}} \; \forall h, t \right\}$$

with $\mathbb{P}(\mathcal{E}) \geq 1 - \delta$. On $\mathcal{E}$, we have that $\|w_h^t - w_h^\star\|_{\Lambda_h^t} \leq \beta_t$, from Theorem 2 of Abbasi-Yadkori et al. (2011)

To claim optimistic $Q$-estimation $\hat{Q}_h^t(s, a) \geq Q_h^{\pi^*}(s, a)$ for all $(s, a, h)$, we proceed by backward induction on $h$.

For $h = H$, since $V_{H+1}^t = V_{H+1}^{\pi^*} = 0$, Lemma E.2 with $\pi = \pi^*$ gives

$$Q_H^{\pi^*}(s, a) \leq \langle \phi_H(s, a), w_H^t \rangle + \beta_t \|\phi_H(s, a)\|_{(\Lambda_H^t)^{-1}} = \hat{Q}_H^t(s, a).$$

Assume $\hat{Q}_{h+1}^t \geq Q_{h+1}^{\pi^*}$. Then $V_{h+1}^t \geq V_{h+1}^{\pi^*}$, so $P_h(V_{h+1}^t - V_{h+1}^{\pi^*}) \geq 0$. Applying Lemma E.2 again,

$$Q_h^{\pi^*}(s, a) \leq \langle \phi_h(s, a), w_h^t \rangle + \beta_t \|\phi_h(s, a)\|_{(\Lambda_h^t)^{-1}} = \hat{Q}_h^t(s, a).$$

Thus the property holds for $h$ whenever it holds for $h + 1$. By backward induction it holds for all layers.

Define for each episode $t$ and layer $h$ the gap

$$\delta_h^t := V_h^t(s_h^t) - V_h^{\pi_t}(s_h^t), \quad \delta_{H+1}^t := 0.$$

Set $\sigma_{h,t} := \|\phi_h^t\|_{(\Lambda_h^t)^{-1}}$, by the optimism $\hat{Q}_h^t \geq Q_h^{\pi_t}$ one shows exactly as in (Jin et al., 2020, Lemma B.6) that

$$\delta_h^t \leq \delta_{h+1}^t + \zeta_{h+1}^t + 2 \beta_t \sigma_{h,t}, \quad \zeta_{h+1}^t := P_h V_{h+1}^{\pi_t}(s_h^t, a_h^t) - V_{h+1}^{\pi_t}(s_{h+1}^t) \text{ is a martingale difference.}$$

Summing this inequality over $h = 1, \ldots, H$ gives

$$\delta_1^t = V_1^t(s_1^t) - V_1^{\pi_t}(s_1^t) \leq \sum_{h=1}^H \zeta_h^t + 2 \beta_t \sum_{h=1}^H \sigma_{h,t}.$$

Summing over episodes gives

$$R_T = \sum_{t=1}^T \delta_1^t \leq \sum_{t=1}^T \sum_{h=1}^H \zeta_h^t + 2 \sum_{t=1}^T \beta_t \sum_{h=1}^H \sigma_{h,t}.$$

The first double-sum is a martingale over $HT$ bounded differences $\zeta_h^t \in [-H, H]$, so by Azuma–Hoeffding with probability at least $1 - \delta$

$$\sum_{t=1}^T \sum_{h=1}^H \zeta_h^t = O\big(H \sqrt{T \log(1/\delta)}\big).$$

Then we have

$$\sum_{t=1}^T \beta_t \sum_{h=1}^H \sigma_{h,t} \leq \max_t(\beta_t) \cdot \sum_{t=1}^T \sum_{h=1}^H \sigma_{h,t}$$

Define $\Lambda_h^t = \lambda_t I_d + \sum_{\tau \leq t} \phi_h^\tau \phi_h^{\tau \top}$, with an auxiliary sequence with constant ridge term: $\bar{\Lambda}_h^t := \lambda_1 I_d + \sum_{\tau \leq t} \phi_h^\tau \phi_h^{\tau \top}$. Because $\lambda_t \geq \lambda_1$ for every $t$, $\Lambda_h^t \succeq \bar{\Lambda}_h^t$, we have $(\Lambda_h^t)^{-1} \preceq (\bar{\Lambda}_h^t)^{-1}$.

Hence

$$\sigma_{h,t} \leq \bar{\sigma}_{h,t} := \|\phi_h^t\|_{(\bar{\Lambda}_h^t)^{-1}}.$$

Since $\|\phi_h\|_2 \leq 1$, Lemma 11 in Abbasi-Yadkori et al. (2011) gives

$$\sum_{t=1}^T \sum_{h=1}^H \bar{\sigma}_{h,t}^2 \leq 2H \log \frac{\det(\bar{\Lambda}_h^T)}{\det(\lambda_1 I_d)} \leq 2H \log \frac{(\lambda_1 + T)^d}{\lambda_1^d} \leq 2dH \log(1 + T/\lambda_1) \leq 2dH \log(1 + T), \quad (85)$$

and therefore $\sum_{t=1}^{T} \sum_{h=1}^{H} \sigma_{h,t}^2 \leq 2dH \log(1+T)$. Then

$$\sum_{t=1}^{T} \sum_{h=1}^{H} \sigma_{h,t} \leq \sqrt{HT} \left( \sum_{t=1}^{T} \sum_{h=1}^{H} \sigma_{h,t}^2 \right)^{1/2} = O\big(\sqrt{d}\, H \sqrt{T \log(1+T)}\big).$$

Altogether,

$$R_T = O\big(H \sqrt{T \log(1/\delta)}\big) + 2 \cdot \widetilde{O}\big((Hd\, T^{1/2})^{1/2}\big) \cdot O\big(\sqrt{d}\, H \sqrt{T \log(1+T)}\big) = \widetilde{O}\big(dH^{3/2} T^{3/4}\big).$$

Dividing by $\varepsilon^*$ then gives the attack-cost bound $\widetilde{O}\big(dH^{3/2} T^{3/4} / \varepsilon^*\big)$.

$\square$

