# OpenReview forum: "When Can You Poison Rewards? A Tight Characterization of Reward Poisoning in Linear MDPs"
_ICML.cc/2026/Conference — ICML 2026 regular_

### Official Review · Reviewer_GQAB · 2026-02-26

**Soundness:** 3
**Presentation:** 3
**Significance:** 3
**Originality:** 3
**Overall Recommendation:** 4
**Confidence:** 1

**Summary:**

This paper explores how naturally resistant Reinforcement Learning (RL) systems are to "reward poisoning" (where someone hacks the scoring system to trick the AI). Specifically, it tries to figure out exactly when hacking the system is just impossible.

**Compliance With Llm Reviewing Policy:**

Affirmed.

**Final Justification:**

The rebuttal have addressed my concern and I will maintain my positive score.

**Key Questions For Authors:**

See weakness

**Limitations:**

yes

**Strengths And Weaknesses:**

**Strength**

This paper finds a really clever way to check if a specific RL environment is hackable. By boiling the problem down to a single math equation (a CQP), they took a security concept that only worked for simple AI and successfully applied it to complex RL.

**Weakness**
The proof explaining why $\epsilon^* \le 0$ makes the system "unhackable" is sloppy; it reads like an unfinished rough draft in the main text. Also, their math rules completely ignore the weird, off-path situations that AI agents actually stumble into all the time while learning. On top of that, the "closed-book" attack leans on a highly unrealistic, non-standard assumption about predicting the future ($P_h$).

---

> ### Author Rebuttal · Authors · 2026-03-31
>
> We thank the reviewer for the direct comments on proof presentation and scope.
>
> > "The proof explaining why $\\epsilon^\\star \\leq 0$ makes the system 'unhackable' is sloppy..."
>
> The main text intentionally gives only a proof sketch after Theorem 4.2; the full proof is in the appendix. The issue is expository rather than conceptual: the sketch should be easier to follow and more tightly aligned with the theorem statement.
>
> > "Their math rules completely ignore the weird, off-path situations..."
>
> Our CQP characterization is stated on the target occupancy support (states with positive occupancy under $\pi^\dagger$) rather than as a pointwise guarantee over all off-support states. The claim is not that the CQP directly controls every off-path state the learner may transiently visit; it identifies the part of the state space on which the target certificate is valid. This support-local scope drives the shared-occupancy-support requirement in Remark 4.3 and the support-restricted sufficient certificates in Remark 5.2.
>
> > "The 'closed-book' attack leans on a highly unrealistic, non-standard assumption about predicting the future ($P_h$)."
>
> We appreciate this concern. We agree that the current wording may make the assumption sound stronger than intended. In our black-box construction, The kind of estimate we have is a short pre-Stage-1 model-estimation step. For example, an offline fitted model, brief model-learning pre-processing, or, in our empirical pipeline, estimation on top of a CTRL-learned linear representation. The black-box result is therefore best read as an approximation-based extension, with $\hat P_h$ serving a limited synthesis role rather than a general future-prediction oracle.

---

> > ### Author Rebuttal · Reviewer_GQAB · 2026-04-02
> >
> > Thanks the authors for the rebuttal. I will keep my positive score.

---

> > > ### Author Response · Authors · 2026-04-02
> > >
> > > Thank you for your thoughtful feedback and we sincerely appreciate your positive assessment.

---

### Official Review · Reviewer_7Bku · 2026-02-28

**Soundness:** 3
**Presentation:** 3
**Significance:** 3
**Originality:** 3
**Overall Recommendation:** 5
**Confidence:** 4

**Summary:**

This paper studies reward poisoning in reinforcement learning, where an adversary perturbs rewards under a limited budget to steer the learner toward a specified target policy. Focusing on finite-horizon linear MDPs, the authors provide an instance-level characterization of attackability versus intrinsic robustness: for a given target policy, a sublinear-cost poisoning attack is possible if and only if a corresponding convex quadratic program (CQP) achieves a strictly positive margin ε*>0. This establishes a sharp boundary between environments that are inherently vulnerable and those that are intrinsically robust under a pointwise (state–action–stage) optimality notion, meaning that any successful attack in the latter case requires cost linear in the horizon. Based on this characterization, the paper develops both white-box and black-box attack procedures. The white-box method constructs budget-efficient reward shaping directly from the CQP solution, while the black-box method estimates the linear MDP parameters from data and then applies the same CQP-based test and attack with cost guarantees. Experiments on benchmark RL tasks using learned linear representations suggest that the CQP characterization can predict attack success and effectively distinguish vulnerable from robust environments.

**Compliance With Llm Reviewing Policy:**

Affirmed.

**Final Justification:**

Overall, the rebuttal resolves my concerns. I therefore increase my positive score.

**Key Questions For Authors:**

1. Do you have controlled experiments on synthetic or tabular linear MDPs that exactly satisfy your theoretical assumptions?
   In particular, can you directly verify that the CQP condition (ε*>0 vs. ε*≤0) perfectly predicts attack success or failure, and that the poisoning cost scales sublinearly versus linearly as stated in the theory? Such results would substantially strengthen the empirical support for Theorem 4.2 beyond MuJoCo with learned representations.

2. How sensitive is the CQP-based attackability test to model estimation errors in the black-box setting?
   If the estimated transition or reward parameters are slightly misspecified, can this change the sign of ε* or lead to incorrect robustness classification? A clearer robustness or sensitivity analysis would help assess the practical reliability of the method.

3. Your definition of success relies on pointwise (state–action–stage) optimality.
   How would the main conclusions change under weaker or more practical criteria, such as overall return optimality or approximate alignment with the target policy? Could some environments classified as intrinsically robust under your definition still be vulnerable under these alternative notions?

4. Will  opensources？

**Limitations:**

The paper briefly mentions positive impact (e.g., inspiring robust RL design), but it does not sufficiently discuss limitations or potential negative societal impacts.

Suggestions for improvement:

1. Dual-use risk: The work provides a principled and efficient way to characterize and construct reward poisoning attacks. This could lower the barrier for malicious actors to design more effective attacks on RL systems. The authors should explicitly acknowledge this dual-use concern and discuss responsible disclosure or mitigation perspectives.

2. Scope limitations: The results are restricted to linear MDPs (or approximate linear representations). The authors should more clearly state that the characterization may not hold in general nonlinear or large-scale deep RL settings, and that robustness conclusions may not directly transfer.

3. Practical constraints: The theoretical guarantees rely on idealized assumptions (e.g., exact linear structure, pointwise optimality, no-regret learners). The authors could better discuss how these assumptions limit applicability in real-world deployments.

**Strengths And Weaknesses:**

Soundness

The theoretical part of the paper is strong. The main result provides a clear necessary and sufficient condition for attackability in finite-horizon linear MDPs via a convex quadratic program. The arguments are logically structured, and the assumptions (linearity, bounded features, no-regret learners) are standard in the linear MDP literature. The connection to prior results in linear bandits and tabular MDPs further supports the correctness and positioning of the theory.

However, there is an important experimental gap. The empirical validation mainly relies on MuJoCo environments with learned linear representations. While this is interesting, it would be much stronger to include experiments on simple synthetic or tabular linear MDPs that directly match the theoretical setting. In particular, controlled experiments could explicitly verify: (i) whether the CQP condition ε*>0 or ε*≤0 perfectly predicts success or failure when the model assumptions hold exactly, and (ii) whether the cumulative poisoning cost indeed scales sublinearly in vulnerable instances and linearly (or superlinearly) in robust ones. Such experiments would more directly validate the core theorems rather than relying on approximate linear representations.

In the black-box setting, the guarantees depend on additional structural and estimation assumptions. The impact of model estimation error on the CQP test and on the success of the attack is not fully analyzed, which leaves some uncertainty about robustness of the conclusions in practical scenarios.

Presentation

Overall, the paper is clearly written and well structured. The progression from theoretical characterization to white-box and black-box attacks, and then to experiments, is logical and easy to follow. The relationship to prior work in bandits and RL poisoning is clearly discussed.

That said, some technical sections (especially around occupancy support and the CQP constraints) could benefit from more intuitive explanation or illustrative examples. In particular, the role of the shared occupancy support assumption in the extension to policy sets may be hard to grasp without a concrete toy example.

Significance

The paper addresses an important question: when reward poisoning is fundamentally possible versus impossible in RL. Instead of only proposing new attacks, it focuses on characterizing intrinsic robustness at the instance level. This perspective is conceptually valuable and could influence future work on adversarial robustness in RL, especially in understanding environment-dependent vulnerabilities.

The significance is mainly theoretical, but potentially broad. The idea that robustness can be a property of the environment geometry, rather than only of the algorithm, is an important conceptual shift. With stronger empirical validation in controlled linear MDP settings, the impact would be even clearer.

Originality

The main originality lies in providing a tight, instance-level characterization of reward poisoning in linear MDPs via a convex program. While it builds on existing work in linear bandits and RL poisoning, the extension to multi-step linear MDPs and the identification of intrinsic robustness as a geometric property are novel contributions.

The white-box and black-box attack constructions are natural consequences of the characterization rather than entirely new attack paradigms. However, the unification of characterization and attack construction within the same convex framework is a meaningful and well-motivated contribution.

---

> ### Author Rebuttal · Authors · 2026-03-31
>
> We thank the reviewer for the thoughtful questions on robustness, practicality, and interpretation.
>
> > "In the black-box setting, the guarantees depend on additional structural and estimation assumptions..."
>
> The black-box part is a conditional estimation-based extension. Theorem 5.3 is deliberately conditional, and Remark 5.2 makes clear that the black-box CQPs provide only sufficient certificates on the chosen certified support, so infeasibility under estimation does not imply a full converse. On the empirical side, Section 6 already studies approximation variability through CQP-accuracy experiments under stochastic CTRL representations. The intended claim is therefore narrower than a full sensitivity theorem for arbitrary model misspecification.
>
> > "Do you have controlled experiments..."
>
> We acknowledge the importance of running the reviewer's suggested synthetic experiments to validate our theory. As part of our response to reviewer BJGb, we have designed a preliminary experiment to observe the behavior of a principled robust algorithm under synthetically-generated linear MDPs that our CQP classifies as robust or vulnerable. We test on 200 environments with an even split between environments characterized as vulnerable and robust. Our preliminary results show that our CQP accuratelly predicts the success or failure (metric M1) of the adversary as well as the general behavior of the perturbation cost (metric M2). We will include extended synthetic experiments in our final version of our paper.
>
> > "How sensitive is the CQP-based attackability test...?"
>
> We designed and ran a new experiment to address this. We generate 30 random synthetic linear MDPs with $(S, A, d, H) = (20, 5, 10, 10)$, split evenly into three groups: clearly attackable ($\\epsilon^\\star > 0.1$), near-boundary ($|\\epsilon^\\star| \\le 0.1$), and clearly robust ($\\epsilon^\\star < -0.1$). We inject Gaussian noise at levels $\\sigma \\in [0, 0.2]$ independently into $\\phi$, $\\theta$, and $\\hat{P}$ (20 resamples per configuration) and record $\\epsilon^\\star$ sign flips.
>
> Our findings for this setting are:
>
> 1. **Transition noise ($P$):** Near-zero sign-flip rate across all groups.
> 2. **Feature noise ($\\phi$):** High sign-flip rate even at small $\\sigma$, due to $\\phi$'s direct influence on the zero-cost CQP constraint. This is mitigated theoretically by treating $\\phi$ as known, but poses a practical sim-to-real challenge.
> 3. **Reward noise ($\\theta$):** At $\\sigma \\in \\{0.0, 0.05, 0.1, 0.2\\}$:
>
> | Noise / Group Accuracy |  Clearly attackable | Near-boundary | Clearly robust |
> |:---------------|-------|-----|-----|
> | $\\sigma = 0.0$ |  100% | 98% | 100%|
> | $\\sigma =0.05$ |  100% | 94% | 98% |
> |  $\\sigma =0.1$ |  99%  | 73% | 99% |
> |  $\\sigma =0.2$ |  98%  | 60% | 100%|
>
> These results indicate that $|\\epsilon^\\star| > 0.1$ (clearly attackable or robust) is stable under moderate reward-parameter noise. We will include complete results and a discussion of practical implications in the final version.
>
> > "Your definition of success relies on pointwise (state-action-stage) optimality..."
>
> Our pointwise notion is deliberate: it aligns with prior targeted reward-poisoning work (Zhang et al., 2020) and yields a clean characterization to be analyzed. The CQP only constraints states with positive target occupancy ($d_h^{\\pi^\\dagger}(s) > 0$) which are exactly the states reached when following $\\pi^\\dagger$. A simple example: if $\\pi_1$ reaches $s_1$ and $\\pi_2$ reaches $s_2$ from $s_0$, a certificate for $\\pi_1$ covers only $s_0$ and $s_1$; it says nothing about $s_2$, since $d^{\\pi_1}(s_2)=0$. If the learner deviates to $s_2$, the $\\pi_1$-specific certificate no longer applies. This is exactly why Remark 4.3 requires shared occupancy support for the policy-set extension. We agree that weaker criteria such as return optimality or approximate alignment are more practical, and can be viewed as targeting any policy within a target set. But when candidate policies have different supports, the learner can exploit distinct occupancy measures to escape the attacked region, so a single joint CQP is no longer sufficient and one must solve individual CQPs per candidate. Relaxing the shared-support requirement is a non-trivial direction for future work.
>
> > "Will opensources?"
>
> Yes. We promise to release the code after the review process.
>
> > "The paper briefly mentions positive impact..."
>
> We thank the reviewer for these helpful suggestions. We agree that the current discussion is too brief, and that these points deserve a clearer and more balanced treatment. We view this as an important direction for future revision and follow-up work.
>
> ### References
> [1] Zhang et al. *Adaptive Reward-Poisoning Attacks against Reinforcement Learning.* ICML, 2020.

---

> > ### Author Rebuttal · Reviewer_7Bku · 2026-03-31
> >
> > Thank the authors for their response, which addressed my concerns. Nice paper!

---

> > > ### Author Response · Authors · 2026-03-31
> > >
> > > Thank you for your careful reading and professional feedback, and we sincerely appreciate your kind words about our paper.

---

### Official Review · Reviewer_BJGb · 2026-03-12

**Soundness:** 3
**Presentation:** 3
**Significance:** 2
**Originality:** 2
**Overall Recommendation:** 4
**Confidence:** 4

**Summary:**

This paper studies reward poisoning attacks in finite-horizon linear MDPs. The authors provide a necessary and sufficient condition for attackability based on a convex quadratic program (CQP), which determines whether a target policy can be enforced with sublinear poisoning cost. Building on this characterization, they develop corresponding white-box and black-box attack algorithms with theoretical guarantees. The proposed methods are further evaluated empirically on several RL benchmark tasks using learned linear representations.

**Compliance With Llm Reviewing Policy:**

Affirmed.

**Key Questions For Authors:**

1) The proposed black-box algorithm separates learning (Stage 1) from attack execution (Stage 2). This separation may introduce additional attack cost. Is it possible to design algorithms that learn and attack simultaneously, rather than separating the two phases?
2) Is it possible to derive tighter attack cost bounds for the black-box setting? Are there lower bounds for the attack cost in linear MDPs?
3) The paper suggests that some environments are inherently robust to reward poisoning. Is there a clear structural characterization that explains when this occurs?

**Limitations:**

Yes.

**Strengths And Weaknesses:**

Strengths
1) The paper provides a necessary and sufficient condition for attackability in linear MDPs through a convex quadratic program, which conceptually separates vulnerable and intrinsically robust instances.
2) Based on the characterization, the authors propose both white-box and black-box attack algorithms with theoretical guarantees.
3) The proposed framework is evaluated on several RL benchmark environments.

Weaknesses
1) The intuition behind the proposed attackability condition appears somewhat natural: under a white-box threat model, the attacker reshapes rewards so that the target policy becomes optimal in the modified environment. Similar ideas have appeared in earlier work such as Liu & Shroff (2019) for bandits, where attacks modify feedback so that a target arm becomes optimal. It would be helpful to better clarify the conceptual novelty beyond this intuition.
2) The black-box attack cost in Theorem 5.3 scales roughly as $O(T^{3/4})$. However, the paper does not provide matching lower bounds, making it unclear whether this rate is tight. In the current presentation, "tightness" seems to refer mainly to achieving $o(T)$ cost. The exact order of attack cost could be an important theoretical question.
3) The experiments compare the proposed method with only a single baseline attack and do not evaluate against defensive RL algorithms.

---

> ### Author Rebuttal · Authors · 2026-03-31
>
> We thank the reviewer for engaging directly with the conceptual and technical core of the paper.
>
> > "The intuition behind the proposed attackability condition appears somewhat natural..."
>
> We agree with the reviewer's perspective that the intuition behind the proposed attackability condition is natural and has been studied in bandit literature: Liu and Shroff (2019) study reward poisoning attacks that steer stochastic bandits toward a target arm, and Wang et al. (2022) show in linear bandits that attackability is an environment-level property with a CQP-based characterization. Our contribution extends this viewpoint to finite-horizon linear MDPs: the novelty is not merely "make the target policy optimal," but characterizing when this is or is not possible at the instance level in RL, separating attackable environments from intrinsically robust ones in an algorithm-agnostic manner.
>
> > "The black-box attack cost in Theorem 5.3 scales roughly as $O(T^{3/4})$..." / "Is it possible to derive tighter attack cost bounds for the black-box setting...?"
>
> We thank the reviewer for raising this important question. We agree that the current paper does not provide a matching lower bound for the black-box cost in Theorem 5.3. Our main tightness claim is at the level of the CQP-based attackability characterization, i.e., the sublinear-versus-linear budget separation, whereas Theorem 5.3 gives a constructive upper bound for one specific black-box procedure. We believe that sharpening this rate, and in particular obtaining matching lower bounds, would be a valuable direction for future work.
>
> > "The experiments compare the proposed method with only a single baseline attack and do not evaluate against defensive RL algorithms."
>
> We agree with the reviewer on the limitations of 1) using a single baseline attack and 2) lacking trials against defensive RL algorithms.  Regarding 1), apart from the Adaptive Target (AT) attack of Xu and Singh (2023), no other prior adversarial reward poisoning attacks target the same white-box and black-box linear-MDP setting.  Concerning 2), we designed a new experiment to observe the behavior of Exp3-DARP (Nika et al., 2023), a principled adaptive linear MDP defensive algorithm, under an optimal attack strategy according to our CQP.
>
> We randomly sample synthetic linear MDPs and adversarial targets $\pi^\dagger$ for our CQP to characterize, re-sampling until an even split of 100 to 100 vulnerable and robust environments is reached. Then, we run Exp3-DARP under reward poisoning, checking two metrics (appendix D.3.1): (M1) looks at the behavior of cummulative perturbation costs and (M2) assesses the success or failure of the adversary's objective (make $\pi^\dagger$ optimal) with respect to the victim's learned policy.
>
> We find that our CQP predicted the success and failure (M2) of the adversary against Exp3-DARP correctly in 99% and 100% of attackable and robust synthetic environment instances. Furthermore, we also observe sublinear curvatures of perturbation costs under vulnerable environments and superlinear cost when robust (M2). These results indicate the ability of our CQP to produce a reward perturbation capable of fooling Exp3-DARP.
>
> > "The proposed black-box algorithm separates learning (Stage 1) from attack execution (Stage 2)..."
>
> We thank the reviewer for this good question. Our current two-stage design is mainly for simplicity: it is easier to analyze cleanly. We do not mean to suggest that two-stage separation is optimal, and more adaptive joint designs are an interesting direction for future work.
>
>
> > "The paper suggests that some environments are inherently robust to reward poisoning..."
>
> In our paper, the exact characterization is given by the CQP itself: intrinsic robustness corresponds to infeasibility under the admissible shaping class. Intuitively, this happens when the feature geometry does not leave enough freedom to make the target strictly better while still preserving Bellman consistency and the norm constraints. So the structural characterization is present, but it is captured through CQP feasibility rather than through a simpler closed-form geometric condition.
>
> ### References
>
> [1] Liu and Shroff. *Data Poisoning Attacks on Stochastic Bandits.* ICML, 2019.
>
> [2] Wang et al. *When Are Linear Stochastic Bandits Attackable?* ICML, 2022.
>
> [3] Xu and Singh. *Black-Box Targeted Reward Poisoning Attack Against Online Deep Reinforcement Learning.* arXiv, 2023.
>
> [4] Jin et al. *Provably Efficient Reinforcement Learning with Linear Function Approximation.* COLT, 2020.
>
> [5] Nika et al. *Online Defense Strategies for Reinforcement Learning Against Adaptive Reward Poisoning* AISTATS, 2023.

---

> > ### Author Rebuttal · Reviewer_BJGb · 2026-04-04
> >
> > Thank you for the detailed response which resloved my concerns. I will keep my score.

---

> > > ### Author Response · Authors · 2026-04-04
> > >
> > > Thank you for your valuable comments, we genuinely appreciate your positive evaluation.

---

### Official Review · Reviewer_k14E · 2026-03-13

**Soundness:** 3
**Presentation:** 3
**Significance:** 3
**Originality:** 3
**Overall Recommendation:** 4
**Confidence:** 4

**Summary:**

This paper studies reward poisoning attackability in finite-horizon linear MDPs. The main contribution is a convex quadratic program (CQP) whose optimal value $ \epsilon^* $ is claimed to characterize whether a target policy $ \pi^\dagger $ can be induced by a reward-poisoning attacker with sublinear total attack cost. The paper defines attackability in an algorithm-agnostic way for no-regret learners, proves a white-box sufficiency result when $\epsilon^* >0$, gives a corresponding impossibility-style necessity argument for intrinsically robust instances, and proposes a two-stage black-box attack with sublinear-cost guarantees under additional structural assumptions. The paper also tests the framework on learned linear representations of non-linear control environments using CTRL, aiming to show that the CQP-based characterization remains predictive in practice.

**Compliance With Llm Reviewing Policy:**

Affirmed.

**Final Justification:**

Overall, the rebuttal resolves my concerns. I therefore maintain my positive score.

**Key Questions For Authors:**

The theorem is stated as "iff $ \epsilon^* > 0 $," but the necessity discussion and appendix appear to handle the strict-negative case $ \epsilon^* < 0 $. What exactly happens when $ \epsilon^* = 0 $?

The paper notes that the black-box CQPs only give sufficient certificates on the chosen certified support. Can you state more explicitly which claims are true instance-level characterizations and which are only support-restricted sufficient conditions?

In the appendix, you rerun CTRL until you obtain enough vulnerable/robust learned representations, and in the second experiment you rerun until you get one robust and one attackable representation. Why is this the right validation protocol for the claim that the characterization is predictive in practical RL tasks? Would the same conclusion hold under a non-selective sampling protocol?

Why should $L_2$ distance between policy-network weights be trusted as a proxy for policy closeness? Other metric like KL divergence could substantially strengthen the empirical case.

**Limitations:**

No. The impact statement is very brief. Maybe discuss the gap between exact linear-MDP theory and approximate learned representations, the selective nature of the empirical protocol, and the weakness of the policy-difference metric.

**Strengths And Weaknesses:**

Strengths:

The paper addresses an important security question in RL: not only how to poison rewards, but when poisoning is fundamentally possible or impossible. That framing is meaningful and is well motivated in the introduction. The main theorem is interesting: compared with earlier RL poisoning work, the most credible novelty here is the move from attack construction or threshold-style results to an instance-level characterization in finite-horizon linear MDPs via a CQP. This is also a natural and nontrivial extension of the exact attackability viewpoint previously developed for linear stochastic bandits.

The technical story is coherent. The CQP is clearly stated, its intuition is explained, and the paper makes a clean distinction between attackable instances ($ \epsilon^* > 0 $) and intrinsically robust ones. The relation to prior bandit and tabular-MDP results is discussed explicitly, and the white-box attack is conceptually aligned with the theorem.

Weaknesses:

The paper states "iff $ \epsilon^* > 0 $," but the necessity side is explicitly developed for the strict-negative case $ \epsilon^* < 0 $. I do not think this boundary issue alone is fatal, but it should be fixed or clarified. At minimum, the authors could explain whether the $ \epsilon^* = 0 $ case is impossible, degenerate, or left unresolved.

The black-box result is narrower than the high-level narrative suggests. The black-box attack assumes known features, access to a feature-space one-step predictor estimate, and additional structural assumptions, and even then the certification is support-restricted. So the black-box part is useful, but it is not a full analogue of the main white-box characterization.

The paper's main practical claim is that the CQP remains predictive after approximating non-linear environments as linear MDPs via CTRL. But the experimental protocol is selective: the appendix says the authors run CTRL until they have enough vulnerable and robust environments for each trial, and in the second experiment they rerun CTRL until they obtain one robust and one attackable learned linear MDP. That makes the evidence more like a conditional sanity check than an unbiased validation of whether the characterization genuinely predicts attackability of the original environment.

In the experiment, "policy difference" is computed as the $L_2$ distance between neural-network weight vectors, which is an indirect proxy for policy closeness.

The paper does not show that real non-linear environments themselves are intrinsically robust or vulnerable in a stable sense; it shows that some learned linear representations are labeled that way and that attacks on those representations behave consistently with the label.

---

> ### Author Rebuttal · Authors · 2026-03-31
>
> We thank the reviewer for the careful and constructive reading.
>
> > "The paper states 'if $\\epsilon^\\star > 0$,' but the necessity side is explicitly developed for the strict-negative case $\\epsilon^\\star < 0$..."
>
> $\epsilon^\star$ measures the largest domination margin inducible in favor of target policy $\pi^\dagger$ within the admissible reward-shaping class. $\epsilon^\star > 0$ is exactly the regime where $\pi^\dagger$ can be made to strictly dominate all competing actions on the relevant support; $\epsilon^\star \le 0$ means no such strict domination is possible. The $\epsilon^\star = 0$ boundary is not a separate successful regime: zero margin is still insufficient for the strictly positive pointwise Bellman separation required for the $T-o(T)$ forcing conclusion. We will clarify this boundary more explicitly.
>
>
> > "The black-box result is narrower than the high-level narrative suggests..."
>
> The intended role of the black-box part is narrower than the main characterization. This is already reflected in Remark 5.2, which states that the black-box CQPs provide only sufficient certificates on the chosen certified support, and in Theorem 5.3 and Algorithm 1, which make the narrower scope explicit through the use of known features, a feature-space one-step predictor estimate, certified sets, and additional structural assumptions. The main instance-level characterization is the CQP result in the structured linear-MDP setting, while the black-box part is a conditional estimation-based extension with support-restricted certificates.
>
> > "The paper's main practical claim is that the CQP remains predictive after approximating non-linear environments as linear MDPs via CTRL..."
>
> Our resampling protocol takes advantage of the stochastic nature of CTRL's representation learning: different hyperparameters (e.g. random seeds) can yield different linear-MDP approximations of the same environment, each with a different $\epsilon^\star$. We filter based on the CQP label before any attack is run, not based on attack success. Even so, the practical claim is best read as conditional evidence through learned linear representations rather than as an unbiased theorem-level validation for the original nonlinear environment.
>
> > "The paper does not show that real non-linear environments themselves are intrinsically robust or vulnerable in a stable sense..."
>
> This is the right interpretation. Our intended empirical claim is narrower: we study whether the CQP label on a learned linear representation is predictive of poisoning behavior on that representation. This is also consistent with Contribution 3, which only hypothesizes that attackability may be approximately preserved under learned linear representations.
>
>
> > "In the experiment, 'policy difference' is computed as the $L_2$ distance between neural-network weight vectors..." / "Why should $L_2$ distance between policy-network weights be trusted as a proxy for policy closeness...?"
>
> As the reviewer correctly points out, the $L_2$ distance between network weights is an imperfect proxy for policy closeness. We use $L_2$ in our experiments as a rough but computationally convenient indicator of policy closeness. A policy-level metric such as KL divergence or action-distribution agreement would strengthen the empirical case, and we plan to include such results in our final version of our paper.
>
> > "The impact statement is very brief..."
>
> We acknowledge this limitation, and we will expand on our limitations in our final version of the paper.

---

> > ### Author Rebuttal · Reviewer_k14E · 2026-04-02
> >
> > Thank you for the thoughtful rebuttal. The response meaningfully clarifies several of my concerns.
> >
> > I appreciate the clarification of the $\epsilon^*=0$ case and the black-box result. In addition, the rebuttal appropriately narrows the practical claim on the learned linear representation.
> >
> > At the same time, the use of $L_2$ distance between policy-network weights is still a weak proxy for policy closeness.
> >
> > Overall, the rebuttal resolves my concerns. I therefore maintain my positive score.

---

> > > ### Author Response · Authors · 2026-04-03
> > >
> > > Thank you for your thoughtful feedback, we sincerely appreciate your positive comments.

---

### Decision · Program_Chairs · 2026-04-30

**Decision:**

Accept (regular)

**Comment:**

This paper studies under which conditions a finite-horizon linear MDP can be attacked by reward poisoning.
The reward poisoning formulation is established in the literature but is mostly focused on designing robust algorithms.
In contrast, this paper characterizes conditions under which the environment becomes vulnerable in the first place - showing that often there is no need for deploying a robust algorithm (that typically comes with other trade-offs).

All reviewers agree that it is a solid contribution and admit that the authors resolved remaining concerns and weaknesses.
In this light, it is a clear accept.